# PARALLEL SIMULATION FOR SAMPLING UNDER ISOPERIMETRY AND SCORE-BASED DIFFUSION MODELS

## ABSTRACT

In recent years, there has been a surge of interest in proving discretization bounds for sampling under isoperimetry and for diffusion models. As data size grows, reducing the iteration cost becomes an important goal. Inspired by the great success of the parallel simulation of the initial value problem in scientific computation, we propose parallel Picard methods for sampling tasks. Rigorous theoretical analysis reveals that our algorithm achieves better dependence on dimension $d$ than prior works in iteration complexity (i.e., reduced from $\widetilde{\mathcal{O}}(\mathrm{poly}(\log d))$ to $\widetilde{\mathcal{O}}(\log d)$), which is even optimal for sampling under isoperimetry with specific iteration complexity. Our work highlights the potential advantages of simulation methods in scientific computation for dynamics-based sampling and diffusion models.

## 1 INTRODUCTION

We study the problem of sampling from a probability distribution with density $\pi(\boldsymbol{x}) \propto \exp(-f(\boldsymbol{x}))$ where $f : \mathbb{R}^d \to \mathbb{R}$ is a smooth potential. We consider two types of setting. **Problem (a):** the distribution is known only up to a normalizing constant (Chewi, 2023), and this kind of problem is fundamental in many fields such as Bayesian inference, randomized algorithms, and machine learning (Marin et al., 2007; Nakajima et al., 2019; Robert et al., 1999). **Problem (b):** known as the score-based generative models (SGMs) (Song & Ermon, 2019), we are given an approximation of $\nabla \log \pi_t$, where $\pi_t$ is the density of a specific process at time $t$. The law of this process converges to $\pi$ over time. SGMs are now the state-of-the-art in many fields, such as computer vision and image generation (Ho et al., 2022a; Dhariwal & Nichol, 2021), audio and video generation (Ho et al., 2022b; Yang et al., 2023), and inverse problems (Song et al., 2021).

For Problem (a), specifically log-concave sampling, starting from the seminal papers of Dalalyan & Tsybakov (2012), Dalalyan (2017), and Durmus & Moulines (2017), there has been a flurry of recent works on proving non-asymptotic guarantees based on simulating a process which converges to $\pi$ over time (Wibisono, 2018; Vempala & Wibisono, 2019; Altschuler & Talwar, 2022; Mou et al., 2021). Moreover, these processes, such as Langevin dynamics, converge exponentially quickly to $\pi$ under mild conditions (Dalalyan, 2017; Bernard et al., 2022; Mou et al., 2021). Such dynamics-based algorithms for Problem (a) share a common feature with the inference process of SGMs that they are actually a numerical simulation of an initial-value problem of differential equations (Hodgkinson et al., 2021). Thanks to the exponentially fast convergence of the process, significant efforts have been conducted on discretizing these processes using numerical methods such as the forward Euler, backward Euler (proximal method), exponential integrator, mid-point, and high-order Runge-Kutta methods (Vempala & Wibisono, 2019; Wibisono, 2019; Oliva & Akyildiz, 2024; Shen & Lee, 2019; Li et al., 2019).

Furthemore, in recent years, there have been increasing interest and significant advances in understanding the convergence of inherently dynamics-based SGMs (De Bortoli, 2022; Lee et al., 2023; Chen et al., 2024b; 2022; Tang & Zhao, 2024; Pedrotti et al., 2023; Li & Yan, 2024). Notably, polynomial-time convergence guarantees have been established (Chen et al., 2022; 2024b; Benton et al., 2024; Liang et al., 2024), and various discretization schemes for SGMs have been analyzed (Lu et al., 2022a;b; Huang et al., 2024).

Table 1: Comparison with existing parallel methods for sampling under isoperimetry.

| Work dynamics | Measure | Iteration Complexity | Space Complexity |
|---|---|---|---|
| (Shen & Lee, 2019, Theorem 4) underdamped Langevin diffusion | $W_2$ | $\widetilde{\mathcal{O}}\left(\operatorname{poly}\log\left(\frac{\sqrt{d}}{\varepsilon}\right)\right)$ | $\widetilde{\mathcal{O}}\left(\frac{d^{3/2}}{\varepsilon}\right)$ |
| (Yu & Dalalyana, 2024, Corollary 2) underdamped Langevin diffusion | $W_2$ | $\widetilde{\mathcal{O}}\left(\operatorname{poly}\log\left(\frac{d}{\varepsilon^2}\right)\right)$ | $\widetilde{\mathcal{O}}\left(\frac{d^{3/2}}{\varepsilon}\right)$ |
| (Anari et al., 2024, Theorem 13) overdamped Langevin diffusion | KL | $\widetilde{\mathcal{O}}\left(\operatorname{poly}\log\left(\frac{d}{\varepsilon^2}\right)\right)$ | $\widetilde{\mathcal{O}}\left(\frac{d^2}{\varepsilon^2}\right)$ |
| (Anari et al., 2024, Theorem 15) underdamped Langevin diffusion | KL | $\widetilde{\mathcal{O}}\left(\operatorname{poly}\log\left(\frac{d}{\varepsilon^2}\right)\right)$ | $\widetilde{\mathcal{O}}\left(\frac{d^{3/2}}{\varepsilon}\right)$ |
| Theorem 4.3 overdamped Langevin diffusion | KL | $\widetilde{\mathcal{O}}\left(\log\left(\frac{d}{\varepsilon^2}\right)\right)$ | $\widetilde{\mathcal{O}}\left(\frac{d^2}{\varepsilon^2}\right)$ |

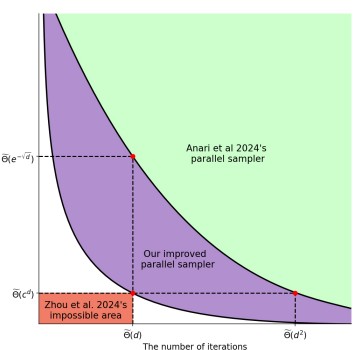

Figure 1: Comparison with existing parallel methods and lower bound for sampling under isoperimetry.

The algorithms underlying the above results are highly sequential. However, with the increasing size of data sets for sampling, we need to develop a theory for algorithms with limited iterations. For example, the widely-used denoising diffusion probabilistic models (Ho et al., 2020) may take 1000 denoising steps to generate one sample, while the evaluations of a neural network-based score function can be computationally expensive (Song et al., 2020).

As a comparison, recently, the (naturally parallelizable) Picard methods for diffusion models reduced the number of steps to around 50 (Shih et al., 2024). Furthermore, in terms of the dependency on the dimension $d$ and accuracy $\varepsilon$, Picard methods for both Problems (a) and (b) were proven to be able to return an $\varepsilon$-accurate solution within $\mathcal{O}(\operatorname{poly}(\log d))$ iterations, improved from previous $\mathcal{O}(d^a)$ with some $a > 0$. However, for Problem (a), a large gap remains relative to the recent lower bound shown in Zhou et al. (2024), and the $\mathcal{O}(\operatorname{poly}(\log d))$ iteration complexity is not yet optimal for diffusion models.

OUR CONTRIBUTIONS

In this work, we propose a novel sampling method that employs a highly parallel discretization approach for continuous processes, with applications to the overdamped Langevin diffusion and the stochastic differential equation (SDE) implementation of processes in SGMs for Problems (a) and (b), respectively.

**Faster parallel sampling under isoperimetry**[1]. We first present an improved result for parallel sampling from a distribution satisfying the log-Sobolev inequality and log-smoothness. Specifically, we improve the upper bound from $\widetilde{\mathcal{O}}\left(\log^2\left(\frac{d}{\varepsilon^2}\right)\right)$ (Anari et al., 2024) to $\widetilde{\mathcal{O}}\left(\log\left(\frac{d}{\varepsilon^2}\right)\right)$, with slightly scaling the number of processors and gradient evaluations from $\mathcal{O}\left(\frac{d}{\varepsilon^2}\right)$ to $\mathcal{O}\left(\frac{d}{\varepsilon^2}\log\left(\frac{d}{\varepsilon^2}\right)\right)$. Furthermore, our result matches the recent lower bound for log-concave distributions shown in Zhou et al. (2024) for almost linear iterations and exponentially small accuracy. We summarize the comparison in Figure 1.

Compared with methods based on underdamped Langevin diffusion, our method exhibits higher space complexity[2]. This is primarily because underdamped Langevin diffusion typically follows a smoother trajectory than overdamped Langevin diffusion, allowing for larger grid spacing and consequently, a reduced number of grids. We summarize the comparison in Table 1. In this paper, we will focus on the iteration complexity and discretization schemes for overdamped Langevin diffusion.

**Faster parallel sampling for diffusion models.** We then present an improved result for diffusion models. Specifically, we propose an efficient algorithm with $\widetilde{\mathcal{O}}\left(\log\left(\frac{d}{\varepsilon^2}\right)\right)$ iteration complexity for

---

[1]In this work, we refer isoperimetry as the condition under which the target distribution satisfies the log-Sobolev inequality. More generally, isoperimetry refers to isoperimetric inequalities that are implied by the functional inequality such as the log-Sobolev inequality (Boucheron et al., 2003).

[2]We note, in this paper, that the space complexity refers to the number of words (Chen et al., 2024a; Cohen-Addad et al., 2023) instead of the number of bits (Goldreich, 2008) to denote the approximate required storage.

Table 2: Comparison with existing parallel methods for sampling for diffusion models.

| Work Implementation | Measure | Iteration Complexity | Space Complexity |
|---|---|---|---|
| (Chen et al., 2024a, Theorem 3.3) SDE / Picard method | KL | $\widetilde{\mathcal{O}}\left(\text{poly}\log\left(\frac{d}{\varepsilon^2}\right)\right)$ | $\widetilde{\mathcal{O}}\left(\frac{d^2}{\varepsilon^2}\right)$ |
| (Chen et al., 2024a, Theorem 3.5) ODE / Picard method | TV | $\widetilde{\mathcal{O}}\left(\text{poly}\log\left(\frac{d}{\varepsilon^2}\right)\right)$ | $\widetilde{\mathcal{O}}\left(\frac{d^{3/2}}{\varepsilon^2}\right)$ |
| (Gupta et al., 2024, Theorem B.13) ODE / Parallel midpoint method | TV | $\widetilde{\mathcal{O}}\left(\text{poly}\log\left(\frac{d}{\varepsilon^2}\right)\right)$ | $\widetilde{\mathcal{O}}\left(\frac{d^{3/2}}{\varepsilon^2}\right)$ |
| Theorem 5.4 SDE / Parallel Picard method | KL | $\widetilde{\mathcal{O}}\left(\log\left(\frac{d}{\varepsilon^2}\right)\right)$ | $\widetilde{\mathcal{O}}\left(\frac{d^2}{\varepsilon^2}\right)$ |

SDE implementations of diffusion models (Song & Ermon, 2019). Our method surpasses all the existing parallel methods for diffusion models having $\widetilde{\mathcal{O}}\left(\text{poly}\log\left(\frac{d}{\varepsilon^2}\right)\right)$ iteration complexity (Chen et al., 2024a; Gupta et al., 2024), with slightly increasing the number of the processors and gradient evaluations and the space complexity for SDEs. We summarize the comparison in Table 2. Similarly, the better space complexity of the ordinary differential equation (ODE) implementations is attributed to the smoother trajectories of ODEs, which are more readily discretized.

## 2 PROBLEM SET-UP

In this section, we introduce some preliminaries and key ingredients of sampling under isoperimetry and diffusion models in Sections 2.1 and 2.2, respectively. Subsequently, the basics of Picard iterations are introduced in Section 2.3.

### 2.1 SAMPLING UNDER ISOPERIMETRY

**Problem (a)** (**Sampling task**). *Given the potential function $f : \mathcal{D} \to \mathbb{R}$, the goal of the sampling task is to draw a sample from the density $\pi_f = Z_f^{-1}\exp(-f)$, where $Z_f := \int_{\mathcal{D}}\exp(-f(\boldsymbol{x}))\mathrm{d}\boldsymbol{x}$ is the normalizing constant.*

**Distribution and function class.** If $f$ is (strongly) convex, the density $\pi_f$ is said to be (strongly) *log-concave*. If $f$ is twice-differentiable and $\nabla^2 f \preceq \beta\boldsymbol{I}$ (where $\preceq$ denotes the Loewner order and $\boldsymbol{I}$ is the identity matrix), we say the potential $f$ is $\beta$-*smooth* and the density $\pi_f$ is $\beta$-*log-smooth*.

We say $\pi$ satisfies a *log-Sobolev inequality* (LSI) with constant $\alpha > 0$ if for all smooth $f : \mathbb{R} \to \mathbb{R}$,

$$\mathsf{Ent}_\pi[f^2] := \mathbb{E}_\pi[f^2\log(f^2/\mathbb{E}_\pi(f^2))] \leq \frac{2}{\alpha}\mathbb{E}_\pi[\|\nabla f\|^2],$$

where $\|\cdot\|$ represents the $l_2$-norm. By the Bakry–Émery criterion (Bakry & Émery, 2006), if $\pi$ is $\alpha$-strongly log-concave then $\pi$ satisfies LSI with constant $\alpha$.

We define *relative Fisher information* of probability density $\rho$ w.r.t. $\pi$ as $\mathsf{FI}(\rho\|\pi) = \mathbb{E}_\rho[\|\nabla\log(\rho/\pi)\|^2]$ and the *Kullback–Leibler (KL) divergence* of $\rho$ from $\pi$ as $\mathsf{KL}(\rho\|\pi) = \mathbb{E}_\rho\log(\rho/\pi)$. By taking $f = \sqrt{\rho/\pi}$ in the above definition of the LSI The LSI is equivalent to the following relation between KL divergence and Fisher information:

$$\mathsf{KL}(\rho\|\pi) \leq \frac{1}{2\alpha}\mathsf{FI}(\rho\|\pi) \text{ for all probability measures } \rho.$$

**Langevin Dynamics.** One of the most commonly-used dynamics for sampling is Langevin dynamics (Chewi, 2023), which is the solution to the following SDE, $\mathrm{d}\boldsymbol{x} = -\nabla f(\boldsymbol{x})\mathrm{d}t + \sqrt{2}\mathrm{d}\boldsymbol{B}_t$, where $(\boldsymbol{B}_t)_{t\in[0,T]}$ is a standard Brownian motion in $\mathbb{R}^d$. If $\pi \propto \exp(-f)$ satisfies an LSI, then the law of the Langevin diffusion converges exponentially fast to $\pi$ (Bakry et al., 2014).

**Score function for sampling task.** We assume the score function $\boldsymbol{s} : \mathbb{R}^d \to \mathbb{R}$ is a pointwise accurate estimate of $\nabla V$, i.e., $\|\boldsymbol{s}(\boldsymbol{x}) - \nabla V(\boldsymbol{x})\| \leq \delta$ for all $\boldsymbol{x} \in \mathbb{R}^d$ and some sufficiently small $\delta \in \mathbb{R}_+$.

**Measures of the output.** For two densities $\rho$ and $\pi$, we define the *total variation* (TV) as $\mathsf{TV}(\rho, \pi) = \sup\{\rho(E) - \pi(E) \mid E \text{ is an event}\}$. We have the following relation between the KL divergence and TV distance, known as the *Pinsker inequality*,

$$\mathsf{TV}(\rho, \pi) \leq \sqrt{\frac{1}{2}\mathsf{KL}(\rho\|\pi)}.$$

We denote by $\mathsf{W}_2$ the *Wasserstein distance* between $\rho$ and $\pi$, which is defined as $\mathsf{W}_2^2(\rho, \pi) = \inf\left\{\mathbb{E}_{(X,Y)\sim\Pi}\left[\|X - Y\|^2\right] \mid \Pi \text{ is a coupling of } \rho, \pi\right\}$, where the infimum is over coupling distributions $\prod$ of $(X, Y)$ such that $X \sim \rho, Y \sim \pi$. If $\pi$ satisfies an LSI with constant $\alpha$, the following transport-entropy inequality, known as Talagrand's $\mathsf{T}_2$ inequality, holds (Otto & Villani, 2000) for all $\rho \in \mathcal{P}_2(\mathbb{R}^d)$, i.e., with finite second moment,

$$\frac{\alpha}{2}\mathsf{W}_2^2(\rho, \pi) \leq \mathsf{KL}(\rho\|\pi).$$

**Complexity.** For any sampling algorithm, we consider the *iteration complexity* defined as unparallelizable evaluations of the score function (Chen et al., 2024a; Zhou et al., 2024), and use the notion of the *space complexity* to denote the approximate required storage during the inference. We note, in this paper, that the space complexity refers to the number of words (Chen et al., 2024a; Cohen-Addad et al., 2023) instead of the number of bits (Goldreich, 2008) to denote the approximate required storage.

### 2.2 SCORE-BASED DIFFUSION MODELS

**Sampling for diffusion models.** In score-based diffusion models, one considers forward process $(\boldsymbol{x}_t)_{t\in[0,T]} \in \mathbb{R}^d$ governed by the canonical Ornstein-Uhlenbeck (OU) process (Ledoux, 2000):

$$\mathrm{d}\boldsymbol{x}_t = -\boldsymbol{x}_t\mathrm{d}t + \mathrm{d}\boldsymbol{B}_t, \qquad \boldsymbol{x}_0 \sim \boldsymbol{q}_0, \qquad t \in [0, T], \tag{1}$$

where $\boldsymbol{q}_0$ is the initial distribution over $\mathbb{R}^d$. The corresponding backward process $(\bar{\boldsymbol{x}}_t)_{t\in[0,T]} \in \mathbb{R}^d$ follows an SDE defined as

$$\mathrm{d}\bar{\boldsymbol{x}}_t = -\left[\frac{1}{2}\bar{\boldsymbol{x}}_t + \nabla\log\bar{p}_t(\bar{\boldsymbol{x}}_t)\right]\mathrm{d}t + \mathrm{d}\boldsymbol{B}_t, \qquad \bar{\boldsymbol{x}}_0 \sim \boldsymbol{p}_0 \approx \mathcal{N}(\boldsymbol{0}_d, \boldsymbol{I}_d), \qquad t \in [0, T], \tag{2}$$

where $\mathcal{N}(\cdot, \cdot)$ represents the normal distribution over $\mathbb{R}^d$. In practice, the score function $\nabla\log\bar{p}_t(\bar{\boldsymbol{x}}_t)$ is estimated by neural network (NN) $\boldsymbol{s}_t^\theta : \mathbb{R}^d \mapsto \mathbb{R}^d$, where $\theta$ is the parameters of NN. The backward process is approximated by

$$\mathrm{d}\boldsymbol{y}_t = -\left[\frac{1}{2}\boldsymbol{y}_t + \boldsymbol{s}_t^\theta(\boldsymbol{y}_t)\right]\mathrm{d}t + \mathrm{d}\boldsymbol{B}_t, \qquad \boldsymbol{y}_0 \sim \mathcal{N}(\boldsymbol{0}_d, \boldsymbol{I}_d), \qquad t \in [0, T]. \tag{3}$$

**Problem (b)** (**Sampling for SGMs**). *Given the learned NN-based score function $\boldsymbol{s}_t^\theta$, the goal is to simulate the approximated backward process such that the law of the output is close to $\boldsymbol{q}_0$.*

**Distribution class.** For SGMs, we assume the data density $p_0$ has finite second moments and is normalized such that $\mathrm{cov}_{p_0}(\boldsymbol{x}_0) = \mathbb{E}_{p_0}\left[(\boldsymbol{x}_0 - \mathbb{E}_{p_0}[\boldsymbol{x}_0])(\boldsymbol{x}_0 - \mathbb{E}_{p_0}[\boldsymbol{x}_0])^\top\right] = \boldsymbol{I}_d$. Such a finite moment assumption is standard across previous theoretical works on SGMs (Chen et al., 2023; 2024b; 2022) and we adopt the normalization to simplify true score function-related computations as Benton et al. (2024) and Chen et al. (2024a) did.

**OU process and inverse process** The OU process and its inverse process also converge to the target distribution exponentially fast in various divergences and metrics such as the 2-Wasserstein metric $\mathsf{W}_2$; see Ledoux (2000). Furthermore, under mild conditions, the backward process (Eq. (2)) and its approximation version (Eq. (3)) contract exponentially, with TV between their distributions diminishing exponentially as time progresses (Huang et al. (2024, Theorem 3.5) or setting the step size $h \to 0$ for the results in Chen et al. (2023; 2024b; 2022)).

**Score function for SGMs.** For the NN-based score, we assume the score function is $L^2$-accurate, bounded and Lipschitz; we defer the details in Section 5.2.

## 2.3 PICARD ITERATIONS

Consider the integral form of the initial value problem, $\boldsymbol{x}_t = \boldsymbol{x}_0 + \int_0^t f_t(\boldsymbol{x}_s)\mathrm{d}s + \sqrt{2}\boldsymbol{B}_t$. The main idea (Clenshaw, 1957) is to approximate the difference over time slice $[t_n, t_{n+1}]$ as

$$\begin{aligned}
\boldsymbol{x}_{t_{n+1}} - \boldsymbol{x}_{t_n} &= \int_{t_n}^{t_{n+1}} f_t(\boldsymbol{x}_s)\mathrm{d}s + \sqrt{2}(\boldsymbol{B}_{t_{n+1}} - \boldsymbol{B}_{t_n}) \\
&\approx \sum_{i=1}^{M} \boldsymbol{w}_i f_t(\boldsymbol{x}_i)\mathrm{d}s + \sqrt{2}(\boldsymbol{B}_{t_{n+1}} - \boldsymbol{B}_{t_n}),
\end{aligned}$$

with a discrete grid of $M$ collocation points as $\boldsymbol{x}_{t_n} = \boldsymbol{x}_0 \leq \boldsymbol{x}_1 \leq \cdots \leq \boldsymbol{x}_M = \boldsymbol{x}_{t_{n+1}}$. We update the points in a wave-like fashion, which inherently allows for parallelization:

$$\boldsymbol{x}_i^{p+1} = \boldsymbol{x}_0 + \sum_{i=1}^{M} \boldsymbol{w}_i f_t(\boldsymbol{x}_i^p) + \sqrt{2}(\boldsymbol{B}_i - \boldsymbol{B}_{t_n}), \quad \text{for } i = 1, \ldots, M.$$

Various collocation points have been proposed, including uniform points and Chebyshev points (Bai & Junkins, 2011). In this paper, however, we focus exclusively on the simplest case of uniform points, and extension to other cases is future work. Picard iterations are known to converge exponentially fast and, under certain conditions, even factorially fast for ODEs and backward SDEs (Hutzenthaler et al., 2021).

## 3 TECHNICAL OVERVIEW

We adopt the time splitting for the time horizon used in the existing parallel methods (Gupta et al., 2024; Chen et al., 2024a; Anari et al., 2024; Yu & Dalalyana, 2024; Shen & Lee, 2019). Our algorithm, however, depart crucially from prior work in the design of parallelism across the time slices, and the modification for controlling the score estimation error. Below we summarize these notion contributions and technical novelties.

**Recap of existing parallel sampling methods.** Existing works for parallel sampling apply the following generic discretization schemes (Gupta et al., 2024; Chen et al., 2024a; Anari et al., 2024; Yu & Dalalyana, 2024; Shen & Lee, 2019). At a high level, these methods divide the time horizon into many large time slices and each slice is further subdivided into grids with a small enough step size. Instead of sequentially updating the grid points, they update all grids at the same time slice simultaneously using exponentially fast converging Picard iterations (Alexander, 1990), or randomized midpoint methods (Shen & Lee, 2019; Yu & Dalalyana, 2024; Gupta et al., 2024). With $\widetilde{\mathcal{O}}(\log d)$ Picard iterations for $\widetilde{\mathcal{O}}(\log d)$ time slices, the total iteration complexity of their algorithms is $\widetilde{\mathcal{O}}(\log^2 d)$. However, while sequential updating of each time slice is not necessary for simulating the process, it remains unclear how to parallelize across time slices for sampling to obtain $\mathcal{O}(\log d)$ time complexity.

**Algorithmic novelty: parallel methods across time slices.** Naively, if we directly update all the grids simultaneously, the Picard iterations will not converge when the total length is $T = \widetilde{\mathcal{O}}(\log d)$. Instead of updating all time slices together or updating the time slice sequentially, we update the time slices in a *diagonal* style as illustrated in Figure 2. For any $j$-th update at the $n$-th time slice (corresponding the rectangle in the $n$-th column from the left and the $j$-th row from the top in Figure 2), there will be two inputs: (a) the right boundary point of the previous time slice, which has been updated $j$ times, and (b) the points on the girds of the same time slice that have been updated $j - 1$ times. Then we perform $P$ times Picard iterations with these inputs, where the hyperparameter $P$ depends on the smoothness of the score function. The main difference compared to the existing Picard methods is that for a fixed time slice, the starting points in our method are updated gradually, whereas in existing methods, the starting points remain fixed once processed.

**Challenges for convergence.** Similar to the arguments for sequentially updating the time slices, we use the standard techniques such as the interpolation method or Girsanov's theorem (Chewi, 2023; Vempala & Wibisono, 2019; Oksendal, 2013) and decompose the total error w.r.t. KL into three components: (i) convergence error of the continuous process, (ii) discretization error, and (iii) score estimation error. For (i) the convergence error of the continuous process, it is rather straightforward

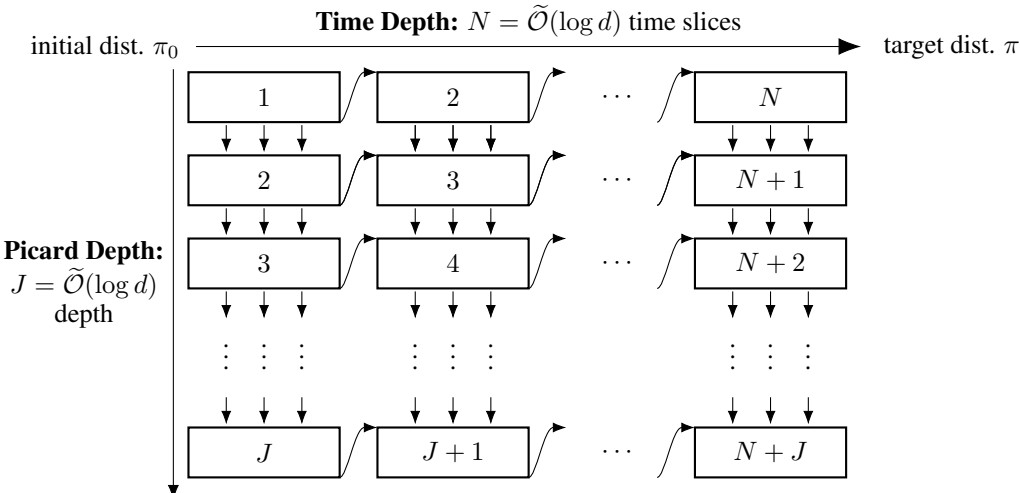

Figure 2: Illustration of the parallel Picard method: each rectangle represents an update, and the number within each rectangle indicates the index of the Picard iteration. The approximate time complexity is $N + J = \widetilde{\mathcal{O}}(\log d)$.

to control and is actually independent of the specific method used to update the time slices. The technical challenges rise from controlling the remaining two errors, which we summarize below.

*(ii) Discretization error:* Discretization error mainly arise from the truncation errors on discrete grids with the grids gap as $\mathcal{O}(1/d)$. In existing parallel methods, the sequential update across time slices benefits the convergence of truncation errors along the time direction. Assuming the truncation errors in the previous time slice have converged, its right boundary serves as the starting point for all grids in the current $O(1)$-length time slice which results in an initial bias of $\mathcal{O}(d)$. Subsequently, by performing $\mathcal{O}(\log d)$ exponentially fast Picard iterations, the truncation error will converge. However, in our diagonal-style updating scheme across time, the truncation error interacts with inputs from both the previous time slice and prior updates in the same time slice. Consequently, the bias-convergence loop that holds in sequential updating no longer holds.

*(iii) Score estimation error:* If the score function itself is Lipschitz continuous (Assumption 5.3 for Problem (b)), no additional score matching error will arise during the Picard iterations. This allows the total score estimation error to remain bounded under mild conditions (Assumption 5.1). However, for Problem (a), since it is the velocity field $\nabla f$ instead of the score function $s$ that is Lipschitz, additional score estimation errors will occur during each update. For the sequential algorithm, these additional score estimation errors are contained within the bias-convergence loop, ensuring the total score estimation error remains to be bounded. Conversely, for our diagonal-style updating algorithm, the absence of convergence along the time direction causes these additional score estimation errors to accumulate exponentially over the time direction.

**Technical novelty.** Our technical contributions address these challenges by the appropriate selection of the number of Picard iterations within each update $P$ and the depth of the Picard iterations $J$. We outline the details of the choices below.

In the following, we assume that the truncation error at the $n$-th time slice and the $j$-th iteration scales with $L_n^j$, and that the additional score estimation error for each update scales with $\delta^2$.

To address the initial challenge related to the truncation error, we choose the Picard depth as $J = \mathcal{O}(N + \log d)$. We first bound the error of the output for each update with respect to its inputs as $L_n^j \leq \mathsf{a} L_{n-1}^j + \mathsf{b} L_n^{j-1}$, where $\mathsf{a}$ and $\mathsf{b}$ are constants. By carefully choosing the length of the time slices, we can ensure that $\mathsf{b} < 1$ along the Picard iteration direction. Consequently, the truncation error will converge if the iteration depth $J$ is sufficiently large, such that $\mathsf{a}^N \mathsf{b}^J$ is sufficiently small. This requirement implies that $J = \mathcal{O}(N + \log d)$.

To mitigate the additional score estimation error for Problem (a), we perform $P$ Picard iterations within each update. The interaction between the truncation error and additional score estimation error

can be expressed as $L_n^j \leq \mathsf{a} L_{n-1}^j + \mathsf{b} L_n^{j-1} + \mathsf{c}\delta^2$, where $\mathsf{a}, \mathsf{b}, \mathsf{c}$ are constants. To ensure the total score estimation error remains bounded, it is necessary to have $\mathsf{a}, \mathsf{b} < 1$, which guarantees convergence along both the time and Picard directions. By the convergence of the Picard iteration, we can achieve $\mathsf{b} < 1$. For $\mathsf{a}$, the right boundary point of the previous time slice, and prior updates within the same time slice introduce discrepancies in the truncation error. For the impact from the previous time slice, we make use of the contraction of gradient decent to ensure convergence. However, since the grid gap scale as $1/d$, the contraction factor is close to 1. Consequently, we have to minimize the impact from prior updates within the same time slice, which scales as $\mathcal{O}(1)$ by repeating $P = \log \mathcal{O}(1)$ Picard iterations for each update.

**Balance between time and Picard directions.** We note that the Picard method, despite being the simplest approach for time parallelism, has achieved optimal performance in certain specific settings. On the one hand, the continuous processes need to run for at least $\mathcal{O}(\log d)$ time. To ensure convergence within every time slice, the time slice length have to be set as $\mathcal{O}(1)$, resulting in a necessity for at least $\mathcal{O}(\log d)$ iterations. On the other hand, with a proper initialization $\mathcal{O}(d)$, Picard iterations converge within $\mathcal{O}(\log d)$ iterations. Our parallelization balances the convergence of the continuous diffusion and the Picard iterations to achieve the improved results.

**Realed works in scientific computation.** Similar parallelism across time slices has also been proposed in scientific computation (Gear, 1991; Ong & Schroder, 2020; Gander, 2015), especially for parallel Picard iterations (Wang, 2023). Compared with prior work in scientific computation, our approach exhibits several significant differences. Firstly, our primary objective differs from that in simulation. In sampling, we aim to ensure that the output distribution closely approximates the target distribution, whereas simulation seeks to make each point on the discrete grid closely match the true dynamics. Second, our algorithm differs significantly from that of Wang (2023). In our algorithm, each update takes the inputs without the corrector operation. Furthermore, we perform $P$ Picard iterations in each update to prevent error accumulation over time $T = \widetilde{\mathcal{O}}(\log d)$. In comparison, the algorithm proposed in Wang (2023) performs a single Picard iteration in each update for simulation on a finite time interval. However, these two fields are connected through the sampling strategies that ensure each discrete point closely approximates the true process at every sampling step.

## 4 PARALLEL PICARD METHOD FOR SAMPLING UNDER ISOPERIMETRY

In this section, we present parallel Picard methods for sampling under isoperimetry (Algorithm 1) and show it holds improved convergence rate w.r.t. the KL divergence and total variance under an Log-Sobolev Inequality (Theorem 4.3 and Corollary 4.4). We illustrate the algorithm in Section 4.1, and give a proof sketch in Section 4.3. All the missing proofs can be found in Appendix B.

### 4.1 ALGORITHM

Our parallel Picard method for sampling under isoperimetry is summarized in Algorithm 1. In Lines 1–3, we generate the noise part and fix them. In Lines 4–7, we initialize the value at the grid via Langevin Monte Carlo (Chewi, 2023) with a stepsize $h = \mathcal{O}(1)$. In Lines 8–19, the time slices are updated in a diagonal manner within the outer loop, as illustrated in Figure 2. In Lines 11–12 and Lines 17-18, we repeat $P$ Picard iterations for each update.

**Remark 4.1.** *Parallelization should be understood as evaluating the score function concurrently, with each time slice potentially being computed in an asynchronous parallel manner, resulting in the overall $P(N + J) + N$ iteration complexity.*

**Remark 4.2.** *If provided with a warm start, initialization becomes unnecessary. Additionally, in practice, once the Picard iterations converge within a time slice, further updates are redundant. The convergence can be verified by calculating the maximum changes of values across the girds.*

### 4.2 THEORETICAL GUARANTEES

The following theorem summarizes our theoretical analysis for Algorithm 1.

---

**Algorithm 1:** Parallel Picard Method for sampling

---

**Input :** $\boldsymbol{x}_0 \sim \mu_0$, approximate score function $\boldsymbol{s} \approx \nabla f$, the number of the iterations in outer loop $J$, the number of the iteration in inner loop $P$, the number of time slices $N$, the length of time slices $h$, the number of points on each time slices $M$.

1 **for** $n = 0, \ldots, N-1$ **do**
2    **for** $m = 0, \ldots, M$ *(in parallel)* **do**
3      $B_{nh+m/Mh} = B_{nh} + \mathcal{N}(0, (mh/M)\boldsymbol{I}_d)$          $\triangleright$ generate the noise

4 **for** $n = 0, \ldots, N-1$ **do**
5    **for** $m = 0, \ldots, M$ *(in parallel)* **do**
6      $\boldsymbol{x}^j_{-1,M} = \boldsymbol{x}_0$, for $j = 0, \ldots, J$,          $\triangleright$ initialization
7      $\boldsymbol{x}^0_{n,m} = \boldsymbol{x}^0_{n-1,M} - \frac{hm}{M}\boldsymbol{s}(\boldsymbol{x}^0_{n-1,M}) + \sqrt{2}(B_{nh+mh/M} - B_{nh})$,

8 **for** $k = 1, \ldots, N$ **do**
9    **for** $j = 1, \ldots, \min\{k-1, J\}$ *and* $m = 1, \ldots, M$ *(in parallel)* **do**
10      let $n = k - j$, $\boldsymbol{x}^j_{n,0} = \boldsymbol{x}^j_{n-1,M}$, and $\boldsymbol{x}^{j,0}_{n,m} = \boldsymbol{x}^{j-1}_{n,m}$,
11      **for** $p = 1, \ldots, P$ **do**
12        $\boldsymbol{x}^{j,p}_{n,m} = \boldsymbol{x}^j_{n,0} - \frac{h}{M}\sum_{m'=0}^{m-1}\boldsymbol{s}(\boldsymbol{x}^{j,p-1}_{n,m'}) + \sqrt{2}(B_{nh+mh/M} - B_{nh})$,
13      $\boldsymbol{x}^j_{n,m} = \boldsymbol{x}^{j,P}_{n,m}$,

14 **for** $k = N+1, \ldots, N+J-1$ **do**
15    **for** $n = \max\{0, k-J\}, \ldots, N-1$ *and* $m = 1, \ldots, M$ *(in parallel)* **do**
16      let $j = k - n$, $\boldsymbol{x}^j_{n,0} = \boldsymbol{x}^j_{n-1,M}$, and $\boldsymbol{x}^{j,0}_{n,m} = \boldsymbol{x}^{j-1}_{n,m}$,
17      **for** $p = 1, \ldots, P$ **do**
18        $\boldsymbol{x}^{j,p}_{n,m} = \boldsymbol{x}^j_{n,0} - \frac{h}{M}\sum_{m'=0}^{m-1}\boldsymbol{s}(\boldsymbol{x}^{j,p-1}_{n,m'}) + \sqrt{2}(B_{nh+mh/M} - B_{nh})$,
19      $\boldsymbol{x}^j_{n,m} = \boldsymbol{x}^{j,P}_{n,m}$,

20 **return** $\boldsymbol{x}^J_{N-1,M}$.

---

**Theorem 4.3.** *Suppose the potential function $f$ is $\beta$-smooth and $\pi$ satisfies a log-Sobolev inequality with constant $\alpha$, and the score function $s$ is $\delta$-accurate. Let $\kappa = \beta/\alpha$. Suppose*

$$\beta h = 0.1, \qquad M \geq \frac{\kappa d}{\varepsilon^2}, \qquad N \geq 10\kappa \log\left(\frac{\mathsf{KL}(\mu_0 \| \pi)}{\varepsilon^2}\right), \qquad \delta \leq 0.2\sqrt{\alpha}\varepsilon,$$

$$P \geq \frac{2\log \kappa}{3} + 4 \quad \text{and} \quad J - N \geq \log\left(N^3\left(\frac{\kappa\delta^2 h + \kappa\mathsf{KL}(\mu_0\|\pi) + \kappa^2 d}{\varepsilon^2}\right)\right).$$

*then Algorithm 1 runs within $N + (N+J)P$ iterations with $MN$ queries per iteration and outputs a sample with marginal distribution $\rho$ such that*

$$\max\left\{\frac{\sqrt{\alpha}}{2}\mathsf{W}_2(\rho, \pi), \mathsf{TV}(\rho, \pi)\right\} \leq \sqrt{\frac{\mathsf{KL}(\rho, \pi)}{2}} \leq 2\varepsilon.$$

To make the guarantee more explicit, we can combine it with the following well-known initialization bound, see, e.g., Dwivedi et al. (2019, Section 3.2).

**Corollary 4.4.** *Suppose that $\pi = \exp(-f)$ is $\alpha$-strongly log-concave and $\beta$-log-smooth, and let $\kappa = \beta/\alpha$. Let $\boldsymbol{x}^\star$ be the minimizer of $f$. Then, for $\mu_0 = \mathcal{N}(\boldsymbol{x}^\star, \beta^{-1})$, it holds that $\mathsf{KL}(\mu_0\|\pi) \leq \frac{d}{2}\log \kappa$. Consequently, setting*

$$h = \frac{1}{10\beta}, \qquad N = 10\kappa \log\left(\frac{d\log \kappa}{\varepsilon^2}\right), \qquad \delta \leq 0.2\sqrt{\alpha}\varepsilon, \qquad M = \frac{\kappa d}{\varepsilon^2},$$

$$P \geq \frac{2\log \kappa}{3} + 4 \quad \text{and} \quad J - N = \mathcal{O}\left(\log \frac{\kappa^2 d\log \kappa}{\varepsilon^2}\right),$$

*then Algorithm 1 runs within $N + (N + J)P = \widetilde{\mathcal{O}}(\kappa \log \frac{d}{\varepsilon^2})$ iterations with $MN = \widetilde{\mathcal{O}}(\frac{\kappa^2 d}{\varepsilon^2} \log \frac{d}{\varepsilon^2})$ queries per iteration and outputs a sample with marginal distribution $\rho$ such that*

$$\max\left\{\frac{\sqrt{\alpha}}{2}\mathsf{W}_2(\rho, \pi), \mathsf{TV}(\rho, \pi)\right\} \leq \sqrt{\frac{\mathsf{KL}(\rho, \pi)}{2}} \leq 2\varepsilon.$$

**Remark 4.5.** *Compared to the existing parallel methods, our method improves the iteration complexity from $\mathcal{O}(\mathrm{poly}(\log \frac{d}{\varepsilon^2}))$ to $\mathcal{O}(\log \frac{d}{\varepsilon^2})$, which matches the lower bound for exponentially small accuracy shown in Zhou et al. (2024). The main drawback of our method is the sub-optimal space complexity due to its application to overdamped Langevin diffusion which has a less smooth trajectory compared to underdamped Langevin diffusion. However, we anticipate that our method could achieve comparable space complexity when adapted to underdamped Langevin diffusion.*

### 4.3 PROOF SKETCH OF THEOREM 4.3: PERFORMANCE ANALYSIS OF ALGORITHM 1

The detailed proof of Theorem 4.3 is deferred to Appendix B. By interpolation methods (Anari et al., 2024), we decompose the error w.r.t. the KL divergence into four error components (corollary B.4):

$$\mathsf{KL} \lesssim e^{-\Theta(N)}\mathsf{KL}(\mu_0\|\pi) + \sum_{n=1}^{N-1} e^{-\Theta(n)}\mathcal{E}_{N-n}^J + \frac{dh}{M} + \delta^2,$$

where $\mathcal{E}_n^j$ represents the truncation error of the grids at $n$-th time slice after $j$ update. For the right terms, with the choice of $N = \mathcal{O}(\log d/\varepsilon^2)$, $M = \mathcal{O}(dh/\varepsilon^2)$ and $\delta \leq \varepsilon$, we can conclude that

$$e^{-\Theta(N)}\mathsf{KL}(\mu_0\|\pi) + \frac{dh}{M} + \delta^2 \lesssim \varepsilon^2$$

Thus, we will focus on proving the convergence of the truncation error in the Picard iterations, and avoiding the accumulation of the score estimation error as discussed before.

Considering that the truncation error expands at most exponentially along the time direction, but diminishes exponentially with an increased depth of the Picard iterations, convergence can be achieved by ensuring that the depth of the Picard iterations surpasses the number of time slices as $J \geq N + \mathcal{O}(\log d/\varepsilon^2)$ with initialization error bounded by $\mathcal{O}(d)$ (the second part of Corollary B.7 and second part of Corollary B.9).

Due to the non-Lipschitzness of the score function, we can only bound $\mathcal{E}_n^j$ by quantity $\mathsf{a}\Delta_{n-1}^j + \mathsf{b}\mathcal{E}_n^{j-1} + \mathsf{c}\delta^2 h^2$ (Lemma B.5 and Lemma B.8), where $\Delta_{n-1}^j$ represents the truncation error from the previous time slice. To control the increase of the score error, it is essential to ensure that the coefficients $\mathsf{a}$ and $\mathsf{b}$ remain below one. To achieve this, the proof leverages the contraction properties of the gradient descent map and executes $P$ Picard iterations in each update.

## 5 PARALLEL PICARD METHOD FOR SAMPLING OF DIFFUSION MODELS

In this section, we present parallel Picard methods for diffusion models in Section 5.1 and assumptions in Section 5.2. Then we show it holds improved convergence rate w.r.t. the KL divergence (Theorem 5.4). All the missing details can be found in Appendix C.

### 5.1 ALGORITHM

Due to the space limit, we refer the readers to Appendix C.1 and Algorithm 2 for the details of our parallelization of Picard methods for diffusion models. It keeps same parallel structure as that illustrated in Figure 1. Notably, it has the following distinctions compared with parallel Picard methods for sampling (Algorithm 1):

- Instead of uniform discrete grids, we employ a shrinking step size discretization scheme towards the data end, and the early stopping technique which is unvoidable to show the convergence for diffusion models (Chen et al., 2024a). We show the details in Appendix C.1;
- We use an exponential integrator instead of the Euler-Maruyama Integrator in Picard iterations, where an additional high-order discretization error term would emerge (Chen et al., 2023), which we believe would not affect the overall $\mathcal{O}(\log d)$ iteration complexity with parallel sampling;

- Since the score function itself is Lipschitz, there will not be additional score matching error during Picard iterations. As a result, we perform single Picard iteration in one update, i.e., $P = 1$.

## 5.2 ASSUMPTIONS

Our theoretical analysis of the algorithm assumes mild conditions regarding the data distribution's regularity and the approximation properties of NNs. These assumptions align with those established in previous theoretical works, such as those described by Chen et al. (2024a; 2023; 2024b; 2022).

**Assumption 5.1** (($L^2([0, t_N])$ $\delta$-**accurate learned score**). *The learned NN-based score $s_t^\theta$ is $\delta_2$-accurate in the sense of*

$$\mathbb{E}_{\bar{p}}\left[\sum_{n=0}^{N-1}\sum_{m=0}^{M_n-1}\epsilon_{n,m}\left\|s_{t_n+\tau_{n,m}}^\theta(\bar{\boldsymbol{x}}_{t_n+\tau_{n,m}}) - \nabla\log\bar{p}_{t_n+\tau_{n,m}}(\bar{\boldsymbol{x}}_{t_n+\tau_{n,m}})\right\|^2\right] \le \delta_2^2.$$

**Assumption 5.2** (**Regular and normalized data distribution**). *The data density $p_0$ has finite second moments and is normalized such that $\mathrm{cov}_{p_0}(\boldsymbol{x}_0) = \boldsymbol{I}_d$.*

**Assumption 5.3** (**Bounded and Lipschitz learned NN-based score**). *The learned NN-based score function $s_t^\theta$ has a bounded $\mathcal{C}^1$ norm, i.e. , $\left\|\left\|s_t^\theta(\cdot)\right\|\right\|_{L^\infty([0,T])}$ with Lipschitz constant $L_{\boldsymbol{s}}$.*

## 5.3 THEORETICAL GUARANTEES

**Theorem 5.4.** *Under Assumptions 5.1, 5.2, and 5.3, given the following choices of the order of the parameters*

$$h = \Theta(1), \quad N = \mathcal{O}\left(\log\frac{d}{\varepsilon^2}\right), \quad M = \mathcal{O}\left(\frac{d}{\varepsilon^2}\log\frac{d}{\varepsilon^2}\right),$$

$$T = \mathcal{O}\left(\log\frac{d}{\varepsilon^2}\right), \quad and \quad J = \mathcal{O}\left(N + \log\frac{Nd}{\varepsilon^2}\right),$$

*the parallel Picard algorithm for diffusion models (Algorithm 2) generates samples from satisfies the following error bound,*

$$\mathsf{KL}(p_\eta\|\widetilde{q}_{t_N}) \lesssim de^{-T} + \frac{dT}{M} + \varepsilon^2 + \delta_2^2 \lesssim \varepsilon^2, \tag{4}$$

*with total $2N + J = \widetilde{\mathcal{O}}\left(\log\frac{d}{\varepsilon^2}\right)$ iteration complexity and $dM = \widetilde{\mathcal{O}}\left(\frac{d^2}{\varepsilon^2}\right)$ space complexity for parallelizable $\delta_2$-accurate score function computations.*

**Remark 5.5.** *Compared to existing parallel methods, our method improves the iteration complexity from $\mathcal{O}(\mathrm{poly}(\log\frac{d}{\varepsilon^2}))$ to $\mathcal{O}(\log\frac{d}{\varepsilon^2})$. The main drawback of our method is the sub-optimal space complexity due to its application to SDE implementations which has a less smooth trajectory compared to ODE implementations. However, we believe that our method could achieve comparable space complexity when adapted to ODE implementations.*

## 6 DISCUSSION AND CONCLUSION

In this work, we proposed novel parallel Picard methods for various sampling tasks. Notably, we obtain $\varepsilon^2$-accurate sample w.r.t. the $\mathsf{KL}$ divergence within $\widetilde{\mathcal{O}}\left(\log\frac{d}{\varepsilon^2}\right)$, which is the tight rate for exponentially small accuracy for sampling with isoperimetry and represents a significant improvement from $\widetilde{\mathcal{O}}\left(\mathrm{poly}\log\frac{d}{\varepsilon^2}\right)$ for diffusion models. Furthermore compared with the existing methods applied to the overdamped Langevin dynamics or the SDE implementations for diffusion models, our space complexity only scales by a logarithmic factor.

Several promising theoretical directions for future research emerge from our study. First, by serving as an analogue of simulation methods in scientific computation, our work demonstrates the potentials for developing rapid and efficient sampling methods through other discretization techniques for simulation. Another avenue involves exploring smoother dynamics, aiming to reduce the space complexity associated with these methods.

Lastly, although our highly parallel methods may introduce engineering challenges, such as the memory bandwidth, we believe our theoretical works will motivates the empirical development of parallel algorithms for both sampling and diffusion models.

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

## A  USEFUL TOOLS

### A.1  GIRSANOV'S THEOREM

**Theorem A.1** (**Properties of $f$-divergence**). *Suppose $p$ and $q$ are two probability measures on a common measurable space $(\Omega, \mathcal{F})$ with $p \ll q$. The $f$-divergence between $p$ and $q$ is defined as*

$$D_f(p\|q) = \mathbb{E}_X \left[ f\left( \frac{\mathrm{d}p}{\mathrm{d}q} \right) \right],$$

*where $\frac{\mathrm{d}p}{\mathrm{d}q}$ is the Radon-Nikodym derivative of $p$ with respect to $q$, and $f : \mathbb{R}^+ \to \mathbb{R}$ is a convex function. In particular, $D_f(\cdot\|\cdot)$ coincides with the Kullback–Leibler (KL) divergence when $f(x) = x \log x$ and $D_f(\cdot\|\cdot) = \mathsf{TV}$ coincides with the total variation (TV) distance when $f(x) = \frac{1}{2}|x - 1|$.*

*For the $f$-divergence defined above, we have the following properties:*

1. *(**Data-processing inequality**). Suppose $\mathcal{H}$ is a sub-$\sigma$-algebra of $\mathcal{F}$, the following inequality holds*

$$D_f(p|_{\mathcal{H}}\|q|_{\mathcal{H}}) \leq D_f(p\|q),$$

   *for any $f$-divergence $D_f(\cdot\|\cdot)$.*

2. *(**Chain rule**). Suppose $X$ is a random variable generating a sub-$\sigma$-algebra $\mathcal{F}_X$ of $\mathcal{F}$, and $p(\cdot|X) \ll q(\cdot|X)$ holds for any value of $X$, then*

$$\mathsf{KL}(p\|q) = \mathsf{KL}(p_{\mathcal{F}_X}\|q|_{\mathcal{F}_X}) + \mathbb{E}|_{\mathcal{F}_X} \left[ \mathsf{KL}(p(\cdot|X)\|q(\cdot|X)) \right].$$

Similar as Chen et al. (2024a), for the diffusion model, we consider a probability space $(\Omega, \mathcal{F}, p)$ on which $(\boldsymbol{w}_t(\omega))_{t \geq 0}$ is a Wiener process in $\mathbb{R}^d$. The Wiener process $(\boldsymbol{w}_t(\omega))_{t \geq 0}$ generates the filtration $\{\mathcal{F}_t\}_{t \geq 0}$ on the measurable space $(\Omega, \mathcal{F})$. For an Itô process $\boldsymbol{z}_t(\omega)$ with the following governing SDE:

$$\mathrm{d}\boldsymbol{z}_t(\omega) = \boldsymbol{\alpha}(t, \omega)\mathrm{d}t + \boldsymbol{\Sigma}(t, \omega)\mathrm{d}\boldsymbol{w}_t(\omega),$$

for any time $t$, we denote the marginal distribution of $\boldsymbol{z}_t$ by $p_t$, i.e.,

$$p_t := p\left( \boldsymbol{z}_t^{-1}(\cdot) \right), \quad \text{where } \boldsymbol{z}_t : \Omega \to \mathbb{R}^m, \omega \mapsto \boldsymbol{z}_t(\omega),$$

as well as the path measure of the process $z_t$ in the sense of

$$p_{t_1:t_2} := p\left( \boldsymbol{z}_{t_1:t_2}^{-1}(\cdot) \right), \quad \text{where } \boldsymbol{z}_{t_1:t_2} : \Omega \to \mathcal{C}([t_1, t_2], \mathbb{R}^m), \omega \mapsto (\boldsymbol{z}_t(\omega))_{t \in [t_1, t_2]}.$$

For the sake of simplicity, we define the following class of functions:

**Definition A.2.** *For any $0 \leq t_1 < t_2$, we define $\mathcal{V}(t_1, t_2)$ as the class of functions $f(t, \omega) : [0, +\infty) \times \Omega \to \mathbb{R}$ such that:*

1. *$f(t, \omega)$ is $\mathcal{B} \times \mathcal{F}_t$-measurable, where $\mathcal{B}$ is the Borel $\sigma$-algebra on $\mathbb{R}^d$;*

2. *$f(t, \omega)$ is $\mathcal{F}_t$-adapted for all $t \geq 0$;*

3. *The following Novikov condition holds:*

$$\mathbb{E} \left[ \exp \left( \int_{t_1}^{t_2} f^2(t, \omega)dt \right) \right] < +\infty.$$

*and $\mathcal{V} = \cap_{\epsilon > 0} \mathcal{V}(\epsilon)$. For vectors and matrices, we say it belongs to $\mathcal{V}^n(t, \omega)$ or $\mathcal{V}^{m \times n}(t, \omega)$ if each component of the vector or each entry of the matrix belongs to $\mathcal{V}(t, \omega)$.*

For such class of functions, we remind the following generalized version of Girsanov's theorem

**Theorem A.3** (**Girsanov's Theorem (Oksendal, 2013, Theorem 8.6.6)**). *Let $\boldsymbol{\alpha}(t, \omega) \in \mathcal{V}^m$, $\boldsymbol{\Sigma}(t, \omega) \in \mathcal{V}^{m \times n}$, and $(\boldsymbol{w}_t(\omega))_{t \geq 0}$ be a Wiener process on the probability space $(\Omega, \mathcal{F}, q)$. For $t \in [0, T]$, suppose $\boldsymbol{z}_t(\omega)$ is an Itô process with the following SDE:*

$$\mathrm{d}\boldsymbol{z}_t(\omega) = \boldsymbol{\alpha}(t, \omega)\mathrm{d}t + \boldsymbol{\Sigma}(t, \omega)\mathrm{d}\boldsymbol{w}_t(\omega), \tag{5}$$

*and there exist processes $\boldsymbol{\delta}(t, \omega) \in \mathcal{V}^n$ and $\boldsymbol{\beta}(t, \omega) \in \mathcal{V}^m$ such that:*

1. $\boldsymbol{\Sigma}(t,\omega)\boldsymbol{\delta}(t,\omega) = \boldsymbol{\alpha}(t,\omega) - \boldsymbol{\beta}(t,\omega)$;

2. *The process $M_t(\omega)$ as defined below is a martingale with respect to the filtration $\{\mathcal{F}_t\}_{t\geq 0}$ and probability measure $q$:*

$$M_t(\omega) = \exp\left(-\int_0^t \boldsymbol{\delta}(s,\omega)^\top \mathrm{d}\boldsymbol{w}_s(\omega) - \frac{1}{2}\int_0^t \|\boldsymbol{\delta}(s,\omega)\|^2 \mathrm{d}s\right),$$

*then there exists another probability measure $p$ on $(\Omega, \mathcal{F})$ such that:*

1. $p \ll q$ *with the Radon-Nikodym derivative $\frac{\mathrm{d}p}{\mathrm{d}q}(\omega) = M_T(\omega)$,*

2. *The process $\widetilde{\boldsymbol{w}}_t(\omega)$ as defined below is a Wiener process on $(\Omega, \mathcal{F}, p)$:*

$$\widetilde{\boldsymbol{w}}_t(\omega) = \boldsymbol{w}_t(\omega) + \int_0^t \boldsymbol{\delta}(s,\omega)\mathrm{d}s,$$

3. *Any continuous path in $\mathcal{C}([t_1, t_2], \mathbb{R}^m)$ generated by the process $\boldsymbol{z}_t$ satisfies the following SDE under the probability measure $p$:*

$$\mathrm{d}\widetilde{\boldsymbol{z}}_t(\omega) = \boldsymbol{\beta}(t,\omega)\mathrm{d}t + \boldsymbol{\Sigma}(t,\omega)\mathrm{d}\widetilde{\boldsymbol{w}}_t(\omega). \tag{6}$$

**Corollary A.4.** *Suppose the conditions in Theorem A.3 hold, then for any $t_1, t_2 \in [0, T]$ with $t_1 < t_2$, the path measure of the SDE equation 6 under the probability measure $p$ in the sense of $p_{t_1:t_2} = p(\boldsymbol{z}_{t_1:t_2}^{-1}(\cdot))$ is absolutely continuous with respect to the path measure of the SDE equation 5 in the sense of $q_{t_1:t_2} = q(\boldsymbol{z}_{t_1:t_2}^{-1}(\cdot))$. Moreover, the KL divergence between the two path measures is given by*

$$\mathsf{KL}(p_{t_1:t_2}\|q_{t_1:t_2}) = \mathsf{KL}(p_{t_1}\|q_{t_1}) + \mathbb{E}_{\omega\sim p|_{\mathcal{F}_{t_1}}}\left[\frac{1}{2}\int_{t_1}^{t_2}\|\boldsymbol{\delta}(t,\omega)\|^2\mathrm{d}t\right].$$

## A.2 COMPARISON INEQUALITIES

**Theorem A.5 (Gronwall inequality (Dragomir, 2003, Theorem 1)).** *Let $x$, $\Psi$ and $\chi$ be real continuous functions defined in $[a, b]$, $\chi(t) \geq 0$ for $t \in [a, b]$. We suppose that on $[a, b]$ we have the inequality*

$$x(t) \leq \Psi(t) + \int_a^t \chi(s)x(s)\mathrm{d}s.$$

*Then*

$$x(t) \leq \Psi(t) + \int_a^t \chi(s)\Psi(s)\exp\left[\int_s^t \chi(u)\mathrm{d}u\right]\mathrm{d}s.$$

## A.3 HELP LEMMAS FOR DIFFUSION MODELS

**Lemma A.6 (Lemma 9 in Chen et al. (2023)).** *For $\widehat{q}_0 \sim \mathcal{N}(0, I_d)$ and $\breve{p} = p_T$ is the distribution of the solution to the forward process (Eq. (2)), we have*

$$\mathsf{KL}(\breve{p}_0\|\widehat{q}_0) \lesssim de^{-T}.$$

# B MISSING PROOF FOR SAMPLING UNDER ISOPERIMETRY

## B.1 ONE STEP ANALYSIS OF $\mathsf{KL}_n^j$: FROM $\mathsf{KL}$'S CONVERGENCE TO PICARD CONVERGENCE

In this section, we use the interpolation method to analyse the change of $\mathsf{KL}_n^j$ along time direction, which will be bounded by discretization error and score error.

**Lemma B.1.** *Assume $\beta h \leq 0.1$. For any $j = 1, \ldots, J$, $n = 1, \ldots, N-1$, we have*

$$\mathsf{KL}_n^j \leq \exp(-1.2\alpha h)\mathsf{KL}_{n-1}^j + \frac{0.5\beta dh}{M} + 4.4\beta^2 h\mathcal{E}_n^j + 2.1\delta^2 h.$$

*Furthermore, for initialization part, i.e., $j = 0$, $n = 0, \ldots, N-1$, we have*

$$\mathsf{KL}_n^0 \leq \exp\left(-\alpha(n+1)h\right)\mathsf{KL}(\mu_0\|\pi) + \frac{8\beta^2 dh}{\alpha},$$

**Remark B.2.** *In the first equation, the term $\exp(-1.2\alpha h)\mathsf{KL}_{n-1}^{j}$ characterizes the convergence of the continuous diffusion. Additionally, the second and third terms quantify the discretization error. Adopting $P = 0$ and $M = 1$ reverts to the classical scenario, where the discretization error approximates $\mathcal{O}(hd)$, as discussed in Section 4.1 of Chewi (2023). Moreover, the second term is influenced by the density of the grids, while the third term is dependent on the convergence of the Picard iterations. The fourth term accounts for the score error.*

*Proof.* We will use the interpolation method and follow the proof of Theorem 13 in Anari et al. (2024). For $j \in [J]$, $n = 0, \ldots, N - 1$ and $m = 0, \ldots, M - 1$, it is easy to see that

$$\boldsymbol{x}_{n,m+1}^{j} = \boldsymbol{x}_{n,m}^{j} - \frac{h}{M}\boldsymbol{s}(\boldsymbol{x}_{n,m}^{j,P-1}) + \sqrt{2}(B_{nh+(m+1)/h} - B_{nh+mh/M}).$$

Let $\boldsymbol{x}_t$ denote the linear interpolation between $\boldsymbol{x}_{n,m+1}^{j}$ and $\boldsymbol{x}_{n,m}^{j}$, i.e., for $t \in \left[nh + \frac{mh}{M}, nh + \frac{(m+1)hh}{M}\right]$, let

$$\boldsymbol{x}_t = \boldsymbol{x}_{n,m}^{j} - \left(t - nh - \frac{mh}{M}\right)\boldsymbol{s}(\boldsymbol{x}_{n,m}^{j,P-1}) + \sqrt{2}(B_t - B_{nh+mh/M}).$$

Note that $\boldsymbol{s}(\boldsymbol{x}_{n,m}^{j,P})$ is a constant vector field. Let $\mu_t$ be the law of $\boldsymbol{x}_t$. The same argument as in (Vempala & Wibisono, 2019, Lemma 3/Equation 32) yields the differential inequality

$$\partial_t\mathsf{KL}(\mu_t\|\pi) = -\mathsf{FI}(\mu_t\|\pi) + \mathbb{E}\left\langle\nabla f(\boldsymbol{x}_t) - \boldsymbol{s}(\boldsymbol{x}_{n,m}^{j,P-1}), \nabla\log\frac{\mu_t(\boldsymbol{x}_t)}{\pi(\boldsymbol{x}_t)}\right\rangle$$

$$\leq -\frac{3}{4}\mathsf{FI}(\mu_t\|\pi) + \mathbb{E}\left[\left\|\nabla f(\boldsymbol{x}_t) - \boldsymbol{s}(\boldsymbol{x}_{n,m}^{j,P-1})\right\|^2\right], \tag{7}$$

where we used $(a, b) \leq \frac{1}{4}\|a\|^2 + \|b\|^2$ and $\mathbb{E}\left[\left\|\nabla\log\frac{\mu_t(\boldsymbol{x}_t)}{\pi(\boldsymbol{x}_t)}\right\|^2\right] = \mathsf{FI}(\mu_t\|\pi)$. For the first term, by LSI, we have $\mathsf{KL}(\mu_t\|\pi) \leq \frac{1}{2\alpha}\mathsf{FI}(\mu_t\|\pi)$. For the second term, we have

$$\mathbb{E}\left[\left\|\nabla f(\boldsymbol{x}_t) - \boldsymbol{s}(\boldsymbol{x}_{n,m}^{j,P-1})\right\|^2\right]$$

$$\leq 2\mathbb{E}\left[\left\|\nabla f(\boldsymbol{x}_t) - \nabla f(\boldsymbol{x}_{n,m}^{j,P-1})\right\|^2\right] + 2\mathbb{E}\left[\left\|\nabla f(\boldsymbol{x}_{n,m}^{j,P-1}) - \boldsymbol{s}(\boldsymbol{x}_{n,m}^{j,P-1})\right\|^2\right]$$

$$\leq 2\beta^2\mathbb{E}\left[\left\|\boldsymbol{x}_t - \boldsymbol{x}_{n,m}^{j,P-1}\right\|^2\right] + 2\delta^2. \tag{8}$$

Moreover,

$$\mathbb{E}\left[\left\|\boldsymbol{x}_t - \boldsymbol{x}_{n,m}^{j,P-1}\right\|^2\right] \leq 2\mathbb{E}\left[\left\|\boldsymbol{x}_t - \boldsymbol{x}_{n,m}^{j}\right\|^2\right] + 2\mathbb{E}\left[\left\|\boldsymbol{x}_{n,m}^{j,P} - \boldsymbol{x}_{n,m}^{j,P-1}\right\|^2\right] \tag{9}$$

For the first term, which will be influenced by density of grids, we have

$$\mathbb{E}\left[\left\|\boldsymbol{x}_t - \boldsymbol{x}_{n,m}^{j}\right\|^2\right]$$

$$\leq \left(t - nh - \frac{mh}{M}\right)^2\mathbb{E}\left[\left\|\boldsymbol{s}(\boldsymbol{x}_{n,m}^{j,P-1})\right\|^2\right] + d\left(t - nh - \frac{mh}{M}\right)$$

$$\leq \frac{h^2}{M^2}\mathbb{E}\left[\left\|\boldsymbol{s}(\boldsymbol{x}_{n,m}^{j,P-1})\right\|^2\right] + d\left(t - nh - \frac{mh}{M}\right)$$

$$\leq \frac{2h^2}{M^2}\mathbb{E}\left[\left\|\nabla f(\boldsymbol{x}_{n,m}^{j,P-1})\right\|^2\right] + \frac{2\delta^2 h^2}{M^2} + \frac{dh}{M}$$

$$\leq \frac{4\beta^2 h^2}{M^2}\mathbb{E}\left[\left\|\boldsymbol{x}_t - \boldsymbol{x}_{n,m}^{j,P-1}\right\|^2\right] + \frac{4h^2}{M^2}\mathbb{E}\left[\left\|\nabla f(\boldsymbol{x}_t)\right\|^2\right] + \frac{2\delta^2 h^2}{M^2} + \frac{dh}{M}. \tag{10}$$

Taking $\beta h \leq \frac{1}{10}$, and combining Eq. (9) and Eq. (10), we have

$$\mathbb{E}\left[\left\|\boldsymbol{x}_t - \boldsymbol{x}_{n,m}^{j,P-1}\right\|^2\right] \leq \frac{4.4h^2}{M^2}\mathbb{E}\left[\left\|\nabla f(\boldsymbol{x}_t)\right\|^2\right] + \frac{2.2\delta^2 h^2}{M^2} + \frac{1.1dh}{M} + 2.2\mathbb{E}\left[\left\|\boldsymbol{x}_{n,m}^{j} - \boldsymbol{x}_{n,m}^{j,P-1}\right\|^2\right]. \tag{11}$$

For the first term, we recall the following lemma.

**Lemma B.3** (**Lemma 16 in Chewi et al. (2024)**).

$$\mathbb{E}\left[\|\nabla f(\boldsymbol{x}_t)\|^2\right] \le \mathsf{FI}(\mu_t\|\pi) + 2\beta d.$$

Combining Eq. (7), Eq. (8), Eq. (11) and $\beta h \le \frac{1}{10}$, we have for $j \in [J]$, $n = 0, \ldots, n-1$, $m = 0, \ldots, M-1$, and $t \in \left[nh + \frac{mh}{M}, nh + \frac{(m+1)hh}{M}\right]$,

$$\partial_t \mathsf{KL}(\mu_t\|\pi)$$

$$\le -\frac{3}{4}\mathsf{FI}(\mu_t\|\pi) + \mathbb{E}\left[\left\|\nabla f(\boldsymbol{x}_t) - \boldsymbol{s}(\boldsymbol{x}_{n,m}^{j,P-1})\right\|^2\right]$$

$$\le -\frac{3}{4}\mathsf{FI}(\mu_t\|\pi) + 2\beta^2\mathbb{E}\left[\left\|\boldsymbol{x}_t - \boldsymbol{x}_{n,m}^{j,P-1}\right\|^2\right] + 2\delta^2$$

$$\le -\frac{3}{4}\mathsf{FI}(\mu_t\|\pi) + \frac{8.8\beta^2 h^2}{M^2}\mathbb{E}\left[\|\nabla f(\boldsymbol{x}_t)\|^2\right] + \frac{4.4\beta^2\delta^2 h^2}{M^2} + \frac{2.2\beta^2 dh}{M} + 4.4\beta^2\mathbb{E}\left[\left\|\boldsymbol{x}_{n,m}^{j,P} - \boldsymbol{x}_{n,m}^{j,P-1}\right\|^2\right] + 2\delta^2$$

$$\le -\frac{3}{4}\mathsf{FI}(\mu_t\|\pi) + \frac{0.1}{M^2}\mathbb{E}\left[\|\nabla V(X_t)\|^2\right] + \frac{0.1\delta^2}{M^2} + \frac{2.2\beta^2 dh}{M} + 4.4\beta^2\mathcal{E}_n^j + 2\delta^2$$

$$\le -\frac{3}{4}\mathsf{FI}(\mu_t\|\pi) + \frac{0.1}{M^2}\left(\mathsf{FI}(\mu_t\|\pi) + 2\beta d\right) + \frac{0.1\delta^2}{M^2} + \frac{2.2\beta^2 dh}{M} + 4.4\beta^2\mathcal{E}_n^j + 2\delta^2$$

$$\le -1.2\alpha\mathsf{KL}(\mu_t\|\pi) + \frac{0.5\beta d}{M} + 4.4\beta^2\mathcal{E}_n^j + 2.1\delta^2$$

Since this inequality holds independently of $m$, we integral from $t = nh$ to $t = (n+1)h$,

$$\mathsf{KL}_n^j \le \exp(-1.2\alpha h)\mathsf{KL}_{n-1}^j + \frac{0.5\beta dh}{M} + 4.4\beta^2 h\mathcal{E}_n^j + 2.1\delta^2 h.$$

As for $j = 0$, actually, Line 4-7 performs a Langevin Monte Carlo with step size $h$, by Theorem 4.2.6 in Chewi (2023), we have

$$\mathsf{KL}_n^0 \le \exp\left(-\alpha nh\right)\mathsf{KL}_0^0 + \frac{8dh\beta^2}{\alpha},$$

with $0 < h \le \frac{1}{4L}$.

$\square$

**Corollary B.4.** *Assume $\beta h \le 0.1$. We have*

$$\mathsf{KL}_{N-1}^J \le e^{-1.2\alpha(N-1)h}\left(\mathsf{KL}(\mu_0\|\pi) + 4.4\beta^2 h\Delta_0^J\right) + \sum_{n=1}^{N-1}e^{-1.2\alpha(n-1)h}4.4\beta^2 h\mathcal{E}_{N-n}^J + \frac{0.5\beta d}{\alpha M} + \frac{2.1\delta^2}{\alpha}.$$

*Furthermore, if $\mathcal{E}_{N-n}^J$ has a uniform bound as $\mathcal{E}_{N-n}^J \le \mathcal{E} + 500\delta^2 h^2$, we have*

$$\mathsf{KL}_{N-1}^J \le e^{-1.2\alpha(N-1)h}\left(\mathsf{KL}(\mu_0\|\pi) + 4.4\beta^2 h\Delta_0^J\right) + 5\beta\kappa\mathcal{E} + \frac{0.5\beta d}{\alpha M} + \frac{2.5\delta^2}{\alpha}.$$

*Proof.* By Lemma B.1, we decompose $\mathsf{KL}_{N-1}^J$ as

$$\mathsf{KL}_{N-1}^J \le e^{-1.2\alpha(N-1)h}\mathsf{KL}_0^J + \sum_{n=1}^{N-1}e^{-1.2\alpha(n-1)h}\left(\frac{0.5\beta dh}{M} + 4.4\beta^2 h\mathcal{E}_{N-n}^J + 2.1\delta^2 h\right)$$

$$\le e^{-1.2\alpha(N-1)h}\left(\mathsf{KL}(\mu_0\|\pi) + 4.4\beta^2 h\Delta_0^J\right)$$

$$\quad + \frac{4.4\beta^2 h(\mathcal{E} + 500\delta^2 h^2) + \frac{0.5\beta dh}{M} + 2.1\delta^2 h}{1 - \exp(-1.2\alpha h)}$$

$$\le e^{-1.2\alpha(N-1)h}\left(\mathsf{KL}(\mu_0\|\pi) + 4.4\beta^2 h\Delta_0^J\right) + \frac{1.1}{\alpha h}4.4\beta^2 h\mathcal{E} + \frac{1.1}{\alpha h}\frac{0.5\beta dh}{M}$$

$$\quad + \frac{1.1}{\alpha h}25\delta^2 h$$

$$= e^{-1.2\alpha(N-1)h}\left(\mathsf{KL}(\mu_0\|\pi) + 4.4\beta^2 h\Delta_0^J\right) + 5\kappa\beta\mathcal{E} + \frac{0.6\beta d}{\alpha M} + \frac{28\delta^2}{\alpha},$$

where the third inequality holds since $0 < x < 0.4$, we have $1.1 - 1.1\exp(-1.2x) - x > 0$. It is clear that $\alpha h < \beta h < 0.1$. $\square$

## B.2 ONE STEP ANALYSIS OF $\Delta_n^j$

In this section, we analyze the one step change of $\Delta_n^j$ first.

**Lemma B.5.** *Assume $\beta h = \frac{1}{10}$ and $P \geq \frac{2 \log \kappa}{3} + 4$. For any $j = 2, \ldots, J$, $n = 1, \ldots, N-1$, we have*

$$\Delta_n^j \leq \left(1 - \frac{0.005}{\kappa}\right) \Delta_{n-1}^j + 4.4 \left(\frac{1}{M} + 10\kappa\right) h^2 \delta^2 + 4.4 \left(\frac{1}{M} + 10\kappa\right) \beta^2 h^2 \mathcal{E}_n^{j-1}.$$

*Furthermore, for $j = 1$, $n = 1, \ldots, N-1$, we have*

$$\Delta_n^1 \leq \Delta_{n-1}^1 + \left(\frac{1}{M} + 10\kappa\right) \left(5\delta^2 h^2 + 6\beta^2 dh^3 + 0.4\beta^2 h^2 \frac{\mathsf{KL}_{n-1}^0}{\alpha}\right).$$

*Proof.* **Decomposition when $j \geq 2$.** In fact, for $j \in [J]$, $n = 0, \ldots, N-1$, $m = 0, \ldots, M-1$, and $p = 1, \ldots, P$, it is easy to see that

$$\boldsymbol{x}_{n,m+1}^{j,p} = \boldsymbol{x}_{n,m}^{j,p} - \frac{h}{M} \boldsymbol{s}(\boldsymbol{x}_{n,m}^{j,p-1}) + \sqrt{2}(B_{nh+(m+1)/h} - B_{nh+mh/M}).$$

For any $j = 2, \ldots, J$, $n = 1, \ldots, N-1$, by the contraction of $\phi(\boldsymbol{x}) = \boldsymbol{x} - \frac{h}{M}\nabla f(\boldsymbol{x})$ (Lemma 2.2 in Altschuler & Talwar (2022)), for any $m = 1, \ldots, M$, we have,

$$\mathbb{E}\left[\left\|\boldsymbol{x}_{n,m}^{j,P} - \boldsymbol{x}_{n,m}^{j-1,P}\right\|^2\right]$$

$$= \mathbb{E}\left[\left\|\boldsymbol{x}_{n,m-1}^{j,P} - \frac{h}{M}\boldsymbol{s}(\boldsymbol{x}_{n,m-1}^{j,P-1}) - \left(\boldsymbol{x}_{n,m-1}^{j-1,P} - \frac{h}{M}\boldsymbol{s}(\boldsymbol{x}_{n,m-1}^{j-1,P-1})\right)\right\|^2\right]$$

$$\leq (1+\eta)\mathbb{E}\left[\left\|\boldsymbol{x}_{n,m-1}^{j,P} - \frac{h}{M}\nabla f(\boldsymbol{x}_{n,m-1}^{j,P}) - \left(\boldsymbol{x}_{n,m-1}^{j-1,P} - \frac{h}{M}\nabla f(\boldsymbol{x}_{n,m-1}^{j-1,P})\right)\right\|^2\right]$$

$$+ \left(2 + \frac{2}{\eta}\right)\mathbb{E}\left[\left\|\frac{h}{M}\nabla f(\boldsymbol{x}_{n,m-1}^{j,P}) - \frac{h}{M}\nabla f(\boldsymbol{x}_{n,m-1}^{j,P-1}) + \frac{h}{M}\nabla f(\boldsymbol{x}_{n,m-1}^{j-1,P}) - \frac{h}{M}\nabla f(\boldsymbol{x}_{n,m-1}^{j-1,P-1})\right\|^2\right]$$

$$+ \left(2 + \frac{2}{\eta}\right)\mathbb{E}\left[\left\|\frac{h}{M}\nabla f(\boldsymbol{x}_{n,m-1}^{j,P-1}) - \frac{h}{M}\boldsymbol{s}(\boldsymbol{x}_{n,m-1}^{j,P-1}) + \frac{h}{M}\nabla f(\boldsymbol{x}_{n,m-1}^{j-1,P-1}) - \frac{h}{M}\boldsymbol{s}(\boldsymbol{x}_{n,m-1}^{j-1,P-1})\right\|^2\right]$$

$$\leq (1+\eta)\left(1 - \frac{\alpha h}{M}\right)^2 \mathbb{E}\left[\left\|\boldsymbol{x}_{n,m-1}^{j,P} - \boldsymbol{x}_{n,m-1}^{j-1,P}\right\|^2\right] + \left(4 + \frac{4}{\eta}\right)\frac{h^2}{M^2}\delta^2$$

$$+ \left(4 + \frac{4}{\eta}\right)\frac{\beta^2 h^2}{M^2}\mathbb{E}\left[\left\|\boldsymbol{x}_{n,m-1}^{j,P} - \boldsymbol{x}_{n,m-1}^{j,P-1}\right\|^2\right] + \left(4 + \frac{4}{\eta}\right)\frac{\beta^2 h^2}{M^2}\mathbb{E}\left[\left\|\boldsymbol{x}_{n,m-1}^{j-1,P} - \boldsymbol{x}_{n,m-1}^{j-1,P-1}\right\|^2\right].$$

By setting $\eta = \frac{\alpha h}{M} = \frac{1}{10\kappa M}$, we have

$$\mathbb{E}\left[\left\|\boldsymbol{x}_{n,M}^{j,P} - \boldsymbol{x}_{n,M}^{j-1,P}\right\|^2\right]$$

$$\leq \left(1 - \frac{\alpha h}{M}\right)^M \mathbb{E}\left[\left\|\boldsymbol{x}_{n,0}^{j,P} - \boldsymbol{x}_{n,0}^{j-1,P}\right\|^2\right] + \left(4 + \frac{4}{\eta}\right)\frac{h^2}{M}\delta^2$$

$$+ \sum_{m=1}^M \left(4 + \frac{4}{\eta}\right)\frac{\beta^2 h^2}{M^2}\mathbb{E}\left[\left\|\boldsymbol{x}_{n,m-1}^{j,P} - \boldsymbol{x}_{n,m-1}^{j,P-1}\right\|^2\right]$$

$$+ \sum_{m=1}^M \left(4 + \frac{4}{\eta}\right)\frac{\beta^2 h^2}{M^2}\mathbb{E}\left[\left\|\boldsymbol{x}_{n,m-1}^{j-1,P} - \boldsymbol{x}_{n,m-1}^{j-1,P-1}\right\|^2\right]$$

$$\leq \exp(-\alpha h)\Delta_{n-1}^j + \left(4 + \frac{4}{\eta}\right)\frac{h^2}{M}\delta^2 + \left(4 + \frac{4}{\eta}\right)\frac{\beta^2 h^2}{M}\mathcal{E}_n^j + \left(4 + \frac{4}{\eta}\right)\frac{\beta^2 h^2}{M}\mathcal{E}_n^{j-1}$$

$$\leq (1 - 0.1\alpha h)\Delta_{n-1}^j + \left(4 + \frac{4}{\eta}\right)\frac{h^2}{M}\delta^2 + \left(4 + \frac{4}{\eta}\right)\frac{\beta^2 h^2}{M}\mathcal{E}_n^j + \left(4 + \frac{4}{\eta}\right)\frac{\beta^2 h^2}{M}\mathcal{E}_n^{j-1}$$

$$= \left(1 - \frac{0.01}{\kappa}\right)\Delta_{n-1}^j + 4\left(\frac{1}{M} + 10\kappa\right)h^2\delta^2 + 4\left(\frac{1}{M} + 10\kappa\right)\beta^2 h^2 \mathcal{E}_n^j$$

$$+ 4\left(\frac{1}{M} + 10\kappa\right)\beta^2 h^2 \mathcal{E}_n^{j-1}. \tag{12}$$

In the following, we further decompose $\mathcal{E}_n^j$. For any $n = 0, \ldots, N-1$, $j \in [J]$, $p = 2, \ldots, P$, and $m = 1, \ldots, M$, we can decompose $\mathbb{E}\left[\left\|\boldsymbol{x}_{n,m}^{j,p} - \boldsymbol{x}_{n,m}^{j,p-1}\right\|^2\right]$ as follows. By definition (Line 12 or 18 in Algorithm 1), we have

$$\mathbb{E}\left[\left\|\boldsymbol{x}_{n,m}^{j,p} - \boldsymbol{x}_{n,m}^{j,p-1}\right\|^2\right]$$

$$= \frac{h^2}{M^2}\mathbb{E}\left[\left\|\sum_{m'=0}^{m-1}\boldsymbol{s}(\boldsymbol{x}_{n,m'}^{j,p-1}) - \sum_{m'=0}^{m-1}\boldsymbol{s}(\boldsymbol{x}_{n,m'}^{j,p-2})\right\|^2\right]$$

$$\leq \frac{h^2 m}{M^2}\sum_{m'=0}^{m-1}\mathbb{E}\left[\left\|\boldsymbol{s}(\boldsymbol{x}_{n,m'}^{j,p-1}) - \boldsymbol{s}(\boldsymbol{x}_{n,m'}^{j,p-2})\right\|^2\right]$$

$$\leq \frac{h^2 m}{M^2}\sum_{m'=0}^{m-1}3\left[\mathbb{E}\left[\left\|\nabla f(\boldsymbol{x}_{n,m'}^{j,p-1}) - \nabla f(\boldsymbol{x}_{n,m'}^{j,p-2})\right\|^2\right] + \mathbb{E}\left[\left\|\nabla f(\boldsymbol{x}_{n,m'}^{j,p-1}) - \boldsymbol{s}(\boldsymbol{x}_{n,m'}^{j,p-1})\right\|^2\right]\right.$$

$$+ \mathbb{E}\left[\left\|\nabla f(\boldsymbol{x}_{n,m'}^{j,p-2}) - \boldsymbol{s}(\boldsymbol{x}_{n,m'}^{j,p-2})\right\|^2\right]\right]$$

$$\leq 3\beta^2 h^2 \max_{m'=1,\ldots,M}\mathbb{E}\left[\left\|\boldsymbol{x}_{n,m'}^{j,p-1} - \boldsymbol{x}_{n,m'}^{j,p-2}\right\|^2\right] + 6\delta^2 h^2. \tag{13}$$

Furthermore,

$$\mathbb{E}\left[\left\|\boldsymbol{x}_{n,m-1}^{j,1} - \boldsymbol{x}_{n,m-1}^{j,0}\right\|^2\right]$$

$$= \mathbb{E}\left[\left\|\boldsymbol{x}_{n-1,M}^j - \frac{h}{M}\sum_{m'=0}^{m-1}\boldsymbol{s}(\boldsymbol{x}_{n,m'}^{j,0}) - \left(\boldsymbol{x}_{n-1,M}^{j-1} - \frac{h}{M}\sum_{m'=0}^{m-1}\boldsymbol{s}(\boldsymbol{x}_{n,m'}^{j-1,P-1})\right)\right\|^2\right]$$

$$\leq 2\mathbb{E}\left[\left\|\boldsymbol{x}_{n-1,M}^j - \boldsymbol{x}_{n-1,M}^{j-1}\right\|^2\right] + 2\frac{h^2 m}{M^2}\sum_{m'=0}^{m-1}\mathbb{E}\left[\left\|\boldsymbol{s}(\boldsymbol{x}_{n,m'}^{j-1,P}) - \boldsymbol{s}(\boldsymbol{x}_{n,m'}^{j-1,P-1})\right\|^2\right]$$

$$\leq 2\Delta_{n-1}^j + 6\beta^2 h^2 \mathcal{E}_n^{j-1} + 12\delta^2 h^2. \tag{14}$$

Combining Eq. (13) and Eq. (14), we have

$$\mathcal{E}_n^j = \mathbb{E}\left[\left\|\boldsymbol{x}_{n,m-1}^{j,P} - \boldsymbol{x}_{n,m-1}^{j,P-1}\right\|^2\right] \le 2 \cdot 0.03^{P-1}\Delta_{n-1}^j + 6 \cdot 0.03^P \mathcal{E}_n^{j-1} + 6.6\delta^2 h^2. \qquad (15)$$

Substitute it into Eq. (12), we have for any $j = 2, \ldots, J$, $n = 1, \ldots, N-1$,

$$\Delta_n^j \le \left(1 - \frac{0.01}{\kappa} + 8\left(\frac{1}{M} + 10\kappa\right)0.03^P\right)\Delta_{n-1}^j + 4.4\left(\frac{1}{M} + 10\kappa\right)h^2\delta^2$$

$$+ 4.4\left(\frac{1}{M} + 10\kappa\right)\beta^2 h^2 \mathcal{E}_n^{j-1} \qquad (16)$$

$$\le \left(1 - \frac{0.005}{\kappa}\right)\Delta_{n-1}^j + 4.4\left(\frac{1}{M} + 10\kappa\right)h^2\delta^2 + 4.4\left(\frac{1}{M} + 10\kappa\right)\beta^2 h^2 \mathcal{E}_n^{j-1}, \qquad (17)$$

where the second inequality holds since $P \ge \frac{2\log\kappa}{3} + 4$ implies $8\left(\frac{1}{M} + 10\kappa\right)0.03^P \le \frac{0.005}{\kappa}$.

**Decomposition when $j = 1$.** When $j = 1$, similarly, we have for $p = 1, \ldots, P$,

$$\boldsymbol{x}_{n,m+1}^{1,p} = \boldsymbol{x}_{n,m}^{1,p} - \frac{h}{M}\boldsymbol{s}(\boldsymbol{x}_{n,m}^{1,p-1}) + \sqrt{2}(B_{nh+(m+1)/h} - B_{nh+mh/M}),$$

and

$$\boldsymbol{x}_{n,m+1}^0 = \boldsymbol{x}_{n,m}^0 - \frac{h}{M}\boldsymbol{s}(\boldsymbol{x}_{n-1,M}^0) + \sqrt{2}(B_{nh+(m+1)/h} - B_{nh+mh/M}).$$

Thus by the contraction of $\phi(\boldsymbol{x}) = \boldsymbol{x} - \frac{h}{M}\nabla f(\boldsymbol{x})$ (Lemma 2.2 in Altschuler & Talwar (2022)), we have

$$\mathbb{E}\left[\left\|\boldsymbol{x}_{n,m+1}^{1,P} - \boldsymbol{x}_{n,m+1}^0\right\|^2\right]$$

$$= \mathbb{E}\left[\left\|\boldsymbol{x}_{n,m}^{1,P} - \frac{h}{M}\boldsymbol{s}(\boldsymbol{x}_{n,m'}^{1,P-1}) - \left(\boldsymbol{x}_{n,m}^0 - \frac{h}{M}\boldsymbol{s}(\boldsymbol{x}_{n-1,M}^0)\right)\right\|^2\right]$$

$$\le (1+\eta)\mathbb{E}\left[\left\|\boldsymbol{x}_{n,m}^{1,P} - \frac{h}{M}\nabla f(\boldsymbol{x}_{n,m}^{1,P}) - \left(\boldsymbol{x}_{n,m}^0 - \frac{h}{M}\nabla f(\boldsymbol{x}_{n,m}^0)\right)\right\|^2\right]$$

$$+ \left(2 + \frac{2}{\eta}\right)\mathbb{E}\left[\left\|\frac{h}{M}\nabla f(\boldsymbol{x}_{n,m}^{1,P}) - \frac{h}{M}\nabla f(\boldsymbol{x}_{n,m}^{1,P-1}) + \frac{h}{M}\nabla f(\boldsymbol{x}_{n,m}^0) - \frac{h}{M}\nabla f(\boldsymbol{x}_{n-1,M}^0)\right\|^2\right]$$

$$+ \left(2 + \frac{2}{\eta}\right)\mathbb{E}\left[\left\|\frac{h}{M}\nabla f(\boldsymbol{x}_{n,m}^{1,P-1}) - \frac{h}{M}\boldsymbol{s}(\boldsymbol{x}_{n,m}^{1,P-1}) + \frac{h}{M}\nabla f(\boldsymbol{x}_{n-1,M}^0) - \frac{h}{M}\boldsymbol{s}(\boldsymbol{x}_{n-1,M}^0)\right\|^2\right]$$

$$\le (1+\eta)\left(1 - \frac{\alpha h}{M}\right)^2\mathbb{E}\left[\left\|\boldsymbol{x}_{n,m}^{1,P} - \boldsymbol{x}_{n,m}^0\right\|^2\right] + \left(4 + \frac{4}{\eta}\right)\frac{\delta^2 h^2}{M^2}$$

$$+ \left(4 + \frac{4}{\eta}\right)\frac{\beta^2 h^2}{M^2}\mathbb{E}\left[\left\|\boldsymbol{x}_{n,m}^{1,P} - \boldsymbol{x}_{n,m}^{1,P-1}\right\|^2\right] + \left(4 + \frac{4}{\eta}\right)\frac{\beta^2 h^2}{M^2}\mathbb{E}\left[\left\|\boldsymbol{x}_{n,m}^0 - \boldsymbol{x}_{n-1,M}^0\right\|^2\right].$$

For third term $\mathbb{E}\left[\left\|\boldsymbol{x}_{n,m}^{1,P} - \boldsymbol{x}_{n,m}^{1,P-1}\right\|^2\right]$, we have

$$\mathbb{E}\left[\left\|\boldsymbol{x}_{n,m}^{1,P} - \boldsymbol{x}_{n,m}^{1,P-1}\right\|^2\right]$$

$$= \mathbb{E}\left[\left\|\frac{h}{M}\sum_{m'=0}^m \boldsymbol{s}(\boldsymbol{x}_{n,m'}^{1,P-1}) - \boldsymbol{s}(\boldsymbol{x}_{n,m'}^{1,P-2})\right\|^2\right]$$

$$\le \frac{mh^2}{M^2}\sum_{m'=0}^m \mathbb{E}\left[\left\|\boldsymbol{s}(\boldsymbol{x}_{n,m'}^{1,P-1}) - \boldsymbol{s}(\boldsymbol{x}_{n,m'}^{1,P-2})\right\|^2\right]$$

$$\le 3\beta^2 h^2 \max_{m'=0,\ldots,M} \mathbb{E}\left[\left\|\boldsymbol{x}_{n,m'}^{1,P-1} - \boldsymbol{x}_{n,m'}^{1,P-2}\right\|^2\right] + 6\delta^2 h^2.$$

Thus

$$\mathbb{E}\left[\left\|\boldsymbol{x}_{n,m}^{1,P} - \boldsymbol{x}_{n,m}^{1,P-1}\right\|^2\right] \le 0.03^{P-1} \max_{m'=0,\dots,M} \mathbb{E}\left[\left\|\boldsymbol{x}_{n,m'}^{1,1} - \boldsymbol{x}_{n,m'}^{1,0}\right\|^2\right] + 6.2\delta^2 h^2. \quad (18)$$

For $\mathbb{E}\left[\left\|\boldsymbol{x}_{n,m}^{1,1} - \boldsymbol{x}_{n,m}^{1,0}\right\|^2\right]$, by definition, we have

$$\mathbb{E}\left[\left\|\boldsymbol{x}_{n,m}^{1,1} - \boldsymbol{x}_{n,m}^{1,0}\right\|^2\right]$$

$$= \mathbb{E}\left[\left\|\boldsymbol{x}_{n-1,M}^1 - \frac{h}{M}\sum_{m'=0}^{m-1} \boldsymbol{s}(\boldsymbol{x}_{n,m'}^0) - \left(\boldsymbol{x}_{n-1,M}^0 - \frac{h}{M}\sum_{m'=0}^{m-1} \boldsymbol{s}(\boldsymbol{x}_{n-1,M}^0)\right)\right\|^2\right]$$

$$\le 2\mathbb{E}\left[\left\|\boldsymbol{x}_{n-1,M}^1 - \boldsymbol{x}_{n-1,M}^0\right\|^2\right] + 2\mathbb{E}\left[\left\|\frac{h}{M}\sum_{m'=0}^{m-1} \boldsymbol{s}(\boldsymbol{x}_{n,m'}^0) - \frac{h}{M}\sum_{m'=0}^{m-1} \boldsymbol{s}(\boldsymbol{x}_{n-1,M}^0)\right\|^2\right]$$

$$\le 2\mathbb{E}\left[\left\|\boldsymbol{x}_{n-1,M}^1 - \boldsymbol{x}_{n-1,M}^0\right\|^2\right] + 2\frac{h^2 m}{M^2}\sum_{m'=0}^{m-1} \mathbb{E}\left[\left\|\boldsymbol{s}(\boldsymbol{x}_{n,m'}^0) - \boldsymbol{s}(\boldsymbol{x}_{n-1,M}^0)\right\|^2\right]$$

$$\le 2\mathbb{E}\left[\left\|\boldsymbol{x}_{n-1,M}^1 - \boldsymbol{x}_{n-1,M}^0\right\|^2\right] + 6\beta^2 h^2 \max_{m'\in[M]} \mathbb{E}\left[\left\|\boldsymbol{x}_{n,m'}^0 - \boldsymbol{x}_{n-1,M}^0\right\|^2\right] + 12\delta^2 h^2. \quad (19)$$

For $\mathbb{E}\left[\left\|\boldsymbol{x}_{n,m}^0 - \boldsymbol{x}_{n-1,M}^0\right\|^2\right]$, by definition of $\boldsymbol{x}_{n,m}^0$ (Line 7 in Algorithm 1), we have

$$\mathbb{E}\left[\left\|\boldsymbol{x}_{n,m}^0 - \boldsymbol{x}_{n-1,M}^0\right\|^2\right]$$

$$= \frac{h^2 m^2}{M^2}\mathbb{E}\left[\left\|\boldsymbol{s}(\boldsymbol{x}_{n-1,M}^0)\right\|^2\right] + \frac{dhm}{M}$$

$$\le 2\delta^2 h^2 + 2h^2 \mathbb{E}\left[\left\|\nabla f(\boldsymbol{x}_{n-1,M}^0)\right\|^2\right] + dh$$

$$\le 2\delta^2 h^2 + 2h^2\left(2\beta d + \frac{4\beta^2}{\alpha}\mathsf{KL}(\mu_{n-1,M}^0\|\pi)\right) + dh$$

$$= 4h^2\beta d + 2h^2\delta^2 + \frac{8\beta^2 h^2}{\alpha}\mathsf{KL}_{n-1}^0 + dh, \quad (20)$$

where the last inequality is implied from the following lemma, (Vempala & Wibisono, 2019, Lemma 10)

$$\mathbb{E}\left[\left\|\nabla f(\boldsymbol{x}_{n-1,M}^0)\right\|^2\right] \le 2\beta d + \frac{4\beta^2}{\alpha}\mathsf{KL}(\mu_{n-1,M}^0\|\pi).$$

Combining Eq. (18), Eq. (19), and Eq. (20), and $P \ge 4$, we have

$$\mathbb{E}\left[\left\|\boldsymbol{x}_{n,m}^{1,P} - \boldsymbol{x}_{n,m}^{1,P-1}\right\|^2\right]$$

$$\le 0.03^{P-1} \max_{m'=0,\dots,M} \mathbb{E}\left[\left\|\boldsymbol{x}_{n,m'}^{1,1} - \boldsymbol{x}_{n,m'}^{1,0}\right\|^2\right] + 6.2h^2\delta^2$$

$$\le 0.03^{P-1}\left[2\Delta_{n-1}^1 + 6\beta^2 h^2\left(4h^2\beta d + 2h^2\delta^2 + \frac{8\beta^2 h^2}{\alpha}\mathsf{KL}_{n-1}^0 + dh\right) + 12\delta^2 h^2\right] + 6.2h^2\delta^2$$

$$\le 2\cdot 0.03^{P-1}\Delta_{n-1}^1 + 6.3h^2\delta^2 + 0.01dh + 0.01\frac{\beta^2 h^2}{\alpha}\mathsf{KL}_{n-1}^0. \quad (21)$$

By setting $\eta = \frac{\alpha h}{M} = \frac{1}{10\kappa M}$, we have

$$\mathbb{E}\left[\left\|\boldsymbol{x}_{n,M}^{1,P} - \boldsymbol{x}_{n,M}^0\right\|^2\right]$$

$$\leq \left(1 - \frac{\alpha h}{M}\right)^M \mathbb{E}\left[\left\|\boldsymbol{x}_{n,0}^{1,P} - \boldsymbol{x}_{n,0}^0\right\|^2\right] + \left(4 + \frac{4}{\eta}\right)\frac{\delta^2 h^2}{M}$$

$$+ \left(4 + \frac{4}{\eta}\right)\frac{\beta^2 h^2}{M}\left(2 \cdot 0.03^{P-1}\Delta_{n-1}^1 + 6.3h^2\delta^2 + 0.01dh + 0.01\frac{\beta^2 h^2}{\alpha}\mathsf{KL}_{n-1}^0\right)$$

$$+ \left(4 + \frac{4}{\eta}\right)\frac{\beta^2 h^2}{M}\left(4h^2\beta d + 2h^2\delta^2 + \frac{8\beta^2 h^2}{\alpha}\mathsf{KL}_{n-1}^0 + dh\right)$$

$$\leq \left(1 - \frac{0.01}{\kappa} + 4\left(\frac{1}{M} + 10\kappa\right)0.03^P\right)\Delta_{n-1}^1 + \left(\frac{1}{M} + 10\kappa\right)\left(5\delta^2 h^2 + 6\beta^2 dh^3 + 0.4\beta^2 h^2\frac{\mathsf{KL}_{n-1}^0}{\alpha}\right)$$

$$\leq \Delta_{n-1}^1 + \left(\frac{1}{M} + 10\kappa\right)\left(5\delta^2 h^2 + 6\beta^2 dh^3 + 0.4\beta^2 h^2\frac{\mathsf{KL}_{n-1}^0}{\alpha}\right)$$

where the last inequality holds since $P \geq \frac{2\log\kappa}{3} + 4$ implies $8\left(\frac{1}{M} + 10\kappa\right)0.03^P \leq \frac{0.005}{\kappa}$. $\qquad\square$

When $n = 0$, the update is identical to the Picard iteration shown in Anari et al. (2024), thus we have the following lemma.

**Lemma B.6** (Lemma 18 in Anari et al. (2024)). *For $j = 1, \ldots, J$, we have*

$$\Delta_0^j \leq 0.03^P\Delta_0^{j-1} + 6.2\delta^2 h^2,$$

*with $\Delta_0^0 := \max_{m=0,\ldots,M}\mathbb{E}\left[\left\|\boldsymbol{x}_{0,m}^0 - \boldsymbol{x}_0\right\|^2\right] \leq \frac{4\beta^2 h^2}{\alpha}\mathsf{KL}(\mu_0\|\pi) + 1.4dh + 2\delta^2 h^2.$*

**Corollary B.7.** *For $n = 1, \ldots, N - 1$, we have*

$$\Delta_n^1 \leq n\left(\frac{1}{M} + 10\kappa\right)\left(5.1\delta^2 h^2 + 0.5\frac{\beta^2 h^2}{\alpha}\mathsf{KL}(\mu_0\|\pi) + 10\kappa^2\beta^2 dh^3\right).$$

*Furthermore, for $j = 1, \ldots, J$ and $n = 0$, we have*

$$\Delta_0^j \leq 0.03^{jP}\frac{4\beta^2 h^2}{\alpha}\mathsf{KL}(\mu_0\|\pi) + 1.4 \cdot 0.03^{jP}dh + 6.7\delta^2 h^2.$$

*Proof.* By Lemma B.6, we have

$$\begin{aligned}
\Delta_0^j &\leq 0.03^P\Delta_0^{j-1} + 6.2\delta^2 h^2 \\
&\leq 0.03^{jP}\Delta_0^0 + 6.6\delta^2 h^2 \\
&\leq 0.03^{jP}\left(\frac{4\beta^2 h^2}{\alpha}\mathsf{KL}(\mu_0\|\pi) + 1.4dh + 2\delta^2 h^2\right) + 6.6\delta^2 h^2 \\
&\leq 0.03^{jP}\frac{4\beta^2 h^2}{\alpha}\mathsf{KL}(\mu_0\|\pi) + 1.4 \cdot 0.03^{jP}dh + 6.7\delta^2 h^2.
\end{aligned}$$

Combining Lemma B.1 and Lemma B.5, we have

$$\Delta_n^1 \le \Delta_0^1 + \sum_{i=1}^n \left(\frac{1}{M} + 10\kappa\right)\left(5\delta^2 h^2 + 6\beta^2 dh^3 + 0.4\beta^2 h^2 \frac{\mathsf{KL}_{i-1}^0}{\alpha}\right)$$

$$\le \Delta_0^1 + n\left(\frac{1}{M} + 10\kappa\right)\left(5\delta^2 h^2 + 6\beta^2 dh^3\right)$$

$$+ \sum_{i=1}^n \left(\frac{1}{M} + 10\kappa\right) 0.4 \frac{\beta^2 h^2}{\alpha}\left(\exp\left(-\alpha nh\right)\mathsf{KL}(\mu_0\|\pi) + \frac{8\beta^2 dh}{\alpha}\right)$$

$$\le \Delta_0^1 + n\left(\frac{1}{M} + 10\kappa\right)\left(5\delta^2 h^2 + 6\beta^2 dh^3 + 0.4\frac{\beta^2 h^2}{\alpha}\mathsf{KL}(\mu_0\|\pi) + 3.2\kappa^2\beta^2 dh^3\right)$$

$$\le n\left(\frac{1}{M} + 10\kappa\right)\left(5.1\delta^2 h^2 + 0.5\frac{\beta^2 h^2}{\alpha}\mathsf{KL}(\mu_0\|\pi) + 10\kappa^2\beta^2 dh^3\right).$$

$\qquad\square$

### B.3 ONE STEP ANALYSIS OF $\mathcal{E}_n^j$

In this section, we analyze the one step change of $\mathcal{E}_n^j$.

**Lemma B.8.** *For any* $j = 2, \ldots, J$, $n = 1, \ldots, N-1$, *we have*

$$\mathcal{E}_n^j \le 2 \cdot 0.03^{P-1}\Delta_{n-1}^j + 2 \cdot 0.03^P \mathcal{E}_n^{j-1} + 7\delta^2 h^2.$$

*Furthermore, for* $n = 1, \ldots, N-1$, *we have*

$$\mathcal{E}_n^1 \le 2 \cdot 0.03^{P-1}\Delta_{n-1}^1 + 6.3h^2\delta^2 + 0.01dh + 0.01\frac{\beta^2 h^2}{\alpha}\mathsf{KL}_{n-1}^0.$$

*Proof.* By Eq. (15), the first inequality holds. By Eq. (21), the second inequality holds. $\qquad\square$

**Corollary B.9.** *For* $n = 1, \ldots, N-1$, *we have*

$$\mathcal{E}_n^1 \le n\left(5.5\delta^2 h^2 + 0.1\frac{\beta^2 h^2}{\alpha}\mathsf{KL}(\mu_0\|\pi) + 0.1\kappa^2 dh\right).$$

*Proof.* Combining Lemma B.1, Lemma B.8 and Corollary B.7, we have

$$\mathcal{E}_n^1 \le 2 \cdot 0.03^{P-1}\Delta_{n-1}^1 + 6.3h^2\delta^2 + 0.01dh + 0.01\frac{\beta^2 h^2}{\alpha}\mathsf{KL}_{n-1}^0$$

$$\le 2 \cdot 0.03^{P-1}\Delta_{n-1}^1 + 6.3h^2\delta^2 + 0.01dh$$

$$+ 0.01\frac{\beta^2 h^2}{\alpha}\left(\exp\left(-\alpha(n+1)h\right)\mathsf{KL}(\mu_0\|\pi) + \frac{8\beta^2 dh}{\alpha}\right)$$

$$\le 2 \cdot 0.03^{P-1}\Delta_{n-1}^1 + 6.3h^2\delta^2 + 0.02\kappa dh + 0.01\frac{\beta^2 h^2}{\alpha}\mathsf{KL}(\mu_0\|\pi)$$

$$\le 2 \cdot 0.03^{P-1}\left(n\left(\frac{1}{M} + 10\kappa\right)\left(5.1\delta^2 h^2 + 0.5\frac{\beta^2 h^2}{\alpha}\mathsf{KL}(\mu_0\|\pi) + 10\kappa^2\beta^2 dh^3\right)\right)$$

$$+ 6.3h^2\delta^2 + 0.02\kappa dh + 0.01\frac{\beta^2 h^2}{\alpha}\mathsf{KL}(\mu_0\|\pi)$$

$$\le n \cdot 0.06\left(5.1\delta^2 h^2 + 0.5\frac{\beta^2 h^2}{\alpha}\mathsf{KL}(\mu_0\|\pi) + 0.1\kappa^2 dh\right)$$

$$+ 6.3h^2\delta^2 + 0.02\kappa dh + 0.01\frac{\beta^2 h^2}{\alpha}\mathsf{KL}(\mu_0\|\pi)$$

$$\le n\left(5.5\delta^2 h^2 + 0.1\frac{\beta^2 h^2}{\alpha}\mathsf{KL}(\mu_0\|\pi) + 0.1\kappa^2 dh\right).$$

where the fifth inequality holds since $P \ge \frac{2\log\kappa}{3} + 4$ implies $\left(\frac{1}{M} + 10\kappa\right)0.03^{P-1} \le 0.03$. $\qquad\square$

## B.4 PROOF OF THEOREM 4.3

We define an energy function as

$$L_n^j = \Delta_{n-1}^j + \kappa \mathcal{E}_n^{j-1}.$$

We note that $2 \cdot 0.03^{P-1} L_n^j + 7\delta^2 h^2 \geq \mathcal{E}_n^j$. By Lemma B.5 and Lemma B.8, we can decompose $L_n^j$ as

$$
\begin{aligned}
L_n^j &= \Delta_{n-1}^j + \kappa \mathcal{E}_n^{j-1} \\
&\leq \left(1 - \frac{0.005}{\kappa}\right) \Delta_{n-2}^j + 4.4 \left(\frac{1}{M} + 10\kappa\right) h^2 \delta^2 + 4.4 \left(\frac{1}{M} + 10\kappa\right) \beta^2 h^2 \mathcal{E}_{n-1}^{j-1} \\
&\quad + \kappa(0.03^{P-1} \Delta_{n-1}^{j-1} + 2 \cdot 0.03^P \mathcal{E}_n^{j-2} + 7\delta^2 h^2) \\
&\leq \left(1 - \frac{0.005}{\kappa}\right) \Delta_{n-2}^j + \kappa \left(1 - \frac{0.005}{\kappa}\right) \mathcal{E}_{n-1}^{j-1} + \kappa \cdot 0.03^{P-1} \Delta_{n-1}^{j-1} + \kappa \cdot 0.03^{P-1} \cdot \kappa \mathcal{E}_n^{j-2} \\
&\quad + 56\kappa\delta^2 h^2 \\
&= \left(1 - \frac{0.005}{\kappa}\right) L_{n-1}^j + \left(\kappa \cdot 0.03^{P-1}\right) L_n^{j-1} + 56\kappa\delta^2 h^2.
\end{aligned}
\tag{22}
$$

Combining $P \geq \frac{2\log\kappa}{3} + 4$ implies $\kappa \cdot 0.03^{P-1} \leq 0.04$, we recursively bound $L_n^j$ as

$$
\begin{aligned}
L_n^j &\leq \sum_{a=2}^n 0.04^{j-2} \binom{n-a+j-2}{j-2} L_a^2 + \sum_{b=2}^j \left(\kappa \cdot 0.03^{P-1}\right)^{j-b} \left(1 - \frac{0.005}{\kappa}\right)^{n-1} \binom{n-1+j-b}{j-b} L_1^b \\
&\quad + \sum_{a=2}^j \sum_{b=2}^n \left(1 - \frac{0.001}{\kappa}\right)^{n-b} 0.04^{j-a} 65\kappa\delta^2 h^2 \\
&\leq \sum_{a=2}^n 0.04^{j-2} \binom{n-a+j-2}{j-2} L_a^2 + \sum_{b=2}^j \left(\kappa \cdot 0.03^{P-1}\right)^{j-b} \left(1 - \frac{0.005}{\kappa}\right)^{n-1} \binom{n-1+j-b}{j-b} L_1^b \\
&\quad + 68000\kappa^2\delta^2 h^2.
\end{aligned}
\tag{23}
$$

For the first term $\sum_{a=2}^n 0.04^{j-2} \binom{n-a+j-2}{j-2} L_a^2$, we first bound $L_a^2$. To do so, we first bound $\Delta_n^2$ as follows. Combining Lemma B.5 and Corollary B.9, we have

$$
\begin{aligned}
\Delta_n^2 &\leq \left(1 - \frac{0.005}{\kappa}\right) \Delta_{n-1}^2 + 4.4 \left(\frac{1}{M} + 10\kappa\right) h^2 \delta^2 + 4.4 \left(\frac{1}{M} + 10\kappa\right) \beta^2 h^2 \mathcal{E}_n^1 \\
&\leq \Delta_{n-1}^2 + 48.4\kappa h^2 \delta^2 + 48.4\kappa\beta^2 h^2 \left(n \left(5.5\delta^2 h^2 + 0.1\frac{\beta^2 h^2}{\alpha} \mathsf{KL}(\mu_0 \| \pi) + 0.1\kappa^2 dh\right)\right) \\
&\leq \Delta_{n-1}^2 + 48.4\kappa\beta^2 h^2 n \left(55.5\delta^2 h^2 + 0.1\frac{\beta^2 h^2}{\alpha} \mathsf{KL}(\mu_0 \| \pi) + 0.1\kappa^2 dh\right) \\
&\leq \Delta_0^2 + 48.4\kappa\beta^2 h^2 n^2 \left(55.5\delta^2 h^2 + 0.1\frac{\beta^2 h^2}{\alpha} \mathsf{KL}(\mu_0 \| \pi) + 0.1\kappa^2 dh\right) \\
&\leq 0.03^{2P} \frac{4\beta^2 h^2}{\alpha} \mathsf{KL}(\mu_0 \| \pi) + 1.4 \cdot 0.03^{2P} dh + 6.7\delta^2 h^2 \\
&\quad + 48.4\kappa\beta^2 h^2 n^2 \left(55.5\delta^2 h^2 + 0.1\frac{\beta^2 h^2}{\alpha} \mathsf{KL}(\mu_0 \| \pi) + 0.1\kappa^2 dh\right) \\
&\leq 48.4\kappa\beta^2 h^2 n^2 \left(67.2\delta^2 h^2 + 0.2\frac{\beta^2 h^2}{\alpha} \mathsf{KL}(\mu_0 \| \pi) + 0.2\kappa^2 dh\right).
\end{aligned}
$$

Thus

$$
\begin{aligned}
L_a^2 &= \Delta_{a-1}^2 + \kappa \mathcal{E}_a^1 \\
&\leq 0.49\kappa(a-1)^2 \left( 67.2\delta^2 h^2 + 0.2\frac{\beta^2 h^2}{\alpha}\mathsf{KL}(\mu_0\|\pi) + 0.2\kappa^2 dh \right) \\
&\quad + \kappa \left( a \left( 5.5\delta^2 h^2 + 0.1\frac{\beta^2 h^2}{\alpha}\mathsf{KL}(\mu_0\|\pi) + 0.1\kappa^2 dh \right) \right) \\
&\leq \kappa a^2 \left( 39\delta^2 h^2 + 0.2\frac{\beta^2 h^2}{\alpha}\mathsf{KL}(\mu_0\|\pi) + 0.2\kappa^2 dh \right).
\end{aligned}
$$

Thus by $\binom{m}{n} \leq \left(\frac{em}{n}\right)^n$ for $m \geq n > 0$, we have

$$
\begin{aligned}
&\sum_{a=2}^{n} 0.04^{j-2} \binom{n-a+j-2}{j-2} L_a^2 \\
&\leq \sum_{a=2}^{n} 0.04^{j-2} e^{j-2} \left( \frac{n-a+j-2}{j-2} \right)^{j-2} L_a^2 \\
&\leq \sum_{a=2}^{n} 0.04^{j-2} e^{2j-4} L_a^2 \\
&\leq \sum_{a=2}^{n} 0.3^{j-2} \kappa a^2 \left( 39\delta^2 h^2 + 0.2\frac{\beta^2 h^2}{\alpha}\mathsf{KL}(\mu_0\|\pi) + 0.2\kappa^2 dh \right) \\
&\leq 0.3^{j-2} \kappa n^3 \left( 39\delta^2 h^2 + 0.2\frac{\beta^2 h^2}{\alpha}\mathsf{KL}(\mu_0\|\pi) + 0.2\kappa^2 dh \right). \quad (24)
\end{aligned}
$$

For the second term $\sum_{b=2}^{j} \left( \kappa \cdot 0.03^{P-1} \right)^{j-b} \left( 1 - \frac{0.005}{\kappa} \right)^{n-1} \binom{n-1+j-b}{j-b} L_1^b$, we first bound $L_1^b$. Firstly, for $\mathcal{E}_1^{b-1}$, combining Corollary B.7 and Corollary B.9, we have

$$
\begin{aligned}
\mathcal{E}_1^{b-1} &\leq 2 \cdot 0.03^{P-1} \Delta_0^{b-1} + 2 \cdot 0.03^P \mathcal{E}_1^{b-2} + 7\delta^2 h^2 \\
&\leq 2 \cdot 0.03^{P-1} \left( 0.03^{(b-1)P} \frac{4\beta^2 h^2}{\alpha}\mathsf{KL}(\mu_0\|\pi) + 1.4 \cdot 0.03^{(b-1)P} dh + 6.7\delta^2 h^2 \right) \\
&\quad + 2 \cdot 0.03^P \mathcal{E}_1^{b-2} + 7\delta^2 h^2 \\
&\leq 2 \cdot 0.03^P \mathcal{E}_1^{b-2} + 0.03^b \left( 0.01\frac{4\beta^2 h^2}{\alpha}\mathsf{KL}(\mu_0\|\pi) + 0.01dh \right) + 7.1\delta^2 h^2 \\
&\leq (2 \cdot 0.03^P)^{b-2} \mathcal{E}_1^1 + \sum_{i=0}^{b-3} \left( 2 \cdot 0.03^P \right)^i \left( 0.03^{b-i} \left( 0.01\frac{4\beta^2 h^2}{\alpha}\mathsf{KL}(\mu_0\|\pi) + 0.01dh \right) + 7.1\delta^2 h^2 \right) \\
&\leq (2 \cdot 0.03^P)^{b-2} \mathcal{E}_1^1 + \sum_{i=0}^{b-3} 0.01^i 0.03^i \left( 0.03^{b-i} \left( 0.01\frac{4\beta^2 h^2}{\alpha}\mathsf{KL}(\mu_0\|\pi) + 0.01dh \right) + 7.1\delta^2 h^2 \right) \\
&\leq (2 \cdot 0.03^P)^{b-2} \left( 5.5\delta^2 h^2 + 0.1\frac{\beta^2 h^2}{\alpha}\mathsf{KL}(\mu_0\|\pi) + 0.1\kappa^2 dh \right) \\
&\quad + 0.03^b \left( 0.02\frac{4\beta^2 h^2}{\alpha}\mathsf{KL}(\mu_0\|\pi) + 0.02dh \right) + 7.2\delta^2 h^2 \\
&\leq 0.03^b \left( 0.1\frac{\beta^2 h^2}{\alpha}\mathsf{KL}(\mu_0\|\pi) + 0.1dh \right) + 7.3\delta^2 h^2.
\end{aligned}
$$

As for $\Delta_0^b$ we have

$$
\Delta_0^b \leq 0.03^{bP}\frac{4\beta^2 h^2}{\alpha}\mathsf{KL}(\mu_0\|\pi) + 1.4 \cdot 0.03^{bP} dh + 6.7\delta^2 h^2.
$$

Thus, we bound the first term as

$$
\begin{aligned}
L_1^b &= \Delta_0^b + \kappa \mathcal{E}_1^{b-1} \\
&\le 0.03^{bP} \frac{4\beta^2 h^2}{\alpha} \mathsf{KL}(\mu_0 \| \pi) + 1.4 \cdot 0.03^{bP} dh + 6.7 \delta^2 h^2 \\
&\quad + \kappa 0.03^b \left( 0.1 \frac{\beta^2 h^2}{\alpha} \mathsf{KL}(\mu_0 \| \pi) + 0.1 dh \right) + 7.3 \delta^2 h^2 \\
&\le \kappa 0.03^b \left( 0.2 \frac{\beta^2 h^2}{\alpha} \mathsf{KL}(\mu_0 \| \pi) + 0.2 dh \right) + 14 \delta^2 h^2.
\end{aligned}
$$

Thus by $\binom{m}{n} \le \left( \frac{em}{n} \right)^n$ for $m \ge n > 0$, and $\sum_{i=0}^{m} \binom{n+i}{n} x^i = \frac{1 - (m+1)\binom{m+n+1}{n} B_x(m+1, n+1)}{(1-x)^{n+1}} \le \frac{1}{(1-x)^{n+1}}$ we have

$$
\begin{aligned}
&\sum_{b=2}^{j} \left( \kappa \cdot 0.03^{P-1} \right)^{j-b} \left( 1 - \frac{0.005}{\kappa} \right)^{n-1} \binom{n-1+j-b}{j-b} L_1^b \\
&\le \sum_{b=2}^{j} 0.04^{j-b} \left( 1 - \frac{0.005}{\kappa} \right)^{n-1} \binom{n-1+j-b}{j-b} \left( \kappa 0.03^b \left( 0.2 \frac{\beta^2 h^2}{\alpha} \mathsf{KL}(\mu_0 \| \pi) + 0.2 dh \right) \right) \\
&\quad + \sum_{b=2}^{j} \left( \kappa \cdot 0.03^{P-1} \right)^{j-b} \left( 1 - \frac{0.005}{\kappa} \right)^{n-1} \binom{n-1+j-b}{j-b} 14 \delta^2 h^2 \\
&\le \sum_{b=2}^{j} 0.04^j \binom{n-1+j-b}{j-b} \kappa \left( 0.2 \frac{\beta^2 h^2}{\alpha} \mathsf{KL}(\mu_0 \| \pi) + 0.2 dh \right) \\
&\quad + \sum_{b=2}^{j} \left( \kappa \cdot 0.03^{P-1} \right)^{j-b} \left( 1 - \frac{0.005}{\kappa} \right)^{n-1} \binom{n-1+j-b}{j-b} 14 \delta^2 h^2 \\
&\le \sum_{i=0}^{j-2} 0.04^j e^i \left( 1 + \frac{n-1}{i} \right)^i \kappa \left( 0.2 \frac{\beta^2 h^2}{\alpha} \mathsf{KL}(\mu_0 \| \pi) + 0.2 dh \right) \\
&\quad + \sum_{i=0}^{j-2} \left( \kappa \cdot 0.03^{P-1} \right)^i \left( 1 - \frac{0.005}{\kappa} \right)^{n-1} \binom{n-1+i}{i} 14 \delta^2 h^2 \\
&\le 0.11^j e^{n-1} \kappa \left( 0.2 \frac{\beta^2 h^2}{\alpha} \mathsf{KL}(\mu_0 \| \pi) + 0.2 dh \right) \\
&\quad + \frac{1}{(1 - \kappa \cdot 0.03^{P-1})^n} \left( 1 - \frac{0.005}{\kappa} \right)^{n-1} (6.6 + 7.9 \kappa) \delta^2 h^2 \\
&\le 0.11^j e^{n-1} \kappa \left( 0.2 \frac{\beta^2 h^2}{\alpha} \mathsf{KL}(\mu_0 \| \pi) + 0.2 dh \right) + \frac{1}{(1 - \kappa \cdot 0.03^{P-1})} (6.6 + 7.9 \kappa) \delta^2 h^2 \\
&\le 0.11^j e^{n-1} \left( 2.2 \kappa \left( \frac{4\beta^2 h^2}{\alpha} \mathsf{KL}(\mu_0 \| \pi) + 1.6 dh + 2 \delta^2 h^2 \right) \right) + 20 \kappa \delta^2 h^2,
\end{aligned}
$$

where the second-to-last inequality is implied by $8 \left( \frac{1}{M} + 10\kappa \right) 0.03^P \le \frac{0.005}{\kappa}$.

Combing Eq. (23) and Eq. (24), we bound $L_n^j$ as

$$L_n^j \leq \sum_{a=2}^{n} 0.04^{j-2} \binom{n-a+j-2}{j-2} L_a^2 + \sum_{b=2}^{j} \left( \kappa \cdot 0.03^{P-1} \right)^{j-b} \left( 1 - \frac{0.005}{\kappa} \right)^{n-1} \binom{n-1+j-b}{j-b} L_1^b$$

$$+ 68000 \kappa^2 \delta^2 h^2$$

$$\leq 0.3^{j-2} \kappa n^3 \left( 39 \delta^2 h^2 + 0.2 \frac{\beta^2 h^2}{\alpha} \mathsf{KL}(\mu_0 \| \pi) + 0.2 \kappa^2 dh \right)$$

$$+ 0.11^j e^{n-1} \left( 2.2 \kappa \left( \frac{4 \beta^2 h^2}{\alpha} \mathsf{KL}(\mu_0 \| \pi) + 1.6 dh + 2 \delta^2 h^2 \right) \right) + 20 \kappa \delta^2 h^2 + 68000 \kappa^2 \delta^2 h^2$$

$$\leq 0.3^{j-2} e^{n-1} \kappa n^3 \left( 41 \delta^2 h^2 + 1.8 \kappa^2 dh + 0.5 \kappa h \mathsf{KL}(\mu_0 \| \pi) \right) + 68020 \kappa^2 \delta^2 h^2.$$

Since $8 \left( \frac{1}{M} + 10\kappa \right) 0.03^P \leq \frac{0.005}{\kappa}$ implies $\kappa^2 0.03^{P-1} \leq 0.003$, we have

$$\mathcal{E}_n^j$$

$$\leq 2 \cdot 0.03^{P-1} L_n^j + 7 \delta^2 h^2$$

$$\leq 2 \cdot 0.03^{P-1} \left( 0.3^{j-2} e^{n-1} \kappa n^3 \left( 41 \delta^2 h^2 + 1.8 \kappa^2 dh + 0.5 \kappa h \mathsf{KL}(\mu_0 \| \pi) \right) + 68020 \kappa^2 \delta^2 h^2 \right) + 7 \delta^2 h^2$$

$$\leq 0.3^{j-2} e^{n-1} n^3 \left( \delta^2 h^2 + h \mathsf{KL}(\mu_0 \| \pi) + \kappa dh \right) + 416 \delta^2 h^2.$$

Thus when $J - N \geq \log \left( N^3 \left( \frac{\kappa \delta^2 h + \kappa \mathsf{KL}(\mu_0 \| \pi) + \kappa^2 d}{\varepsilon^2} \right) \right)$, we have for any $n = 0, \ldots, N-1$

$$\mathcal{E}_n^J \leq \frac{\varepsilon^2}{5 \kappa \beta} + 416 \delta^2 h^2.$$

Recall

$$\mathsf{KL}_{N-1}^J \leq e^{-1.2 \alpha (N-1) h} \left( \mathsf{KL}(\mu_0 \| \pi) + 4.4 \beta^2 h \Delta_0^J \right) + 5 \kappa \beta \mathcal{E} + \frac{0.6 \beta d}{\alpha M} + \frac{28 \delta^2}{\alpha},$$

thus when $\delta^2 \leq \frac{\alpha \varepsilon^2}{29}$, $M \geq \frac{\kappa d}{\varepsilon^2}$, and $N \geq 10 \kappa \log \frac{\mathsf{KL}(\mu_0 \| \pi)}{\varepsilon^2}$, we have

$$\mathsf{KL}_{N-1}^J \leq e^{-1.2 \alpha (N-1) h} \left( \mathsf{KL}(\mu_0 \| \pi) + 4.4 \beta^2 h \Delta_0^J \right) + 5 \kappa \beta \mathcal{E} + \frac{0.6 \beta d}{\alpha M} + \frac{28 \delta^2}{\alpha}$$

$$\leq e^{-1.2 \alpha (N-1) h} \left( \mathsf{KL}(\mu_0 \| \pi) + 4.4 \beta^2 h \left( 0.03^{JP} \frac{4 \beta^2 h^2}{\alpha} \mathsf{KL}(\mu_0 \| \pi) + 1.4 \cdot 0.03^{JP} dh + 6.7 \delta^2 h^2 \right) \right)$$

$$+ 5 \kappa \beta \mathcal{E} + \frac{0.6 \beta d}{\alpha M} + \frac{28 \delta^2}{\alpha}$$

$$\leq e^{-1.2 \alpha (N-1) h} \mathsf{KL}(\mu_0 \| \pi) + \varepsilon^2 + 5 \kappa \beta \mathcal{E} + \frac{0.6 \beta d}{\alpha M} + \frac{29 \delta^2}{\alpha}$$

$$\leq 5 \varepsilon^2.$$

## C  MISSING DETAILS FOR SAMPLING FOR DIFFUSION MODELS

In this section, we first present the details of algorithm in Section C.1, then give the detailed analysis in the rest parts.

### C.1  ALGORITHM

**Stepsize scheme.** We first present the stepsize schedule for diffusion models, which is the same as the discretization scheme in Chen et al. (2024a). Specifically, we split the the time horizon $T$ into $N$ time slices with length $h_n \leq h = \frac{T}{N} = \Omega(1)$, forming a large gap grid $(t_n)_{n=0}^N$ with $t_n = \sum_{i=1}^n h_i$.

For any $n \in [0 : N-1]$, we further split the $n$-th time slice into a grid $(\tau_{n,m})_{m=0}^{M_n}$ with $\tau_{n,0} = 0$ and $\tau_{n,M_n} = h_n$. We denote the step size of the $m$-th step in the $n$-th time slice as $\epsilon_{n,m} = \tau_{n,m+1} - \tau_{n,m}$, and the total number of steps in the $n$-th time slice as $M_n$.

---

**Algorithm 2:** Parallel Picard Iteration Method for diffusion models

---

**Input :** $\widehat{\boldsymbol{y}}_0 \sim \widehat{\boldsymbol{q}}_0 = \mathcal{N}(0, I_d)$, the learned NN-based score function $\boldsymbol{s}_t^\theta(\cdot)$, the depth of Picard
iterations $J$, the depth of inner Picard iteration $P$, and a discretization scheme
$(T, (h_n)_{n=1}^N$ and $(\tau_{n,m})_{n\in[0:N-1],m\in[0:M]})$.

**1 for** $n = 0, \ldots, N - 1$ **do**

**2**    **for** $m = 0, \ldots, M$ *(in parallel)* **do**

**3**      $\boldsymbol{\xi}_{n,m} \sim \mathcal{N}(0, I_d)$

**4 for** $n = 0, \ldots, N - 1$ **do**

**5**    **for** $m = 0, \ldots, M_n$ *(in parallel)* **do**

**6**      $\widehat{\boldsymbol{y}}_{-1,M}^j = \widehat{\boldsymbol{y}}_0$, for $j = 0, \ldots, J$,

**7**
$$
\begin{aligned}
\widehat{\boldsymbol{y}}_{n,\tau_{n,m}}^0 &= e^{\frac{\tau_{n,m}}{2}} \widehat{\boldsymbol{y}}_{n-1,\tau_{n,M}}^0 \\
&+ \sum_{m'=0}^{m-1} e^{\frac{\tau_{n,m} - \tau_{n,m'+1}}{2}} \left[ 2(e^{\epsilon_{n,m'}} - 1) \boldsymbol{s}_{t_n + \tau_{n,m'}}^\theta(\widehat{\boldsymbol{y}}_{n-1,\tau_{n,M}}^0) + \sqrt{e^{\epsilon_{n,m'}} - 1} \boldsymbol{\xi}_{m'} \right],
\end{aligned} \tag{25}
$$

**8 for** $k = 1, \ldots, N$ **do**

**9**    **for** $j = 1, \ldots, \min\{k-1, J\}$ *and* $m = 0, \ldots, M_n$ *(in parallel)* **do**

**10**      let $n = k - j$, and $\widehat{\boldsymbol{y}}_{n,0}^j = \widehat{\boldsymbol{y}}_{n-1,M_n}^j$,

**11**
$$
\begin{aligned}
\widehat{\boldsymbol{y}}_{n,\tau_{n,m}}^j &= e^{\frac{\tau_{n,m}}{2}} \widehat{\boldsymbol{y}}_{n,0}^j \\
&+ \sum_{m'=0}^{m-1} e^{\frac{\tau_{n,m} - \tau_{n,m'+1}}{2}} \left[ 2(e^{\epsilon_{n,m'}} - 1) \boldsymbol{s}_{t_n + \tau_{n,m'}}^\theta(\widehat{\boldsymbol{y}}_{n,\tau_{n,m'}}^{j-1}) + \sqrt{e^{\epsilon_{n,m'}} - 1} \boldsymbol{\xi}_{m'} \right],
\end{aligned} \tag{26}
$$

**12 for** $k = N + 1, \ldots, N + J - 1$ **do**

**13**    **for** $n = \max\{0, k - J\}, \ldots, N - 1$ *and* $m = 0, \ldots, M_n$ *(in parallel)* **do**

**14**      let $j = k - n$, and $\widehat{\boldsymbol{y}}_{n,0}^j = \widehat{\boldsymbol{y}}_{n-1,M_n}^j$,

**15**
$$
\begin{aligned}
\widehat{\boldsymbol{y}}_{n,\tau_{n,m}}^j &= e^{\frac{\tau_{n,m}}{2}} \widehat{\boldsymbol{y}}_{n,0}^j \\
&+ \sum_{m'=0}^{m-1} e^{\frac{\tau_{n,m} - \tau_{n,m'+1}}{2}} \left[ 2(e^{\epsilon_{n,m'}} - 1) \boldsymbol{s}_{t_n + \tau_{n,m'}}^\theta(\widehat{\boldsymbol{y}}_{n,\tau_{n,m'}}^{j-1}) + \sqrt{e^{\epsilon_{n,m'}} - 1} \boldsymbol{\xi}_{m'} \right],
\end{aligned} \tag{27}
$$

**16 return** $\widehat{\boldsymbol{y}}_{N-1,M_{N-1}}^J$.

---

For the first $N - 1$ time slice, we simply use the uniform discretization, i.e., $h_n = h$, $\epsilon_{n,m} = \epsilon$, and $M_n = M = \frac{h}{\epsilon}$ for $n = 0, \ldots, N - 2$ and $m = 0, \ldots, M - 1$. For the last time slice, we also apply early stopping at time $t_N = T - \eta$, where $\eta$ is chosen in a way such that the $\mathcal{O}(\sqrt{\eta})$ 2-Wasserstein distance between $\breve{p}_N$ and its smoothed version $\breve{p}_{T-\eta}$ that we aim to sample from alternatively, is tolerable for the downstream tasks. An exponential decay of the step size towards the data end in the last time slice is also employed. Specifically, we let $h_{N-1} = h - \delta$, and discretize the interval $[t_{N-1}, t_N] = [(N-1)h, T - \eta]$ into a grid $(t_{N-1}, m)_{m=0}^{M_{N-1}}$ with step sizes $(\epsilon_{N-1,m})_{m=0}^{M_{N-1}}$ satisfying

$$
\epsilon_{N-1,m} \leq \epsilon \wedge \epsilon \left( h - \tau_{N-1,m+1} \right).
$$

For the simplicity of notations, we introduce the following indexing function: for $\tau \in [t_n, t_{n+1}]$, we define $I_n(\tau) \in \mathbb{N}$ such that $\sum_{j=1}^{I_n(\tau)} \epsilon_{n,j} \leq \tau < \sum_{j=1}^{I_n(\tau)+1} \epsilon_{n,j}$. We define a piecewise function $g$ such that

$$
g_n(\tau) = \sum_{j=1}^{I_n(\tau)} \epsilon_{n,j} \text{ and thus we have } I_n(\tau) = \lfloor \tau/\epsilon \rfloor \text{ and } g_n(\tau) = \lfloor \tau/\epsilon \rfloor \epsilon.
$$

**Exponential integrator for Picard iterations.** Compared with Line 12 and Line 18, where we use a forward Euler-Maruyama scheme for Picard iterations, we use the the following exponential integrator scheme (Zhang & Chen, 2022; Chen et al., 2024a). Specifically, In $n$-th time slice

$[t_n, t_n + \tau_{n, M_n}]$, for each grid $t_n + \tau_{n,m}$, we simulate the approximated backward process (Eq. (3)) with Picard iterations as

$$\widehat{\boldsymbol{y}}_{n,\tau_{n,m}}^{j+1} = e^{\frac{\tau_{n,m}}{2}} \widehat{\boldsymbol{y}}_{n-1,\tau_{n,M}}^{j+1}$$

$$+ \sum_{m'=0}^{m-1} e^{\frac{\tau_{n,m} - \tau_{n,m'+1}}{2}} \left[ 2(e^{\epsilon_{n,m'}} - 1) \boldsymbol{s}_{t_n + \tau_{n,m'}}^{\theta} (\widehat{\boldsymbol{y}}_{n-1,\tau_{n,M}}^{j}) + \sqrt{e^{\epsilon_{n,m'}} - 1} \boldsymbol{\xi}_{m'} \right].$$

We note such update also inherently allows for parallelization for $m = 1, \ldots, M_n$.

## C.2 INTERPOLATION PROCESSES

Following the proof framework in Chen et al. (2024a), we consider the following processes. We first reiterate the *backward process*

$$\mathrm{d}\bar{\boldsymbol{x}}_t = \left[ \frac{1}{2} \bar{\boldsymbol{x}}_t + \nabla \log \bar{p}_t(\bar{\boldsymbol{x}}_t) \mathrm{d}_t \right] + \mathrm{d}\boldsymbol{w}_t, \quad \text{with} \quad \bar{\boldsymbol{x}}_0 \sim p_T, \tag{28}$$

and its *approximate version* with the learned score function

$$\mathrm{d}\boldsymbol{y}_t = \left[ \frac{1}{2} \boldsymbol{y}_t + \boldsymbol{s}_t^{\theta}(\boldsymbol{y}_t) \right] \mathrm{d}t + \mathrm{d}\boldsymbol{w}_t, \quad \text{with} \quad \boldsymbol{y}_0 \sim \mathcal{N}(0, I_d).$$

The filtration $\mathcal{F}_t$ refers to the filtration of the backward SDE equation 28 up to time $t$. For any fixed $n = 0, \ldots, N-1$, $j = 1, \ldots, J$, we define the *auxiliary process* $(\widehat{\boldsymbol{y}}_{t_n,\tau}^j)_{\tau \in [0,h]}$ for $\tau \in [0,h]$ conditioned on the filtration $\mathcal{F}_{t_n}$ at time $t_n$ as the solution to the following SDE for $n \neq 0$,

$$\mathrm{d}\widehat{\boldsymbol{y}}_{t_n,\tau}^j(\omega) = \left[ \frac{1}{2} \widehat{\boldsymbol{y}}_{t_n,\tau}^j(\omega) + \boldsymbol{s}_{t_n+g_n(\tau)}^{\theta} \left( \widehat{\boldsymbol{y}}_{t_n,g_n(\tau)}^{j-1}(\omega) \right) \right] \mathrm{d}\tau + \mathrm{d}\boldsymbol{w}_{t_n+\tau}(\omega) \tag{29}$$

with $\widehat{\boldsymbol{y}}_{t_n,0}^j(\omega) = \widehat{\boldsymbol{y}}_{t_{n-1},\tau_{n-1,M_{n-1}}}^j(\omega)$. The initialization process is defined as

$$\mathrm{d}\widehat{\boldsymbol{y}}_{t_n,\tau}^0(\omega) = \left[ \frac{1}{2} \widehat{\boldsymbol{y}}_{t_n,\tau}^0(\omega) + \boldsymbol{s}_{t_n+g_n(\tau)}^{\theta} \left( \widehat{\boldsymbol{y}}_{t_{n-1},\tau_{n-1,M}}^0(\omega) \right) \right] \mathrm{d}\tau + \mathrm{d}\boldsymbol{w}_{t_n+\tau}(\omega), \tag{30}$$

with $\widehat{\boldsymbol{y}}_{t_0,0}^0 = \widehat{\boldsymbol{y}}_0$ and $\widehat{\boldsymbol{y}}_{t_n,0}^0 = \widehat{\boldsymbol{y}}_{t_{n-1},\tau_{n-1,M}}$.

**Remark C.1.** *The main difference compared to the auxiliary process defined in Chen et al. (2024a) is the change of the start point across each update.*

The iteration should be perceived as a deterministic procedure to each event $\omega \in \Omega$, i.e. each realization of the Wiener process $(\boldsymbol{w}_t)_{t \geq 0}$. The following lemma clarifies this fact and proves the well-definedness and parallelability of the iteration.

**Lemma C.2.** *The auxiliary process $(\widehat{\boldsymbol{y}}_{t_n,\tau}^j(\omega))_{\tau \in [0,h_n]}$ is $\mathcal{F}_{t_n+\tau}$-adapted for any $j = 1, \ldots, j$ and $n = 0, \ldots, n-1$.*

*Proof.* Since the initialization $\widehat{\boldsymbol{y}}_{t_n,\tau}^0(\omega)$ satisfies

$$\mathrm{d}\widehat{\boldsymbol{y}}_{t_n,\tau}^0(\omega) = \left[ \frac{1}{2} \widehat{\boldsymbol{y}}_{t_n,\tau}^0(\omega) + \boldsymbol{s}_{t_n+g_n(\tau)}^{\theta} \left( \widehat{\boldsymbol{y}}_{t_{n-1},\tau_{n-1,M}}^0(\omega) \right) \right] \mathrm{d}\tau + \mathrm{d}\boldsymbol{w}_{t_n+\tau}(\omega),$$

$\widehat{\boldsymbol{y}}_{t_n,\tau}^0(\omega)$ is obliviously $\mathcal{F}_{t_n+\tau}$-adapted. Now suppose that $\boldsymbol{y}_{t_n,\tau}$ is $\mathcal{F}_{t_n+\tau}$-adapted, since $g_n(\tau) \leq \tau$, we have the following Itô integral well-defined and $\mathcal{F}_{t_n+\tau}$-adapted:

$$\int_0^\tau \boldsymbol{s}_{t_n+g_n(\tau')}^{\theta}(\boldsymbol{y}_{t_n,g_n(\tau')}) d\tau',$$

and therefore SDE

$$\mathrm{d}\boldsymbol{y}_{t_n,\tau}'(\omega) = \left[ \frac{1}{2} \boldsymbol{y}_{t_n,\tau}'(\omega) + \boldsymbol{s}_{t_n+g_n(\tau)}^{\theta} \left( \boldsymbol{y}_{t_n,g_n(\tau)}(\omega) \right) \right] \mathrm{d}\tau + \mathrm{d}\boldsymbol{w}_{t_n+\tau}(\omega)$$

has a unique strong solution $(\boldsymbol{y}_{t_n,\tau}'(\omega))_{\tau \in [0,h_n]}$ that is also $\mathcal{F}_{t_n+\tau}$-adapted. The lemma follows by induction. $\qquad \square$

Finally, the following lemma shows the equivalence of our update rule and the auxiliary process, i.e., the auxiliary process is an interpotation of the discrete points.

**Lemma C.3.** *For any $n = 0, \ldots, N-1$, the update rule (Eq. (25)) in Algorithm 2 and the update rule (Eq. (26) or Eq. (27)) are equivalent to the exact solution of the auxiliary process Eq. (30), and Eq. (29) respectively, for any $j = 1, \ldots, J$, and $\tau \in [0, h_n]$.*

*Proof.* Due to the similarity, we only prove the equivalence of the update rule (Eq. (25)). The dependency on $\omega$ will be omitted in the proof below.

For SDE equation 29, by multiplying $e^{-\frac{\tau}{2}}$ on both sides then integrating on both side from 0 to $\tau$, we have

$$
e^{-\frac{\tau}{2}} \widehat{\boldsymbol{y}}_{t_n,\tau}^j - \widehat{\boldsymbol{y}}_{t_n,0}^j = \sum_{m=0}^{M_n} 2 \left( e^{-\frac{\tau \wedge \tau_{n,m}}{2}} - e^{-\frac{\tau \wedge \tau_{n,m+1}}{2}} \right) \boldsymbol{s}_{t_n+\tau_{n,m}}^\theta \left( \widehat{\boldsymbol{y}}_{t_n,\tau_{n,m}}^{j-1} \right) + \int_0^\tau e^{-\frac{\tau'}{2}} \mathrm{d}\boldsymbol{w}_{t_n+\tau'}.
$$

Thus then multiplying $e^{\frac{\tau}{2}}$ on both sides above yields

$$
\widehat{\boldsymbol{y}}_{t_n,\tau}^j = e^{\frac{\tau}{2}} \widehat{\boldsymbol{y}}_{t_n,0}^j + \sum_{m=0}^{M_n} 2 \left( e^{-\frac{\tau \wedge \tau_{n,m} - \tau \wedge \tau_{n,m+1}}{2}} - 1 \right) e^{\frac{0 \vee (\tau - \tau_{n,m+1})}{2}} \boldsymbol{s}_{t_n+\tau_{n,m}}^\theta \left( \widehat{\boldsymbol{y}}_{t_n,\tau_{n,m}}^{j-1} \right)
$$

$$
+ \sum_{m=0}^{M_n} \int_{\tau \wedge \tau_{n,m}}^{\tau \wedge \tau_{n,m+1}} e^{\frac{\tau - \tau'}{2}} \mathrm{d}\boldsymbol{w}_{t_n+\tau'},
$$

where by Itô isometry and let $\tau = \tau_{n,m}$ we get the desired result. $\qquad\square$

### C.2.1 DECOMPOSITION OF KL DIVERGENCE

We invoke Girsanov's theorem (Theorem A.3) as follows, and the applicability of Girsanov's theorem here relies on the $\mathcal{F}_\tau$-adaptivity established by Lemma C.2.

1. We set equation 5 as the auxiliary process Eq. (29) with $j = J$, where $\boldsymbol{w}_t(\omega)$ is a Wiener process under the measure $q|_{\mathcal{F}_{t_n}}$.

2. Defining another process $\widetilde{\boldsymbol{w}}_{t_n+\tau}(\omega)$ governed by the following SDE

$$
\mathrm{d}\widetilde{\boldsymbol{w}}_{t_n+\tau}(\omega) = \mathrm{d}\boldsymbol{w}_{t_n+\tau}(\omega) + \boldsymbol{\delta}(t_n)(\tau,\omega)\mathrm{d}\tau,
$$

where

$$
\boldsymbol{\delta}_{t_n}(\tau,\omega) = \boldsymbol{s}_{t_n+g_n(\tau)}^\theta(\widehat{\boldsymbol{y}}_{t_n,g_n(\tau)}^{J-1}(\omega)) - \nabla \log \breve{p}_{t_n+\tau}(\widehat{\boldsymbol{y}}_{t_n,\tau}^J(\omega)).
$$

3. Concluding that the auxiliary processes (Eq. (29)) with $j = J$ under the measure $q|_{\mathcal{F}_{t_n}}$ satisfies the following SDE

$$
\mathrm{d}\widehat{\boldsymbol{y}}_{t_n,\tau}^J(\omega) = \left[ \frac{1}{2}\widehat{\boldsymbol{y}}_{t_n,\tau}^J(\omega) + \nabla \log \breve{p}_{t_n+\tau}(\widehat{\boldsymbol{y}}_{t_n,\tau}^J(\omega)) \right] \mathrm{d}\tau + \mathrm{d}\widetilde{\boldsymbol{w}}_{t_n+\tau}(\omega),
$$

with $(\widetilde{\boldsymbol{w}}_{t_n+\tau}(\omega))_{\tau \geq 0}$ being a Wiener process under the measure $\breve{p}|_{\mathcal{F}_{t_n}}$. Note this is identical to the original backward SDE equation 28 by variable replacement.

Now we conclude the following lemma by Corollary A.4.

**Lemma C.4.** *Assume $\boldsymbol{\delta}_{t_n}(\tau,\omega) = \boldsymbol{s}_{t_n+g_n(\tau)}^\theta(\widehat{\boldsymbol{y}}_{t_n,g_n(\tau)}^{J-1}(\omega)) - \nabla \log \breve{p}_{t_n+\tau}(\widehat{\boldsymbol{y}}_{t_n,\tau}^J(\omega))$. Then we have the following one-step decomposition,*

$$
\mathsf{KL}(\breve{p}_{t_{n+1}} \| \widehat{q}_{t_{n+1}}) \leq \mathsf{KL}(\breve{p}_{t_n} \| \widehat{q}_{t_n}) + \mathbb{E}_{\omega \sim q|_{\mathcal{F}_{t_n}}} \left[ \frac{1}{2} \int_0^{h_n} \|\boldsymbol{\delta}_{t_n}(\tau,\omega)\|^2 \mathrm{d}\tau \right].
$$

Now, the problem remaining is to bound the discrepancy quantified by

$$
\int_0^{h_n} \|\boldsymbol{\delta}_{t_n}(\tau,\omega)\|^2 \mathrm{d}\tau
$$

$$
= \int_0^{h_n} \left\| \boldsymbol{s}_{t_n+g_n(\tau)}^{\theta}(\widehat{\boldsymbol{y}}_{t_n,g_n(\tau)}^{J-1}(\omega)) - \nabla \log \breve{p}_{t_n+\tau}(\widehat{\boldsymbol{y}}_{t_n,\tau}^{J}(\omega)) \right\|^2 \mathrm{d}\tau
$$

$$
\leq 3 \left( \underbrace{\int_0^{h_n} \left\| \nabla \log \breve{p}_{t_n+g_n(\tau)}(\widehat{\boldsymbol{y}}_{t_n,g_n(\tau)}^{J}(\omega)) - \nabla \log \breve{p}_{t_n+\tau}(\widehat{\boldsymbol{y}}_{t_n,\tau}^{J}(\omega)) \right\|^2 \mathrm{d}\tau}_{:=A_{t_n}(\omega)} \right.
$$

$$
+ \underbrace{\int_0^{h_n} \left\| \boldsymbol{s}_{t_n+g_n(\tau)}^{\theta}(\widehat{\boldsymbol{y}}_{t_n,g_n(\tau)}^{J}(\omega)) - \nabla \log \breve{p}_{t_n+g_n(\tau)}(\widehat{\boldsymbol{y}}_{t_n,g_n(\tau)}^{J}(\omega)) \right\|^2 \mathrm{d}\tau}_{:=B_{t_n}(\omega)}
$$

$$
\left. + \underbrace{\int_0^{h_n} \left\| \boldsymbol{s}_{t_n+g_n(\tau)}^{\theta}(\widehat{\boldsymbol{y}}_{t_n,g_n(\tau)}^{J}(\omega)) - \boldsymbol{s}_{t_n+g_n(\tau)}^{\theta}(\widehat{\boldsymbol{y}}_{t_n,g_n(\tau)}^{J-1}(\omega)) \right\|^2 \mathrm{d}\tau}_{:=C_{t_n}(\omega)} \right), \tag{31}
$$

where $A_{t_n}(\omega)$ measures the discretization error, $B_{t_n}(\omega)$ measures the estimation error of score function, and $C_{t_n}(\omega)$ measures the error by Picard iteration.

### C.3 DISCRETIZATION ERROR AND ESTIMATION ERROR OF SCORE FUNCTION IN EVERY TIME SLICE

The following lemma from Benton et al. (2024); Chen et al. (2024a) bounds the expectation of the discretization error $A_{t_n}$.

**Lemma C.5 (Discretization error (Benton et al., 2024, Section 3.1) and (Chen et al., 2024a, Lemma B.7)).** *We have for $n \in [0 : N-2]$*

$$
\mathbb{E}_{\omega \sim \breve{p}|_{\mathcal{F}_{t_n}}} \left[ A_{t_n}(\omega) \right] \lesssim \epsilon d h_n,
$$

*and*

$$
\mathbb{E}_{\omega \sim \breve{p}|_{\mathcal{F}_{t_n}}} \left[ A_{t_{N-1}}(\omega) \right] \lesssim \epsilon d \log \eta^{-1},
$$

*where $\eta$ is the parameter for early stopping.*

The following lemma from Chen et al. (2024a) bounds the expectation of the estimation error of score function, $B_{t_n}$.

**Lemma C.6 (Estimation error of score function (Chen et al., 2024a, Section B.3)).** $\sum_{n=0}^{N-1} \mathbb{E}_{\omega \sim \breve{p}|_{\mathcal{F}_{t_n}}} [B_{t_n}] \leq \delta_2^2.$

*Proof.* By Assumption 5.1 and the the fact that the process $\widehat{\boldsymbol{y}}_{t_n,\tau}^J(\omega)$ follows the backward SDE with the true score function under the measure $\breve{p}$, we have

$$
\sum_{n=1}^{N-1} \mathbb{E}_{\omega\sim\breve{p}|\mathcal{F}_{t_n}}\left[B_{t_n}(\omega)\right]
$$

$$
\leq \mathbb{E}_{\omega\sim\breve{p}|\mathcal{F}_{t_n}}\left[\sum_{n=1}^{N-1}\int_0^{h_n}\left\|\boldsymbol{s}_{t_n+\tau}^\theta(\widehat{\boldsymbol{y}}_{t_n,\tau}^J(\omega)) - \nabla\log\breve{p}_{t_n+g_n(\tau)}(\widehat{\boldsymbol{y}}_{t_n,\tau}^J(\omega))\right\|^2\mathrm{d}\tau\right]
$$

$$
= \mathbb{E}_{\omega\sim\breve{p}|\mathcal{F}_{t_n}}\left[\sum_{n=1}^{N-1}\sum_{m=0}^{M_n}\epsilon_{n,m}\left\|\boldsymbol{s}_{t_n+\tau}^\theta(\widehat{\boldsymbol{y}}_{t_n,\tau}^J(\omega)) - \nabla\log\breve{p}_{t_n+g_n(\tau)}(\widehat{\boldsymbol{y}}_{t_n,\tau}^J(\omega))\right\|^2\mathrm{d}\tau\right]
$$

$$
= \mathbb{E}_{\omega\sim\breve{p}|\mathcal{F}_{t_n}}\left[\sum_{n=0}^{N-1}\sum_{m=0}^{M_n}\epsilon_{n,m}\left\|\boldsymbol{s}_{t_n+\tau}^\theta(\breve{\boldsymbol{x}}_{t_n+\tau}(\omega)) - \nabla\log\breve{p}_{t_n+g_n(\tau)}(\breve{\boldsymbol{x}}_{t_n+\tau}(\omega))\right\|^2\mathrm{d}\tau\right]
$$

$$
\leq \delta_2^2.
$$

$\square$

## C.4 ANALYSIS FOR INITIALIZATION

By setting the depth of iteration as $K = 1$ in Chen et al. (2024a), our initialization parts (Lines 4-7 in Algorithm 2) and the initialization process (Eq. (30)) are identical to the Algorithm 1 and the the auxiliary process (Definition B.1) in Chen et al. (2024a). We provide a brief overview of their analysis by setting $K = 1$ and reformulate it to align with our initialization. Let

$$
A_{t_n}^0(\omega) := \int_0^{h_n}\left\|\nabla\log\breve{p}_{t_n+g_n(\tau)}(\widehat{\boldsymbol{y}}_{t_n,g_n(\tau)}^0(\omega)) - \nabla\log\breve{p}_{t_n+\tau}(\widehat{\boldsymbol{y}}_{t_n,\tau}^0(\omega))\right\|^2\mathrm{d}\tau
$$

and

$$
B_{t_n}^0(\omega) := \int_0^{h_n}\left\|\boldsymbol{s}_{t_n+g_n(\tau)}^\theta(\widehat{\boldsymbol{y}}_{t_n,g_n(\tau)}^0(\omega)) - \nabla\log\breve{p}_{t_n+g_n(\tau)}(\widehat{\boldsymbol{y}}_{t_n,g_n(\tau)}^0(\omega))\right\|^2\mathrm{d}\tau
$$

**Lemma C.7** (**Lemma B.5 or Lemma B.6 with** $K = 1$ **in Chen et al. (2024a)**). *For any* $n = 0,\ldots,N-1$, *suppose the initialization* $\widehat{\boldsymbol{y}}_{t_n,0}^0$ *follows the distribution of* $\breve{x}_{t_n} \sim \breve{p}_{t_n}$, *if* $3e^{\frac{7}{2}h_n}h_nL_{\boldsymbol{s}} < 0.5$, *then the following estimate*

$$
\sup_{\tau\in[0,h_n]}\mathbb{E}_{\omega\sim\breve{p}|\mathcal{F}_{t_n}}\left[\left\|\widehat{\boldsymbol{y}}_{t_n,\tau}^0(\omega) - \widehat{\boldsymbol{y}}_{t_n,0}^0(\omega)\right\|^2\right] \leq 2h_ne^{\frac{7}{2}h_n}(M_{\boldsymbol{s}}+2d)
$$

$$
+ 6e^{\frac{7}{2}h_n}\mathbb{E}_{\omega\sim\breve{p}|\mathcal{F}_{t_n}}\left[A_{t_n}^0(\omega)+B_{t_n}^0(\omega)\right].
$$

Furthermore, the $A_{t_n}^0(\omega)$ and $B_{t_n}^0(\omega)$ can be bounded as

**Lemma C.8** ((Chen et al., 2024a, Lemma B.7)). *We have for* $n\in[0:N-2]$

$$
\mathbb{E}_{\omega\sim\breve{p}|\mathcal{F}_{t_n}}\left[A_{t_n}^0(\omega)\right] \lesssim \epsilon dh_n,
$$

*and*

$$
\mathbb{E}_{\omega\sim\breve{p}|\mathcal{F}_{t_n}}\left[A_{t_{N-1}}^0(\omega)\right] \lesssim \epsilon d\log\eta^{-1},
$$

*where* $\eta$ *is the parameter for early stopping.*

**Lemma C.9** ((Chen et al., 2024a, Section B.3)). $\sum_{n=1}^{N-1}\mathbb{E}_{\omega\sim\breve{p}|\mathcal{F}_{t_n}}\left[B_{t_n}(\omega)\right] \leq \delta_2^2$.

Thus we have the following conclusion

**Corollary C.10.** *With the same assumption in Lemma C.7, we have*

$$
\sup_{n=0,\ldots,N}\sup_{\tau\in[0,h_n]}\mathbb{E}_{\omega\sim\breve{p}|\mathcal{F}_{t_n}}\left[\left\|\widehat{\boldsymbol{y}}_{t_n,\tau}^0(\omega) - \widehat{\boldsymbol{y}}_{t_n,0}^0(\omega)\right\|^2\right] \lesssim d.
$$

## C.5 CONVERGENCE OF PICARD ITERATION

Similarly, we define

$$\mathcal{E}_n^j = \sup_{\tau \in [0, h_n]} \mathbb{E}_{\omega \sim \bar{p}|\mathcal{F}_{t_n}} \left[ \|\widehat{\boldsymbol{y}}_{t_n, \tau}^j(\omega) - \widehat{\boldsymbol{y}}_{t_n, \tau}^{j-1}(\omega)\|^2 \right],$$

and

$$\Delta_n^j = \mathbb{E}_{\omega \sim \bar{p}|\mathcal{F}_{t_n}} \left[ \|\widehat{\boldsymbol{y}}_{t_n, \tau_{n,M}}^j(\omega) - \widehat{\boldsymbol{y}}_{t_n, \tau_{n,M}}^{j-1}(\omega)\|^2 \right].$$

Furthermore, we let $\mathcal{E}_I = \sup_{n=0,\ldots,N-1} \sup_{\tau \in [0, h_n]} \mathbb{E}_{\omega \sim \bar{p}|\mathcal{F}_{t_n}} \left[ \left\| \widehat{\boldsymbol{y}}_{n,\tau}^0 - \widehat{\boldsymbol{y}}_{n-1,\tau_{n,M}}^0 \right\|^2 \right]$. We note that by Corollary C.10, $\mathcal{E}_I \lesssim d$.

**Lemma C.11** (One-step decomposition of $\mathcal{E}_n^j$). *Assume $L_s^2 e^{2h_n} h_n \leq 0.01$ and and $e^{2h_n} \leq 2$. For any $j = 2, \ldots, J$, $n = 0, \ldots, N-1$, we have*

$$\mathcal{E}_n^j \leq 2\Delta_{n-1}^j + 0.01 \mathcal{E}_n^{j-1}.$$

*Furthermore, for $j = 1$, $n = 1, \ldots, N-1$, we have*

$$\mathcal{E}_n^1 \leq 2\Delta_n^1 + 0.01 \left( \sup_{\tau \in [0, h_n]} \mathbb{E}_{\omega \sim \bar{p}|\mathcal{F}_{t_n}} \left\| \widehat{\boldsymbol{y}}_{t_n, \tau}^0(\omega) - \widehat{\boldsymbol{y}}_{t_{n-1}, \tau_{n-1,M}}^0(\omega) \right\|^2 \right).$$

*Proof.* For each $\omega \in \Omega$ conditioned on the filtration $\mathcal{F}_{t_n}$, consider the auxiliary process defined as in the previous section,

$$\mathrm{d}\widehat{\boldsymbol{y}}_{t_n,\tau}^j(\omega) = \left[ \frac{1}{2} \widehat{\boldsymbol{y}}_{t_n,\tau}^j(\omega) + \boldsymbol{s}_{t_n+g_n(\tau)}^\theta \left( \widehat{\boldsymbol{y}}_{t_n,g_n(\tau)}^{j-1}(\omega) \right) \right] \mathrm{d}\tau + \mathrm{d}\boldsymbol{w}_{t_n+\tau}(\omega),$$

and

$$\mathrm{d}\widehat{\boldsymbol{y}}_{t_n,\tau}^{j-1}(\omega) = \left[ \frac{1}{2} \widehat{\boldsymbol{y}}_{t_n,\tau}^{j-1}(\omega) + \boldsymbol{s}_{t_n+g_n(\tau)}^\theta \left( \widehat{\boldsymbol{y}}_{t_n,g_n(\tau)}^{j-2}(\omega) \right) \right] \mathrm{d}\tau + \mathrm{d}\boldsymbol{w}_{t_n+\tau}(\omega).$$

We have

$$\mathrm{d}\left( \widehat{\boldsymbol{y}}_{t_n,\tau}^j(\omega) - \widehat{\boldsymbol{y}}_{t_n,\tau}^{j-1}(\omega) \right)$$

$$= \left[ \frac{1}{2} \left( \widehat{\boldsymbol{y}}_{t_n,\tau}^j(\omega) - \widehat{\boldsymbol{y}}_{t_n,\tau}^{j-1}(\omega) \right) + \boldsymbol{s}_{t_n+g_n(\tau)}^\theta \left( \widehat{\boldsymbol{y}}_{t_n,g_n(\tau)}^{j-1}(\omega) \right) - \boldsymbol{s}_{t_n+g_n(\tau)}^\theta \left( \widehat{\boldsymbol{y}}_{t_n,g_n(\tau)}^{j-2}(\omega) \right) \right] \mathrm{d}\tau,$$

where the diffusion term $\mathrm{d}\boldsymbol{w}_{t_n+\tau}(\omega)$ cancels each other out. By above equation we can calculate the derivative $\frac{\mathrm{d}}{\mathrm{d}\tau} \left\| \widehat{\boldsymbol{y}}_{t_n,\tau}^j(\omega) - \widehat{\boldsymbol{y}}_{t_n,\tau}^{j-1}(\omega) \right\|^2$ as

$$\frac{\mathrm{d}}{\mathrm{d}\tau} \left\| \widehat{\boldsymbol{y}}_{t_n,\tau}^j(\omega) - \widehat{\boldsymbol{y}}_{t_n,\tau}^{j-1}(\omega) \right\|^2$$

$$= 2 \left( \widehat{\boldsymbol{y}}_{t_n,\tau}^j(\omega) - \widehat{\boldsymbol{y}}_{t_n,\tau}^{j-1}(\omega) \right)^\top \left[ \frac{1}{2} \left( \widehat{\boldsymbol{y}}_{t_n,\tau}^j(\omega) - \widehat{\boldsymbol{y}}_{t_n,\tau}^{j-1}(\omega) \right) + \boldsymbol{s}_{t_n+g_n(\tau)}^\theta \left( \widehat{\boldsymbol{y}}_{t_n,g_n(\tau)}^{j-1}(\omega) \right) - \boldsymbol{s}_{t_n+g_n(\tau)}^\theta \left( \widehat{\boldsymbol{y}}_{t_n,g_n(\tau)}^{j-2}(\omega) \right) \right].$$

By integrating from $0$ to $\tau$, we have

$$\left\| \widehat{\boldsymbol{y}}_{t_n,\tau}^j(\omega) - \widehat{\boldsymbol{y}}_{t_n,\tau}^{j-1}(\omega) \right\|^2 - \left\| \widehat{\boldsymbol{y}}_{t_n,0}^j(\omega) - \widehat{\boldsymbol{y}}_{t_n,0}^{j-1}(\omega) \right\|^2$$

$$= \int_0^\tau \left\| \widehat{\boldsymbol{y}}_{t_n,\tau'}^j(\omega) - \widehat{\boldsymbol{y}}_{t_n,\tau'}^{j-1}(\omega) \right\|^2 \mathrm{d}\tau'$$

$$+ \int_0^\tau 2 \left( \widehat{\boldsymbol{y}}_{t_n,\tau}^j(\omega) - \widehat{\boldsymbol{y}}_{t_n,\tau'}^{j-1}(\omega) \right)^\top \left[ \boldsymbol{s}_{t_n+g_n(\tau')}^\theta \left( \widehat{\boldsymbol{y}}_{t_n,g_n(\tau')}^{j-1}(\omega) \right) - \boldsymbol{s}_{t_n+g_n(\tau')}^\theta \left( \widehat{\boldsymbol{y}}_{t_n,g_n(\tau')}^{j-2}(\omega) \right) \right] \mathrm{d}\tau'$$

$$\leq 2 \int_0^\tau \left\| \widehat{\boldsymbol{y}}_{t_n,\tau'}^j(\omega) - \widehat{\boldsymbol{y}}_{t_n,\tau'}^{j-1}(\omega) \right\|^2 \mathrm{d}\tau' + \int_0^\tau \left\| \boldsymbol{s}_{t_n+g_n(\tau')}^\theta \left( \widehat{\boldsymbol{y}}_{t_n,g_n(\tau')}^{j-1}(\omega) \right) - \boldsymbol{s}_{t_n+g_n(\tau')}^\theta \left( \widehat{\boldsymbol{y}}_{t_n,g_n(\tau')}^{j-2}(\omega) \right) \right\|^2 \mathrm{d}\tau'$$

$$\leq 2 \int_0^\tau \left\| \widehat{\boldsymbol{y}}_{t_n,\tau'}^j(\omega) - \widehat{\boldsymbol{y}}_{t_n,\tau'}^{j-1}(\omega) \right\|^2 \mathrm{d}\tau' + L_s^2 \int_0^\tau \left\| \widehat{\boldsymbol{y}}_{t_n,g_n(\tau')}^{j-1}(\omega) - \widehat{\boldsymbol{y}}_{t_n,g_n(\tau')}^{j-2}(\omega) \right\|^2 \mathrm{d}\tau'.$$

By Theorem A.5, and $\widehat{\boldsymbol{y}}_{t_n,0}^{j,p}(\omega) = \widehat{\boldsymbol{y}}_{t_{n-1},\tau_{n-1,M}}^{j}(\omega)$, we have

$$\left\| \widehat{\boldsymbol{y}}_{t_n,\tau}^{j}(\omega) - \widehat{\boldsymbol{y}}_{t_n,\tau}^{j-1}(\omega) \right\|^2 \le L_{\boldsymbol{s}}^2 e^{2\tau} \int_0^\tau \left\| \widehat{\boldsymbol{y}}_{t_n,g_n(\tau')}^{j-1}(\omega) - \widehat{\boldsymbol{y}}_{t_n,g_n(\tau')}^{j-2}(\omega) \right\|^2 \mathrm{d}\tau' + e^{2\tau} \Delta_{n-1}^{j}.$$

By taking expectation, for all $\tau \in [0, h_n]$

$$\mathbb{E}_{\omega \sim \bar{p}|\mathcal{F}_{t_n}} \left\| \widehat{\boldsymbol{y}}_{t_n,\tau}^{j}(\omega) - \widehat{\boldsymbol{y}}_{t_n,\tau}^{j-1}(\omega) \right\|^2 - e^{2\tau} \Delta_{n-1}^{j}$$

$$\le L_{\boldsymbol{s}}^2 e^{2\tau} \int_0^\tau \mathbb{E}_{\omega \sim \bar{p}|\mathcal{F}_{t_n}} \left\| \widehat{\boldsymbol{y}}_{t_n,g_n(\tau')}^{j-1}(\omega) - \widehat{\boldsymbol{y}}_{t_n,g_n(\tau')}^{j-2}(\omega) \right\|^2 \mathrm{d}\tau'$$

$$\le L_{\boldsymbol{s}}^2 e^{2\tau} \tau \sup_{\tau' \in [0,\tau]} \mathbb{E}_{\omega \sim \bar{p}|\mathcal{F}_{t_n}} \left\| \widehat{\boldsymbol{y}}_{t_n,\tau'}^{j-1}(\omega) - \widehat{\boldsymbol{y}}_{t_n,\tau'}^{j-2}(\omega) \right\|^2.$$

Thus

$$\sup_{\tau \in [0,h_n]} \mathbb{E}_{\omega \sim \bar{p}|\mathcal{F}_{t_n}} \left\| \widehat{\boldsymbol{y}}_{t_n,\tau}^{j-1}(\omega) - \widehat{\boldsymbol{y}}_{t_n,\tau}^{j-2}(\omega) \right\|^2$$

$$\le e^{2h_n} \Delta_{n-1}^{j} + L_{\boldsymbol{s}}^2 e^{2h_n} h_n \mathcal{E}_n^{j-1}.$$

For $j = 1$, we consider the following two processes,

$$\mathrm{d}\widehat{\boldsymbol{y}}_{t_n,\tau}^{1}(\omega) = \left[ \frac{1}{2} \widehat{\boldsymbol{y}}_{t_n,\tau}^{1}(\omega) + \boldsymbol{s}_{t_n+g_n(\tau)}^{\theta} \left( \widehat{\boldsymbol{y}}_{t_n,g_n(\tau)}^{0}(\omega) \right) \right] \mathrm{d}\tau + \mathrm{d}\boldsymbol{w}_{t_n+\tau}(\omega),$$

and

$$\mathrm{d}\widehat{\boldsymbol{y}}_{t_n,\tau}^{0}(\omega) = \left[ \frac{1}{2} \widehat{\boldsymbol{y}}_{t_n,\tau}^{0}(\omega) + \boldsymbol{s}_{t_n+g_n(\tau)}^{\theta} \left( \widehat{\boldsymbol{y}}_{t_{n-1},\tau_{n-1,M}}^{0}(\omega) \right) \right] \mathrm{d}\tau + \mathrm{d}\boldsymbol{w}_{t_n+\tau}(\omega).$$

Similarly, we have

$$\sup_{\tau \in [0,h_n]} \mathbb{E}_{\omega \sim \bar{p}|\mathcal{F}_{t_n}} \left\| \widehat{\boldsymbol{y}}_{t_n,\tau}^{1}(\omega) - \widehat{\boldsymbol{y}}_{t_n,\tau}^{0}(\omega) \right\|^2$$

$$\le e^{2h_n} \Delta_n^{1} + L_{\boldsymbol{s}}^2 e^{2h_n} h_n \left( \sup_{\tau \in [0,h_n]} \mathbb{E}_{\omega \sim \bar{p}|\mathcal{F}_{t_n}} \left\| \widehat{\boldsymbol{y}}_{t_n,\tau}^{0}(\omega) - \widehat{\boldsymbol{y}}_{t_{n-1},\tau_{n-1,M}}^{0}(\omega) \right\|^2 \right).$$

$\square$

**Lemma C.12** (One-step decomposition of $\Delta_n^j$). *Assume $L_{\boldsymbol{s}}^2 e^{2h_n} h_n \le 0.01$ and and $e^{2h_n} \le 2$. For any $j = 2, \ldots, J$, $n = 1, \ldots, N-1$, we have*

$$\Delta_n^{j} \le 3\Delta_{n-1}^{j} + 0.4\mathcal{E}_n^{j-1}.$$

*Furthermore, for $j = 1$, $n = 1, \ldots, N-1$, we have*

$$\Delta_n^{1} \le 3\Delta_{n-1}^{1} + 0.4 \sup_{\tau \in [0,h_n]} \mathbb{E}_{\omega \sim \bar{p}|\mathcal{F}_{t_n}} \left[ \left\| \widehat{\boldsymbol{y}}_{n,\tau}^{0} - \widehat{\boldsymbol{y}}_{n-1,\tau_{n,M}}^{0} \right\|^2 \right].$$

*For $n = 0$, we have $\Delta_0^{j} \le 0.32\Delta_0^{j-1}$, and $\Delta_0^{1} \le \sup_{\tau \in [0,h_0]} \mathbb{E}_{\omega \sim \bar{p}|\mathcal{F}_{t_0}} \left[ \left\| \widehat{\boldsymbol{y}}_{t_0,\tau}^{0}(\omega) - \widehat{\boldsymbol{y}}_{t_0,0}^{0}(\omega) \right\|^2 \right].$*

*Proof.* By definition of $\widehat{\boldsymbol{y}}_{t_n,\tau_{n,M}}^{j}(\omega)$ we have

$$\left\| e^{-\frac{h_n}{2}} \widehat{\boldsymbol{y}}_{t_n,\tau_{n,M}}^{j} - e^{-\frac{h_n}{2}} \widehat{\boldsymbol{y}}_{t_n,\tau_{n,M}}^{j-1} \right\|^2$$

$$= \left\| \widehat{\boldsymbol{y}}_{n,0}^{j} - \widehat{\boldsymbol{y}}_{n,0}^{j-1} + \sum_{m'=0}^{m-1} e^{\frac{-\tau_{n,m'+1}}{2}} 2(e^{\epsilon_{n,m'}} - 1) \left[ \boldsymbol{s}_{t_n+\tau_{n,m'}}^{\theta}(\widehat{\boldsymbol{y}}_{n,\tau_{n,m'}}^{j-1}) - \boldsymbol{s}_{t_n+\tau_{n,m'}}^{\theta}(\widehat{\boldsymbol{y}}_{n,\tau_{n,m'}}^{j-2}) \right] \right\|^2$$

$$\le 2 \left\| \widehat{\boldsymbol{y}}_{n,0}^{j} - \widehat{\boldsymbol{y}}_{n,0}^{j-1} \right\|^2 + 2 \left\| \sum_{m'=0}^{m-1} e^{\frac{-\tau_{n,m'+1}}{2}} 2(e^{\epsilon_{n,m'}} - 1) \left[ \boldsymbol{s}_{t_n+\tau_{n,m'}}^{\theta}(\widehat{\boldsymbol{y}}_{n,\tau_{n,m'}}^{j-1}) - \boldsymbol{s}_{t_n+\tau_{n,m'}}^{\theta}(\widehat{\boldsymbol{y}}_{n,\tau_{n,m'}}^{j-2}) \right] \right\|^2$$

$$\le 2 \left\| \widehat{\boldsymbol{y}}_{n,0}^{j} - \widehat{\boldsymbol{y}}_{n,0}^{j-1} \right\|^2 + 32\epsilon_{n,m'}^2 M \sum_{m'=0}^{M-1} \left\| \left[ \boldsymbol{s}_{t_n+\tau_{n,m'}}^{\theta}(\widehat{\boldsymbol{y}}_{n,\tau_{n,m'}}^{j-1}) - \boldsymbol{s}_{t_n+\tau_{n,m'}}^{\theta}(\widehat{\boldsymbol{y}}_{n,\tau_{n,m'}}^{j-2}) \right] \right\|^2$$

$$\le 2 \left\| \widehat{\boldsymbol{y}}_{n,0}^{j} - \widehat{\boldsymbol{y}}_{n,0}^{j-1} \right\|^2 + 32h_n^2 \sup_{\tau \in [0,h_n]} L_{\boldsymbol{s}}^2 \left\| \widehat{\boldsymbol{y}}_{n,\tau}^{j-1} - \widehat{\boldsymbol{y}}_{n,\tau}^{j-2} \right\|^2,$$

where the second inequality is implied by that $e^x - 1 \leq 2x$ when $x < 1$. By taking expectation, and the assumption that $L_{\boldsymbol{s}}^2 e^{2h_n} h_n \leq 0.1$ and $e^{2h_n} \leq 2$, we have

$$
\begin{aligned}
e^{-\frac{h_n}{2}} \Delta_n^j &= \mathbb{E}_{\omega \sim \bar{p} | \mathcal{F}_{t_n}} e^{-\frac{h_n}{2}} \left[ \left\| \widehat{\boldsymbol{y}}_{t_n, \tau_{n,M}}^j - \widehat{\boldsymbol{y}}_{t_n, \tau_{n,M}}^{j-1} \right\|^2 \right] \\
&\leq 2 \mathbb{E}_{\omega \sim \bar{p} | \mathcal{F}_{t_n}} \left[ \left\| \widehat{\boldsymbol{y}}_{n,0}^j - \widehat{\boldsymbol{y}}_{n,0}^{j-1} \right\|^2 \right] + 32 h_n^2 L_{\boldsymbol{s}}^2 \sup_{\tau \in [0, h_n]} \mathbb{E}_{\omega \sim \bar{p} | \mathcal{F}_{t_n}} \left[ \left\| \widehat{\boldsymbol{y}}_{n,\tau}^{j-1} - \widehat{\boldsymbol{y}}_{n,\tau}^{j-2} \right\|^2 \right] \\
&\leq 2 \Delta_{n-1}^j + 0.32 \mathcal{E}_n^{j-1}.
\end{aligned}
$$

Thus

$$
\Delta_n^j \leq 3 \Delta_{n-1}^j + 0.4 \mathcal{E}_n^{j-1}.
$$

In the remaining part, we will bound $\Delta_n^1$. By definition, we have

$$
\begin{aligned}
&\left\| e^{-\frac{h_n}{2}} \widehat{\boldsymbol{y}}_{t_n, \tau_{n,M}}^1 (\omega) - e^{-\frac{h_n}{2}} \widehat{\boldsymbol{y}}_{t_n, \tau_{n,M}}^0 (\omega) \right\|^2 \\
&= \left\| \widehat{\boldsymbol{y}}_{n,0}^1 - \widehat{\boldsymbol{y}}_{n-1,\tau_{n,M}}^0 + \sum_{m'=0}^{m-1} e^{\frac{-\tau_{n,m'+1}}{2}} 2(e^{\epsilon_{n,m'}} - 1) \left[ \boldsymbol{s}_{t_n + \tau_{n,m'}}^\theta (\widehat{\boldsymbol{y}}_{n-1, \tau_{n,m'}}^0) - \boldsymbol{s}_{t_n + \tau_{n,m'}}^\theta (\widehat{\boldsymbol{y}}_{n-1, \tau_{n,M}}^0) \right] \right\|^2 \\
&\leq 2 \left\| \widehat{\boldsymbol{y}}_{n,0}^1 - \widehat{\boldsymbol{y}}_{n-1,\tau_{n,M}}^0 \right\|^2 + 2 \left\| \sum_{m'=0}^{M-1} e^{\frac{-\tau_{n,m'+1}}{2}} 2(e^{\epsilon_{n,m'}} - 1) \left[ \boldsymbol{s}_{t_n + \tau_{n,m'}}^\theta (\widehat{\boldsymbol{y}}_{n-1, \tau_{n,m'}}^0) - \boldsymbol{s}_{t_n + \tau_{n,m'}}^\theta (\widehat{\boldsymbol{y}}_{n-1, \tau_{n,M}}^0) \right] \right\|^2 \\
&\leq 2 \left\| \widehat{\boldsymbol{y}}_{n,0}^1 - \widehat{\boldsymbol{y}}_{n-1,\tau_{n,M}}^0 \right\|^2 + 32 h_n^2 L_{\boldsymbol{s}}^2 \sup_{\tau \in [0, h_n]} \left\| \widehat{\boldsymbol{y}}_{n-1,\tau}^0 - \widehat{\boldsymbol{y}}_{n-1,\tau_{n,M}}^0 \right\|^2,
\end{aligned}
$$

where the second inequality is implied by that $e^x - 1 \leq 2x$ when $x < 1$. Thus with $L_{\boldsymbol{s}}^2 e^{2h_n} h_n \leq 0.01$ and $e^{2h_n} \leq 2$, we have

$$
\begin{aligned}
e^{-\frac{h_n}{2}} \Delta_n^1 &= \mathbb{E}_{\omega \sim \bar{p} | \mathcal{F}_{t_n}} e^{-\frac{h_n}{2}} \left[ \left\| \widehat{\boldsymbol{y}}_{t_n, \tau_{n,M}}^1 - \widehat{\boldsymbol{y}}_{t_n, \tau_{n,M}}^0 \right\|^2 \right] \\
&\leq 2 \mathbb{E}_{\omega \sim \bar{p} | \mathcal{F}_{t_n}} \left[ \left\| \widehat{\boldsymbol{y}}_{n,0}^1 - \widehat{\boldsymbol{y}}_{n-1,\tau_{n,M}}^0 \right\|^2 \right] + 32 h_n^2 L_{\boldsymbol{s}}^2 \sup_{\tau \in [0, h_n]} \mathbb{E}_{\omega \sim \bar{p} | \mathcal{F}_{t_n}} \left[ \left\| \widehat{\boldsymbol{y}}_{n-1,\tau}^{1,P-1} - \widehat{\boldsymbol{y}}_{n-1,\tau_{n,M}}^0 \right\|^2 \right] \\
&\leq 2 \Delta_{n-1}^1 + 0.32 \sup_{\tau \in [0, h_n]} \mathbb{E}_{\omega \sim \bar{p} | \mathcal{F}_{t_n}} \left[ \left\| \widehat{\boldsymbol{y}}_{n,\tau}^0 - \widehat{\boldsymbol{y}}_{n-1,\tau_{n,M}}^0 \right\|^2 \right].
\end{aligned}
$$

$\square$

Let $L_n^j = 2\Delta_{n-1}^j + 0.01 \mathcal{E}_n^{j-1}$. We note that $L_n^j \geq \mathcal{E}_n^j$. Thus for $n \geq 1$ and $j \geq 2$,

$$
\begin{aligned}
L_n^j &= 2\Delta_{n-1}^j + 0.01 \mathcal{E}_n^{j-1} \\
&\leq 2(80\Delta_{n-1}^j + 0.4 \mathcal{E}_n^{j-1}) + 0.01 L_n^j \\
&\leq 160 L_{n-1}^j + 0.01 L_n^j. \tag{32}
\end{aligned}
$$

We recursively bound $L_n^j$ as

$$
L_n^j \leq \sum_{a=2}^n (0.01)^{j-2} 160^{n-a} \binom{n-a+j-2}{j-2} L_a^2 + \sum_{b=2}^j (0.01)^{j-b} 160^{n-1} \binom{n-1+j-b}{j-b} L_1^b.
$$

**Bound for $\sum_{a=2}^n (0.01)^{j-2} 160^{n-a} \binom{n-a+j-2}{j-2} L_a^2$.** Firstly, we bound $L_a^2$. To do so, by Lemma C.12, we bound $\Delta_n^1$ as

$$
\Delta_n^1 \leq 3\Delta_{n-1}^1 + 4\mathcal{E}_I \leq 3^n \Delta_0^1 + \sum_{i=0}^{n-1} 4 \cdot 3^i \mathcal{E}_I \leq 4 \sum_{i=0}^n 3^i \mathcal{E}_I \leq 3^{n+2} \mathcal{E}_I.
$$

and by Lemma C.11, bound $\mathcal{E}_n^1$ as

$$\mathcal{E}_n^1 \le 2\Delta_n^1 + 0.1\mathcal{E}_I \le 3^{n+3}\mathcal{E}_I.$$

Furthermore, by Lemma C.12, we bound $\Delta_n^2$ as

$$\Delta_n^2 \le 3\Delta_{n-1}^2 + 0.4\mathcal{E}_n^1 \le 3^n\Delta_0^2 + \sum_{i=0}^{n-1} 3^i \mathcal{E}_{n-i}^1 \le 0.32 \cdot 3^n \mathcal{E}_I + 3^{n+3} n\mathcal{E}_I \le 28 \cdot 3^n n\mathcal{E}_I.$$

Thus

$$L_a^2 = 2\Delta_{a-1}^2 + 0.01\mathcal{E}_a^1 \le 28 \cdot 3^a a\mathcal{E}_I.$$

Furthermore, by $\binom{m}{n} \le \left(\frac{em}{n}\right)^n$ for $m \ge n > 0$, we have

$$\sum_{a=2}^{n} (0.01)^{j-2} 160^{n-a} \binom{n-a+j-2}{j-2} L_a^2$$

$$\le (0.01)^{j-2}(28 \cdot 160^n n^2)e^{j-2} \left(\frac{n-a+j-2}{j-2}\right)^{j-2} \mathcal{E}_I$$

$$\le (e^2 \cdot 0.01)^{j-2}(28 \cdot 160^n n^2)\mathcal{E}_I.$$

**Bound for** $\sum_{b=2}^{j} (0.01)^{j-b} 160^{n-1} \binom{n-1+j-b}{j-b} L_1^b$**.** By Lemma C.11, we have

$$\mathcal{E}_1^j \le 0.01\mathcal{E}_1^{j-1} + 2\Delta_0^j$$

$$\le (0.01)^j \mathcal{E}_I + \sum_{i=0}^{j-1} (0.01)^i 2\Delta_0^{j-i}.$$

Combining the fact that $\Delta_0^j \le 0.32^{j-1}\mathcal{E}_I$, we have

$$\mathcal{E}_1^j \le 7 \cdot j \cdot 0.32^j \mathcal{E}_I.$$

Thus

$$L_1^b = 2\Delta_0^j + 0.01\mathcal{E}_1^{b-1}$$

$$\le 2 \cdot 0.32^{b-1}\mathcal{E}_I + 0.01 \cdot 7 \cdot (b-1) \cdot 0.32^{b-1}\mathcal{E}_I$$

$$\le 7 \cdot b \cdot 0.32^{b-1}\mathcal{E}_I.$$

Furthermore, by $\sum_{i=0}^{m} \binom{n+i}{n} x^i = \frac{1-(m+1)\binom{m+n+1}{n}B_x(m+1,n+1)}{(1-x)^{n+1}} \le \frac{1}{(1-x)^{n+1}}$, we have

$$\sum_{b=2}^{j} (0.01)^{j-b} 160^{n-1} \binom{n-1+j-b}{n-1} L_1^b$$

$$\le \sum_{b=2}^{j} (0.01)^{j-b} 160^{n-1} \binom{n-1+j-b}{n-1} 7 \cdot b \cdot 0.32^{b-1}\mathcal{E}_I$$

$$\le 22 \cdot 0.87^j 440^{n-1} j\mathcal{E}_I.$$

Combining the above two results, we have

$$\mathcal{E}_n^J \le (e^2 \cdot 0.01)^{j-2}(28 \cdot 160^n n^2)\mathcal{E}_I + 22 \cdot 0.87^j 440^{n-1} j\mathcal{E}_I.$$

If $J - 45N \gtrsim \log \frac{N\mathcal{E}_I}{\varepsilon^2}$, for any $n = 0, \ldots, N$

$$\mathcal{E}_n^J \le \frac{\varepsilon^2}{N}. \tag{33}$$

### C.5.1 OVERALL ERROR BOUND

By the previous computation, we have

$$\mathsf{KL}(\breve{p}_{t_{n+1}} \| \widehat{q}_{t_{n+1}})$$

$$\leq \mathsf{KL}(\breve{p}_{t_n} \| \widehat{q}_{t_n}) + \mathbb{E}_{\omega \sim q|_{\mathcal{F}_{t_n}}} \left[ \frac{1}{2} \int_0^{h_n} \| \boldsymbol{\delta}_{t_n}(\tau, \omega) \|^2 \, \mathrm{d}\tau \right]$$

$$\leq \mathsf{KL}(\breve{p}_{t_n} \| \widehat{q}_{t_n}) + 3\mathbb{E}_{\omega \sim \breve{p}|_{\mathcal{F}_{t_n}}} \left[ A_{t_n}(\omega) + B_{t_n}(\omega) \right] + 3L_{\boldsymbol{s}}^2 h_n \mathcal{E}_n^J.$$

Combining Lemma A.6, Corollary C.10, and Eq. equation 33, we have

$$\mathsf{KL}(\breve{p}_{t_{n+1}} \| \widehat{q}_{t_{n+1}})$$

$$\leq \mathsf{KL}(\breve{p}_0 \| \widehat{q}_0) + 3 \sum_{n=0}^{N-1} \left( \mathbb{E}_{\omega \sim \breve{p}|_{\mathcal{F}_{t_n}}} \left[ A_{t_n}(\omega) + B_{t_n}(\omega) \right] + L_{\boldsymbol{s}}^2 h_n \mathcal{E}_n^J \right)$$

$$\lesssim de^{-T} + \epsilon d(T + \log \eta^{-1}) + \delta_2^2 + \varepsilon^2,$$

with parameters $J - 45N \geq \mathcal{O}(\log \frac{Nd}{\varepsilon^2})$, $h = \Theta(1)$, $N = \mathcal{O}(\log \frac{d}{\varepsilon^2})$, $T = \mathcal{O}(\log \frac{d}{\varepsilon^2})$ $\epsilon = \Theta(d^{-1}\varepsilon^2 \log^{-1} \frac{d}{\varepsilon^2})$, $M = \mathcal{O}(d\varepsilon^{-2} \log \frac{d}{\varepsilon^2})$.

