# OpenReview forum: "Parallel simulation for sampling under isoperimetry and score-based diffusion models"
_ICLR.cc/2025/Conference — Submitted to ICLR 2025_

### Official Review · Reviewer_xdVr · 2024-11-02

**Soundness:** 3
**Presentation:** 4
**Contribution:** 2
**Rating:** 5
**Confidence:** 4

**Summary:**

The paper studies Picard method based discretization of Overdamped Langevin Dynamics and Diffusion Models. Picard method iterates over the entire trajectory in contrast to the Euler method which iterates point-by-point causally. Thus, Picard method is highly parallelizable and has received a lot of interest in the recent years for efficiently sampling from posterior distribution and from Diffusion models.

This work introduces the parallel picard method which increases the parallelism compared to previous method and decreases the number of Picard iterations from $\mathsf{polylog}(d)$ to $\log (d)$.

**Strengths:**

[1] The paper has a clean exposition of prior work and background material.

[2] The main technique and the idea is clearly presented and the work proposes an interesting trade-off between number of iterations and number of parallel threads. The number of Picard iterations is nearly optimal.

[3] The proof crisply analyzes the changing initial point of the Picard iteration inorder to fully parallelize it.

**Weaknesses:**

[1] There are no empirical evaluations of the proposed method. Diffusion models work with dimension $d \sim 10^4$. Providing an algorithm which requires $d/\epsilon^2 \log (d/epsilon^2)$ parallel threads with $\log(d/epsilon^2)$ iterations instead of $d/epsilon^2$ parallel threads with $\log^2(d/\epsilon^2)$ iterations seems ineffective/ vacuous since such a degree of parallelism cannot be achieved in the first place. Therefore, the relative merit of the currently proposed algorithm has to be established empirically in practical settings. It would help the case made by the paper if suitable empirical evaluations are included.

[2] Since the algorithmic modification proposed in the work is straightforward, and there are no empirical evaluations, the main technical contribution of the paper is the proof. The paper has very little exposition of proof techniques. It would be helpful to add a deeper discussion of this.

[3] The tables 1 and 2 can be improved by stating the exact polylog factors of $d$ in prior works. This is important since the main improvement claimed in the manuscript is the improvement in these factors. I found the comparison to underdamped Langevin dynamics presented together with the results for Overdamped Langevin dynamics in Table 1 very confusing. Consider splitting these comparisons or making it more clear. Similarly, comparison of SDE based methods to ODE based methods in Table 2 is also confusing.

**Questions:**

[1] Address the questions/ concerns raised in the Weaknesses section.

[2] In line 228, the equation description picard iteration, the index i appears both in the RHS as well as inside the summation. This cannot be correct. Please fix this update.

[3] In equation (1), is there a $\sqrt{2}$ missing in the diffusion term ? Without this, the stationary distribution cannot be $\mathcal{N}(0,I)$.

[4] In lines 210-212, it is stated that the reverse process contracts exponentially as per the work of Huang et al 2024. From my reading of the work, I could not find any results in the referenced work which makes this claim. Can you please elaborate?

[5] In Page 4, the score function for SGMs is assumed to be bounded. This seems like a stringent assumption. Can you elaborate and compare this with the assumptions made in prior works ?

---

> ### Author Response · Authors · 2024-11-21
>
> We thank the reviewer for their thoughtful feedback, which highlights both the strengths and areas for improvement in our work, including recognizing the interesting trade-off we propose between the number of iterations and the number of parallel threads, and the crisp analysis of our method. Below, we address the specific weaknesses and questions raised.
>
>
>
> **"...ineffective/ vacuous since such a degree of parallelism cannot be achieved in the first place"**
>
> We thank the reviewer for their suggestion regarding empirical evaluation to establish the practical merit of our algorithm. We acknowledge that, like other theoretical works on parallelization [1, 2, 3], achieving parallelization with  $d = 1000$ threads may not yet be feasible on standard hardware, such as setups with several A100 GPUs. Empirical results from [4] show that practical parallelism is typically limited to 160 threads (Figure 4 of [4]). We also offer theoretical analysis of trade-off between the number of cores used for parallel processing, computation time and memory usage in common response.
>
> However, our method is the first approach in sampling that enables parallelization across time slices, providing significant potential for acceleration. For example, when applied to the midpoint method [6] (with a total query complexity of $d^{1/3}\varepsilon^{-2/3}$), the iteration complexity becomes $\log d$, with $d^{1/3}\varepsilon^{-2/3}$ queries per iteration. This level of complexity is well-suited for standard diffusion models.
>
> To illustrate, [4] considers a latent space with dimension $d = 4 \times 96 \times 96 = 36864$. For lower-accuracy scenarios where $\varepsilon > 0.3$, our method ensures that $d^{1/3}\varepsilon^{-2/3} < 80$, which is within the 160-query limit per iteration (as shown in Figure 4 of [4]). This example demonstrates that our method is scalable and practical under standard conditions for diffusion models.
>
> [1]Faster Diffusion Sampling with Randomized Midpoints: Sequential and Parallel, Shivam Gupta, Linda Cai, Sitan Chen, 2024
>
> [2]Accelerating Diffusion Models with Parallel Sampling: Inference at Sub-Linear Time Complexity, Haoxuan Chen, Yinuo Ren, Lexing Ying, Grant M. Rotskoff, NeurIPS 2024
>
> [3]Fast parallel sampling under isoperimetry, Nima Anari, Sinho Chewi, Thuy-Duong Vuong, COLT 2024
>
> [4]Parallel Sampling of Diffusion Models, Andy Shih, Suneel Belkhale, Stefano Ermon, Dorsa Sadigh, Nima Anari, 2023
>
> [5] Improved Convergence Rate for Diffusion Probabilistic Models, Gen Li, Yuchen Jiao 2024
>
>
> **"the algorithmic modification proposed in the work is straightforward...The paper has very little exposition of proof techniques..."**
>
> We appreciate the reviewer’s comment and would like to clarify the novelty of our approach. While parallelization across time slices has been explored in the simulation community, our work is **the first** to adapt and rigorously analyze this technique within the sampling community.
>
> Regarding the overall technical novelty, we have provided a discussion in lines 311–332. As for the proof techniques:
> - For sampling under isoperimetry, we have provided a proof sketch in Section 4.3 to outline the key ideas.
> - For diffusion models, the main distinction compared to sampling under isoperimetry lies in the use of Girsanov's theorem to decompose the KL divergence, following the framework of [2]. We build on this approach to establish convergence in our setting.
>
> To address the reviewer’s concern, we will include a detailed discussion of the proof for Theorem 5.4 in the revised version, providing additional insights into the key technical contributions of our work.
>
>
> **Regarding questions**
>
> We thank the reviewer for pointing out these issues.
> - All assumptions for diffusion models align with prior work on Picard iteration for diffusion models [2], and the bounded assumption can be easily satisfied by truncation, ensuring computational stabili. We will add a discussion in the revised manuscript clarifying these assumptions.
> - We will correct the inconsistent index in line 228 and revise Equation (1) to include the missing coefficient.
> - We apologize for the inaccurate statement in lines 210-212. The correct statement should be: “The discrepancy between the terminal distributions of the backward process (Eq. (2)) and its approximation version (Eq. (3)) scales polynomially with respect to the length of the time horizon and the score matching error.” We will revise this for accuracy and clarity.

---

> > ### Comment · Reviewer_xdVr · 2024-11-24
> >
> > Thank you for your response. Is there a prior empirical work where the proposed algorithm has been evaluated against other algorithms? This is not clear from the authors' response.

---

> > > ### Author Response · Authors · 2024-11-24
> > >
> > > Thank you for your thoughtful question. To the best of our knowledge, our method is the first to parallelize sampling algorithms across time slices. Consequently, there is currently no prior empirical work that directly compares our approach with existing methods. We appreciate your interest and will investigate empirical comparisons in future studies.

---

> > > > ### Comment · Reviewer_xdVr · 2024-11-25
> > > > **Response**
> > > >
> > > > I believe that empirical evaluations of the algorithmic modifications proposed are important for this work. Thus, I choose to retain the current score of 5.

---

### Official Review · Reviewer_qt8F · 2024-11-02

**Soundness:** 3
**Presentation:** 3
**Contribution:** 3
**Rating:** 6
**Confidence:** 2

**Summary:**

The manuscript proposes a novel parallel simulation technique for sampling under isoperimetry and score-based diffusion models. It leverages a parallel Picard iteration approach that reduces iteration complexity compared to existing methods. By drawing parallels from scientific computation, particularly parallel initial-value problem solvers, the authors introduce a time-parallelized approach to improve sampling efficiency in high-dimensional settings. The manuscript provides theoretical proof for iteration and space complexity improvements and positions the technique as beneficial for tasks involving large data distributions.

**Strengths:**

**Novel Application of Parallel Picard Methods.** Adapting Picard methods for parallel sampling in high-dimensional contexts is innovative, particularly the integration of time-slice parallelization that challenges existing sequential frameworks.

**Strong Theoretical Contributions.** The manuscript rigorously addresses the theoretical guarantees of the proposed method. Its complexity bounds represent an improvement over established sampling methods, particularly with respect to the iteration complexity in the sampling task.

**Applicability to Diffusion Models.** The method has implications for score-based generative models (SGMs), which are widely used in machine learning applications. The proposed algorithm, therefore, has potential relevance in real-world applications like image generation and inverse problems.

**Weaknesses:**

**Practical Feasibility of Assumptions.** The paper relies on several strong assumptions, including accurate score function estimates and Lipschitz conditions. While these are theoretically convenient, they may limit the method's applicability in practical scenarios where these conditions are challenging to achieve.

**Complexity of the Approach.** While the theoretical aspects are well-elaborated, the algorithm’s practical implementation seems complex. There is minimal discussion of the challenges in implementing this parallel algorithm, particularly regarding memory bandwidth and processing demands.

**Lack of Empirical Validation.** The manuscript lacks experimental results. Empirical tests comparing the proposed method with existing sampling techniques would provide crucial insight into its real-world performance and validate the theoretical improvements.

**Questions:**

**Space Complexity and Scalability.** Although the paper mentions an increased space complexity, it lacks a detailed discussion of how this would scale with large data distributions in practical applications. How feasible is this method for scenarios requiring substantial memory resources?

**Empirical Benchmarks.** Could the authors provide insights on the types of empirical tests they would recommend or any preliminary results? Testing on standard datasets or benchmarks in score-based generative models would be particularly valuable for evaluating this method’s efficiency.

---

> ### Author Response · Authors · 2024-11-21
>
> Thank you for your detailed review and for recognizing the strengths of our work, particularly the novel application of parallel Picard methods, the strong theoretical contributions to iteration complexity bounds for sampling and diffusion models. We appreciate your feedbacks and address specific points below:
>
> **"...relies on several strong assumptions, including accurate score function estimates and Lipschitz conditions..."**
>
> We acknowledge that some assumptions, such as accurate score function estimates and Lipschitz conditions, may seem restrictive. However, these are standard in the theoretical analysis of score-based models and diffusion-based generative sampling methods [1-4]. Additionally, many existing methods rely on similar assumptions for their theoretical guarantees. We will add a discussion in the revised manuscript clarifying these assumptions.
>
> [1] Accelerating Diffusion Models with Parallel Sampling: Inference at Sub-Linear Time Complexity, Haoxuan Chen, Yinuo Ren, Lexing Ying, Grant M. Rotskoff, NeurIPS 2024
>
> [2] Improved analysis of score-based generative modeling: User-friendly bounds under minimal smoothness assumptions, Hongrui Chen, Holden Lee, and Jianfeng Lu, ICML 2023
>
> [3] The probability flow ode is provably fast. Sitan Chen, Sinho Chewi, Holden Lee, Yuanzhi Li, Jianfeng Lu, and Adil Salim, NeurIPS 2023
>
> [4] Sampling is as easy as learning the score: theory for diffusion models with minimal data assumptions. Sitan Chen, Sinho Chewi, Jerry Li, Yuanzhi Li, Adil Salim, and Anru R Zhang, ICLR 2023
>
>
>
> **"...large data distributions...How feasible is this method for scenarios requiring substantial memory resources?"**
>
> We thank the reviewer for raising these important points regarding space complexity and scalability.We acknowledge that, under current general-purpose computing environments, large-scale parallelism as envisioned in our method may not yet be fully feasible. In such settings, our approach offers comparable scalability to the best existing parallel methods [5]. The trade-off between total computation time and memory usage is addressed in detail in our common response.
>
> However, our method is the first approach in sampling that enables parallelization across time slices, providing significant potential for acceleration. For example, when applied to the midpoint method [7], which has a total query complexity of $d^{1/3}\varepsilon^{-2/3}$, the iteration complexity becomes $\log d$ with $d^{1/3}\varepsilon^{-2/3}$ queries per iteration. This level of complexity is feasible for standard diffusion models. To illustrate, [6] considers a latent space with dimension $d = 4 \times 96 \times 96 = 36864$. For lower-accuracy scenarios where $\varepsilon > 0.3$, our method ensures that $d^{1/3}\varepsilon^{-2/3} < 80$, while their maximum query number in each iteration is $160$ (Figure 4 in [6]). This demonstrates that our method is scalable and practical under standard conditions for diffusion models.
>
>
> [5] Accelerating Diffusion Models with Parallel Sampling: Inference at Sub-Linear Time Complexity, Haoxuan Chen, Yinuo Ren, Lexing Ying, Grant M. Rotskoff, 2024
>
> [6] Parallel Sampling of Diffusion Models, Andy Shih, Suneel Belkhale, Stefano Ermon, Dorsa Sadigh, Nima Anari, 2023
>
> [7] Improved Convergence Rate for Diffusion Probabilistic Models, Gen Li, Yuchen Jiao 2024
>
>
> **"Could the authors provide insights on the types of empirical tests they would recommend or any preliminary results?"**
>
> We appreciate the reviewer’s suggestion regarding empirical validation. While this paper focuses on advancing the theoretical understanding of our method, we agree that empirical evaluation would provide valuable insights. For future studies, we recommend using standard datasets and benchmarks for evaluating parallel acceleration in score-based generative modeling, such as those in [6] and [8].
>
>
> [8]DistriFusion: Distributed Parallel Inference for High-Resolution Diffusion Models, Muyang Li, Tianle Cai, Jiaxin Cao, Qinsheng Zhang, Han Cai, Junjie Bai, Yangqing Jia, Ming-Yu Liu, Kai Li, Song Han 2024

---

> > ### Comment · Reviewer_qt8F · 2024-11-25
> >
> > Thank you for the response. I choose to retain my score.

---

### Official Review · Reviewer_87Nq · 2024-11-03

**Soundness:** 3
**Presentation:** 3
**Contribution:** 3
**Rating:** 6
**Confidence:** 3

**Summary:**

This paper introduces a parallel Picard method aimed at enhancing the efficiency of sampling under conditions of isoperimetry and score-based diffusion models (SGMs). It addresses two primary sampling problems: sampling from log-concave distributions and sampling for SGMs used in generative modeling. The method presents improvements in iteration complexity from $O(poly(\\log d))$ to $O(\\log d)$ which aligns with known theoretical lower bounds. By leveraging parallelization techniques across both time slices and within Picard iterations, the authors propose a discretization scheme that could potentially reduce the computational burden associated with sampling, especially for large-scale datasets.

**Strengths:**

1. The paper is well-written with good motivations and explicit technical contributions.

2. The diagonal-style parallelization across time slices sounds fresh to me, which addresses limitations in convergence faced by existing methods that do not fully parallelize time slices.

3. The paper provides rigorous theoretical bounds, such as convergence rates with respect to KL divergence. The approach's complexity analysis indicates a substantial improvement over previous methods, achieving nearly optimal bounds in iteration complexity.

4. By adapting the approach for diffusion models and incorporating techniques like shrinking step sizes, the paper shows versatility and application potential across a range of generative modeling tasks.

**Weaknesses:**

While the paper includes theoretical comparisons, empirical validation on real-world datasets or benchmarks would strengthen the paper's claims regarding practical performance. Comparing accuracy, iteration complexity, and space complexity with existing SGMs on these benchmarks, as demonstrated in the experiments by Shih et al. (2024), would provide valuable insights into the practical advantages of the approach.

**Questions:**

1. Regarding the sub-optimal space complexity resulting from the application to overdamped Langevin diffusion, could the authors clarify what hinders the analysis of their methods in the context of underdamped Langevin diffusion?

2. Considering that the authors demonstrate improved iteration complexity at the expense of slightly increased space complexity, a detailed cost-related analysis would be beneficial to more thoroughly discuss the trade-offs. Specifically, evaluating computational time and memory usage under the utility maximization problem could demonstrate how these factors affect performance in practical scenarios, which might inform the method's applicability in resource-limited environments.

Miscellany: In L.3 of Algorithm 1, the subscript should be written as $B\_{nh + mh/M}$ to avoid confusion.

---

> ### Author Response · Authors · 2024-11-21
>
> Thank you for your insightful comments and appreciation of our work, particularly recognizing **the novel diagonal-style parallelization**. Below, we address your questions and minor comments:
>
>
> **"...sub-optimal space complexity...clarify what hinders the analysis...underdamped Langevin diffusion"**
>
> We apologize for the error in the table; the guarantee for underdamped Langevin diffusion should indeed be based on total variation (TV) distance rather than KL divergence, as stated in Theorem 15 of [1]. Extending the analysis to a KL divergence guarantee presents technical challenges due to the lack of the triangle inequality. For the TV guarantee, we believe that combining our framework with the analytical approach in [1] would be sufficient. Nonetheless, as our primary focus is on iteration complexity, exploring space complexity improvements while maintaining the same iteration guarantee for KL divergence via underdamped Langevin diffusion is left for future research.
>
> [1]  Fast parallel sampling under isoperimetry. Nima Anari, Sinho Chewi, and Thuy-Duong Vuong. 2024.
>
>
>
>
> **"...a detailed cost-related analysis would be beneficial to more thoroughly discuss the trade-offs. Specifically, evaluating computational time and memory usage..."**
>
> We appreciate the reviewer's suggestion and recognize the importance of cost-related analysis in understanding the trade-offs between iteration complexity, computational time, and memory usage. We offer theoretical analysis in commone response.
>
>
> We note that in the empirical experiments presented in [2], the maximum batch size per iteration is  160  for  $d \approx 10^4$. Under these conditions, the computational time of our method remains of the same order as that of the Picard method.  Our method is the first approach in sampling that enables parallelization across time slices, providing significant potential for acceleration. Specifically, if the total number of steps is reduced to  $O(d^{1/3})$ such as [3], our method can achieve maximum speed $O(\log d)$ with $O(d^{1/3})$ parallel cores. This scenario is plausible for $d\approx 10^4$. We will add this discussion into the revised version.
>
>
>
> [2] Parallel Sampling of Diffusion Models, Andy Shih, Suneel Belkhale, Stefano Ermon, Dorsa Sadigh, Nima Anari, 2023
>
> [3] Improved Convergence Rate for Diffusion Probabilistic Models, Gen Li, Yuchen Jiao 2024

---

> > ### Comment · Reviewer_87Nq · 2024-11-24
> >
> > I appreciate the authors' clarifications. They have addressed many of my concerns, and the explanation regarding computational time and memory usage seems logical. Currently, I will maintain my score, but I am open to raising it should empirical evidence supporting the authors' claims be presented.

---

### Author Response · Authors · 2024-11-21
**Common Responses**

We sincerely thank all reviewers for their detailed and constructive feedback, as well as for acknowledging the strengths of our work. We are encouraged by the positive remarks highlighting our framework as “innovative” and “fresh,” with “strong theoretical contributions” and “rigorous complexity analysis” that advance established methods and achieve “nearly optimal” results for sampling. Below, we address two key concerns raised by multiple reviewers.

----
**Regarding computational time and memory usage**
-

We appreciate the reviewers' suggestion and recognize the importance of cost-related analysis in understanding the trade-offs between iteration complexity, computational time, and memory usage. We offer theoretical analysis below and leave the empirical validation as future work.

We denote  $t_{{eval}}$  as the unit time required to evaluate the score function, and  $t_{{vec}}$  as the unit time required to perform either the addition of two vectors or the scaling of a vector, each of size  $d$. Supposing that $P$ cores are adopted in parallel algorithms. Furthermore, we assume that all coefficients required for sampling, as determined by the discretization schedule, are precomputed and readily accessible.

- For the sequential method [1], each iteration requires a single query to the score function with two scaling and two addition operations. The total number of steps is $\frac{d}{\varepsilon^2}\log^2 \frac{d}{\varepsilon^2}$ Therefore, the total computation time is $T_{seq} \approx \left(\frac{d}{\varepsilon^2} \log^2 \frac{d}{\varepsilon^2}\right) \cdot (t_{{evl}} + t_{{vec}})$. The maximum memory usage is given by: $M_{seq} = \frac{d}{\varepsilon^2}$, measured in terms of the number of words.

- For Picard method [2], each iteration requires parallel $\frac{d}{\varepsilon^2}\log\frac{d}{\varepsilon^2}$ queries to the score function and  $O(\frac{d}{\varepsilon^2}\log\frac{d}{\varepsilon^2})$ vector operations. The total number of steps is $\log^2\frac{d}{\varepsilon^2}$. Thus, the total computation time is $T_{Picard} \approx \frac{1}{P}\log^2\left(\frac{d}{\varepsilon^2}\right)\cdot \frac{d}{\varepsilon^2}\log\left(\frac{d}{\varepsilon^2}\right) \cdot (t_{{evl}} + t_{{vec}})$. The maximum memory usage is given by: $M_{Picard} = P d$ where $P\leq \frac{d}{\varepsilon^2}\log\frac{d}{\varepsilon^2}$.

- For our method, each iteration requires parallel $\frac{d}{\varepsilon^2}\log^2\frac{d}{\varepsilon^2}$ queries to the score function and  $O(\frac{d}{\varepsilon^2}\log\frac{d}{\varepsilon^2})$ vector operations. The total number of steps is $\log\frac{d}{\varepsilon^2}$. Therefore, the total computation time is $T_{our} \approx \frac{1}{P}\log\left(\frac{d}{\varepsilon^2}\right)\cdot \frac{d}{\varepsilon^2}\log^2\left(\frac{d}{\varepsilon^2}\right) \cdot (t_{{evl}} + t_{{vec}})$. The maximum memory usage is given by: $M_{our} = P d$ where $P\leq \frac{d}{\varepsilon^2}\log^2\frac{d}{\varepsilon^2}$.

We summerize the comparison maximum memory usage, total computation time and maxium speed as following table.


| Work | memory usage |computation time | maxium number of cores $(P)$ |maxium speed|
| -------- |  -------- |-------- |-------- |-------- |
| sequential method     | $\frac{d}{\varepsilon^2}$     |$\left(\frac{d}{\varepsilon^2} \log^2 \frac{d}{\varepsilon^2}\right) \cdot (t_{{evl}} + t_{{vec}})$    |1|  $O(d)$
| Picard method     |  $P d$  |$\frac{1}{P} \frac{d}{\varepsilon^2}\log^3\left(\frac{d}{\varepsilon^2}\right) \cdot (t_{{evl}} + t_{{vec}})$   |$\frac{d}{\varepsilon^2}\log\frac{d}{\varepsilon^2}$| $O(\log^2\frac{d}{\varepsilon^2})$
| our method         | $Pd$     |$\frac{1}{P}\frac{d}{\varepsilon^2}\log^3\left(\frac{d}{\varepsilon^2}\right) \cdot (t_{{evl}} + t_{{vec}})$   |$\frac{d}{\varepsilon^2}\log^2\frac{d}{\varepsilon^2}$| $O(\log\frac{d}{\varepsilon^2})$


Our method reproduces the results of the Picard method while offering the flexibility to accelerate computations by utilizing additional computational cores, thereby improving scalability and efficiency in high parallel processing environments.


[1] Nearly d-Linear Convergence Bounds for Diffusion Models via Stochastic Localization, Joe Benton, Valentin De Bortoli, Arnaud Doucet, George Deligiannidis, 2023


[2] Accelerating Diffusion Models with Parallel Sampling: Inference at Sub-Linear Time Complexity, Haoxuan Chen, Yinuo Ren, Lexing Ying, Grant M. Rotskoff, 2024

---

> ### Author Response · Authors · 2024-11-21
>
> **Regarding empirical validation**
> -
>
> We thank the reviewers for underscoring the significance of numerical experiments in complementing our theoretical findings. While this work primarily focuses on developing a mathematical framework and conducting error analysis for parallel sampling and parallel diffusion models, we acknowledge that empirical validation would provide valuable insights into the practical implications of our results. If time allows, we plan to incorporate numerical experiments in the revised version to better connect our theoretical contributions with practical performance. In future work, we also intend to investigate the broader impact of our framework on algorithm design and analysis, with a focus on addressing practical considerations such as computational efficiency and memory usage.

---

### Meta-Review · Area_Chair_Lend · 2024-12-21

**Metareview:**

This paper introduces parallel Picard methods for sampling tasks, motivated by the success of parallel simulation in solving initial value problems in scientific computation. The proposed algorithm achieves improved iteration complexity, reducing dependence on dimension dd from O(polylog d) to O(log d), which is optimal for sampling under isoperimetry. Theoretical analysis supports these findings, highlighting the potential of leveraging simulation methods from scientific computation for dynamics-based sampling and diffusion models as data sizes grow. This paper is borderline. The main concern raised by the reviewer is the lack of experimental evaluation. While I acknowledge that purely theoretical papers do not always require experiments, the primary contribution of this work is a new algorithm based on parallel simulation rather than the analysis of existing algorithms. Given that the claimed improvement lies in iteration complexity, I agree with the reviewer that experimental validation is crucial in this case. Therefore, I recommend rejection.

**Additional Comments On Reviewer Discussion:**

This paper is borderline. The main concern raised by the reviewer (xdVr) is the lack of experimental evaluation. While I acknowledge that purely theoretical papers do not always require experiments, the primary contribution of this work is a new algorithm based on parallel simulation rather than the analysis of existing algorithms. Given that the claimed improvement lies in iteration complexity, I agree with the reviewer that experimental validation is crucial in this case. After the author rebuttal and author-reviewer discussions, all reviewers maintained their current scores, and no reviewer strongly championed the paper. Therefore, I recommend rejection.

---

### Decision · Program_Chairs · 2025-01-22

Reject